# No-Regret Learning in Dynamic Competition with Reference Effects Under Logit Demand

**Mengzi Amy Guo**
IEOR Department
UC Berkeley
mengzi_guo@berkeley.edu

**Donghao Ying**
IEOR Department
UC Berkeley
donghaoy@berkeley.edu

**Javad Lavaei**
IEOR Department
UC Berkeley
lavaei@berkeley.edu

**Zuo-Jun Max Shen**
IEOR Department
UC Berkeley
maxshen@berkeley.edu

## Abstract

This work is dedicated to the algorithm design in a competitive framework, with the primary goal of learning a stable equilibrium. We consider the dynamic price competition between two firms operating within an opaque marketplace, where each firm lacks information about its competitor. The demand follows the multinomial logit (MNL) choice model, which depends on the consumers' observed price and their reference price, and consecutive periods in the repeated games are connected by reference price updates. We use the notion of stationary Nash equilibrium (SNE), defined as the fixed point of the equilibrium pricing policy for the single-period game, to simultaneously capture the long-run market equilibrium and stability. We propose the online projected gradient ascent algorithm (OPGA), where the firms adjust prices using the first-order derivatives of their log-revenues that can be obtained from the market feedback mechanism. Despite the absence of typical properties required for the convergence of online games, such as strong monotonicity and variational stability, we demonstrate that under diminishing step-sizes, the price and reference price paths generated by OPGA converge to the unique SNE, thereby achieving the no-regret learning and a stable market. Moreover, with appropriate step-sizes, we prove that this convergence exhibits a rate of $\mathcal{O}(1/t)$.

## 1 Introduction

The memory-based reference effect is a well-studied strategic consumer behavior in marketing and economics literature, which refers to the phenomenon that consumers shape their price expectations (known as reference prices) based on the past encounters and then use them to judge the current price (see [1] for a review). A substantial body of empirical research has shown that the current demand is significantly influenced by the historical prices through reference effects (see, e.g., [2, 3, 4, 5]). Driven by the ubiquitous evidence, many studies have investigated various pricing strategies in the presence of reference effects [6, 7, 8, 9]. However, the aforementioned works have all focused on the case with a monopolistic seller, leaving the understanding of how the reference effect functions in a competition relatively limited compared to its practical importance. This issue becomes even more pronounced with the surge of e-commerce, as the increased availability of information incentivizes consumers to make comparisons among different retailers.

37th Conference on Neural Information Processing Systems (NeurIPS 2023).

Moreover, we notice that in competitive markets, the pricing problem with reference effects is further complicated by the lack of transparency, where firms are cautious about revealing confidential information to their rivals. Although the rise of digital markets greatly accelerates data transmission and promotes the information transparency, which is generally considered beneficial because of the improvement in marketplace efficiency [10], many firms remain hesitant to fully embrace such transparency for fear of losing their informational advantages. This concern is well-founded in the literature, for example the work [11] demonstrates that dealers with lower transparency levels generate higher profits than their more transparent counterparts, and [12] highlights that in reverse auctions, transparency typically enables buyers to drive the price down to the firm's marginal cost. Hence, as recommended by [13], companies should focus on the strategic and selective disclosure of information to enhance their competitive edge, rather than pursuing complete transparency.

Inspired by these real-world practices, in this article, we study the *duopoly competition with reference effects in an opaque market*, where each firm has access to its own information but does not possess any knowledge about its competitor, including their price, reference price, and demand. The consumer demand follows the multinomial logit (MNL) choice model, which naturally reflects the cross-product effects among substitutes. Furthermore, given the intertemporal characteristic of the memory-based reference effect, we consider the game in a dynamic framework, i.e., the firms engage in repeated competitions with consecutive periods linked by reference price updates. In this setting, it is natural to question *whether the firms can achieve some notion of stable equilibrium by employing common online learning algorithms to sequentially set their prices.* A vast majority of the literature on online games with incomplete information targets the problem of finding no-regret algorithms that can direct agents toward a Nash equilibrium, a stable state at which the agents have no incentive to revise their actions (see, e.g., [14, 15, 16]). Yet, the Nash equilibrium alone is inadequate to determine the stable state for our problem of interest, given the evolution of reference prices. For instance, even if an equilibrium price is reached in one period, the reference price update makes it highly probable that the firms will deviate from this equilibrium in subsequent periods. As a result, to jointly capture the market equilibrium and stability, we consider the concept of *stationary Nash equilibrium* (SNE), defined as the fixed point of the equilibrium pricing policy for the single-period game. Attaining this long-term market stability is especially appealing for firms in a competitive environment, as a stable market fosters favorable conditions for implementing efficient planning strategies and facilitating upstream operations in supply chain management [17]. In contrast, fluctuating demands necessitate more carefully crafted strategies for effective logistics management [18, 19, 20].

The concept of SNE has also been investigated in [21] and [22]. Specifically, [21] examines the long-run market behavior in a duopoly competition with linear demand and a common reference price for both products. However, compared to the MNL demand in our work, linear demand models generally fall short in addressing interdependence among multiple products [4, 23, 24]. On the other hand, [22] adopts the MNL demand with product-specific reference price formulation; yet, their analysis of long-term market dynamics relies on complete information and is inapplicable to an opaque market. In contrast, our work accommodates both the partial information setting and MNL choice model. We summarize our main contributions below:

1. *Formulation.* We introduce a duopoly competition framework with an opaque market setup that takes into account both cross-period and cross-product effects through consumer reference effects and the MNL choice model. We use the notion of *stationary Nash equilibrium* (SNE) to simultaneously depict the equilibrium price and market stability.

2. *Algorithm and convergence.* We propose a no-regret algorithm, namely the *Online Projected Gradient Ascent* (OPGA), where each firm adjusts its posted price using the first-order derivative of its log-revenue. When the firms execute OPGA with diminishing step-sizes, we show that their prices and reference prices converge to the unique SNE, leading to the long-run market equilibrium and stability. Furthermore, when the step-sizes decrease appropriately in the order of $\Theta(1/t)$, we demonstrate that the prices and reference prices converge at a rate of $\mathcal{O}(1/t)$.

3. *Analysis.* We propose a novel analysis for the convergence of OPGA by exploiting characteristic properties of the MNL demand model. General convergence results for online games typically require restrictive assumptions such as strong monotonicity [25, 14, 16] or variational stability [26, 15, 27, 28], as well as the convexity of the loss function [29, 30]. However, our problem lacks these favorable properties, rendering the existing techniques inapplicable. Additionally, compared to standard games where the underlying environment is static, the reference price evolution in this work further perplexes the convergence analysis.

4. *Managerial insights.* Our study illuminates a common issue in practice, where the firms are willing to cooperate but are reluctant to divulge their information to others. The OPGA algorithm can help address this issue by guiding the firms to achieve the SNE as if they had perfect information, while still preserving their privacy.

## 2    Related literature

Our work on the dynamic competition with reference effects in an opaque market is related to the several streams of literature.

**Modeling of reference effect.** The concept of reference effects can be traced back to the adaptation-level theory proposed by [31], which states that consumers evaluate prices against the level they have adapted to. Extensive research has been dedicated to the formulation of reference effects in the marketing literature, where two mainstream models emerge: memory-based reference price and stimulus-based reference price [4]. The memory-based reference model, also known as the internal reference price, leverages historical prices to form the benchmark (see, e.g., [3, 32, 5]). On the contrary, the stimulus-based reference model, or the external reference price, asserts that the price judgment is established at the moment of purchase utilizing current external information such as the prices of substitutable products, rather than drawing on past memories (see, e.g., [33, 2]). According to the comparative analysis by [4], among different reference models, the memory-based model that relies on a product's own historical prices offers the best fit and strongest predictive power in multi-product settings. Hence, our paper adopts this type of reference model, referred to as the brand-specific past prices formulation in [4].

**Dynamic pricing in monopolist market with reference effect.** The research on reference effects has recently garnered increasing attention in the field of operations research, particularly in monopolist pricing problems. As memory-based reference effect models give rise to the intertemporal nature, the price optimization problem is usually formulated as a dynamic program. Similar to our work, their objectives typically involve determining the long-run market stability under the optimal or heuristic pricing strategies. The classic studies by [7] and [6] show that in either discrete- or continuous-time framework, the optimal pricing policy converges and leads to market stabilization under both loss-neutral and loss-averse reference effects. More recent research on reference effects primarily concentrates on the piecewise linear demand in single-product contexts and delves into more comprehensive characterizations of myopic and optimal pricing policies (see, e.g., [34, 8]). Deviating from the linear demand, [9] and [24] employ the logit demand and analyze the long-term market behaviors under the optimal pricing policy, where the former emphasizes on consumer heterogeneity and the later innovates in the multi-product setting.

While the common assumption in the aforementioned studies is that the firm knows the demand function, another line of research tackles the problem under uncertain demand, where they couple monopolistic dynamic pricing with reference effects and online demand learning [35, 36]. Although these works include an online learning component, our paper distinguishes itself from them in two aspects. Firstly, the uncertainty that needs to be learned is situated in different areas. The works by [35] and [36] assume that the seller recognizes the structure of the demand function (i.e., linear demand) but requires to estimate the model's responsiveness parameters. By contrast, in our competitive framework, the firms are aware of their own demands but lack knowledge about their rivals that should be learned. Secondly, the objectives of [35] and [36] are to design algorithms to boost the total revenues from a monopolist perspective, whereas our algorithm aims to guide firms to reach Nash equilibrium and market stability concurrently.

**Price competition with reference effects.** Our work is pertinent to the studies on price competition with reference effects [37, 38, 21, 39, 22]. In particular, [38] is concerned with single-period price competitions under various reference price formulations. The paper [37] expands the scope to a dynamic competition with symmetric reference effects and linear demand, though their theoretical analysis on unique subgame perfect Nash equilibrium is confined to a two-period horizon. The more recent work [39] further extends the game to the multi-stage setting, where they obtain the Markov perfect equilibrium for consumers who are either loss-averse or loss-neutral. Nonetheless, unlike the discrete-time framework in [37] and this paper, [39] employs the continuous-time framework, which significantly differs from its discrete-time counterpart in terms of analytical techniques. The recent article [22] also studies the long-run market behavior and bears similarity to our work in model formulations. However, a crucial difference exists: [22] assumes a transparent market setting,

whereas we consider the more realistic scenario where the market is opaque and firms cannot access information of their competitors.

More closely related to our work, the work [21] also looks into the long-run market stability of the duopoly price competition in an opaque marketplace. However, there are two notable differences between our paper and theirs. The key distinction lies in the selection of demand function. We favor the logit demand over the linear demand used in [21], as the logit demand exhibits superior performance in the presence of reference effects [23, 9]. However, the logit model imposes challenges for convergence result since a crucial part of the analysis in [21] hinges on the demand linearity [21, Lemma 9.1], which is not satisfied by the logit demand. Second, [21] assumes a uniform reference price for both products, whereas we consider the product-specific reference price, which has the best empirical performance as illustrated in [4]. These adaptations, even though beneficial to the expressiveness and flexibility of the model, render the convergence analysis in [21] not generalizable to our setting.

**General convergence results for online games.** Our paper is closely related to the study of online games, where a typical research question is whether online learning algorithms can achieve the Nash equilibrium for multiple agents who aim to minimize their local loss functions. In this section, we review a few recent works in this field. For games with continuous actions, [14] and [40] show that the online mirror descent converges to the Nash equilibrium in strongly monotone games. The work [16] further relaxes the strong monotonicity assumption and examines the last-iterate convergence for games with unconstrained action sets that satisfy the so-called "cocoercive" condition. In addition, [26] and [15] establish the convergence of the dual averaging method under a more general condition called global variational stability, which encompasses the cocoercive condition as a subcase. More recently, [41, 27, 28] demonstrate that extra-gradient approaches, such as the optimistic gradient method, can achieve faster convergence to the Nash equilibrium in monotone and variationally stable games. We point out that in the cited literature above, either the assumption itself implies the convexity of the local loss function such as the strong monotonicity or the convergence to the Nash equilibrium additionally requires the loss function to be convex. By contrast, in our problem, the revenue function of each firm is not concave in its price and does not satisfy either of the properties listed above. Additionally, incorporating the reference effect further complicates the analysis, as the standard notion of Nash equilibrium is insufficient to characterize convergence due to the dynamic nature of reference price.

## 3 Problem formulation

### 3.1 MNL demand model with reference effects

We study a duopoly price competition with reference price effects, where two firms each offer a substitutable product, labeled as $H$ and $L$, respectively. Both firms set prices simultaneously in each period throughout an infinite-time horizon. To accommodate the interaction between the two products, we employ a multinomial logit (MNL) model, which inherently captures such cross-product effects. The consumers' utility at period $t$, which depends on the posted price $p_i^t$ and reference price $r_i^t$, is defined as:

$$U_i\left(p_i^t, r_i^t\right) = u_i\left(p_i^t, r_i^t\right) + \epsilon_i^t = a_i - b_i \cdot p_i^t + c_i \cdot \left(r_i^t - p_i^t\right) + \epsilon_i^t, \quad \forall i \in \{H, L\}, \qquad (1)$$

where $u_i(p_i^t, r_i^t)$ is the deterministic component, and $(a_i, b_i, c_i)$ are the given parameters. In addition, the notation $\epsilon_i^t$ denotes the random fluctuation following the i.i.d. standard Gumbel distribution. According to the random utility maximization theory [42], the demand/market share at period $t$ with the posted price $\mathbf{p}^t = (p_H^t, p_L^t)$ and reference price $\mathbf{r}^t = (r_H^t, r_L^t)$ for product $i \in \{H, L\}$ is given by

$$d_i\left(\mathbf{p}^t, \mathbf{r}^t\right) = d_i\left((p_i^t, p_{-i}^t), (r_i^t, r_{-i}^t)\right) = \frac{\exp\left(u_i(p_i^t, r_i^t)\right)}{1 + \exp\left(u_i(p_i^t, r_i^t)\right) + \exp\left(u_{-i}(p_{-i}^t, r_{-i}^t)\right)}, \qquad (2)$$

where the subscript $-i$ denotes the other product besides product $i$. Consequently, the expected revenue for each firm/product at period $t$ can be expressed as

$$\Pi_i(\mathbf{p}^t, \mathbf{r}^t) = \Pi_i\left((p_i^t, p_{-i}^t), (r_i^t, r_{-i}^t)\right) = p_i^t \cdot d_i(\mathbf{p}^t, \mathbf{r}^t), \quad \forall i \in \{H, L\}. \qquad (3)$$

The interpretation of parameters $(a_i, b_i, c_i)$ in Eq. (1) is as follows. For product $i \in \{H, L\}$, $a_i$ refers to product's intrinsic value, $b_i$ represents consumers' responsiveness to price, also known as price

sensitivity, and $c_i$ corresponds to the reference price sensitivity. When the offered price exceeds the internal reference price $(r_i^t < p_i^t)$, consumers perceive it as a loss or surcharge, whereas the price below this reference price $(r_i^t > p_i^t)$ is regarded as a gain or discount. These sensitivity parameters are assumed to be positive, i.e., $b_i, c_i > 0$ for $i \in \{H, L\}$, which aligns with consumer behaviors towards substitutable products and has been widely adopted in similar pricing problems (see, e.g., [34, 8, 21, 24, 22]). Precisely, this assumption guarantees that an increase in $p_i^t$ would result in a lower consumer utility for product $i$, which in turn reduces its demand $d_i(\mathbf{p}^t, \mathbf{r}^t)$ and increases the demand of the competing product $d_{-i}(\mathbf{p}^t, \mathbf{r}^t)$. Conversely, a rise in the reference price $r_i^t$ increases the utility for product $i$, consequently influencing the consumer demands in the opposite direction.

We stipulate the feasible range for price and reference price to be $\mathcal{P} = [\underline{p}, \overline{p}]$, where $\underline{p}, \overline{p} > 0$ denote the price lower bound and upper bound, respectively. This boundedness of prices is in accordance with real-world price floors or price ceilings, whose validity is further reinforced by its frequent use in the literature on price optimization with reference effects (see, e.g., [34, 8, 21]).

We formulate the reference price using the brand-specific past prices (*PASTBRSP*) model proposed by [4], which posits that the reference price is product-specific and memory-based. This model is preferred over other reference price models evaluated in [4], as it exhibits superior performance in terms of fit and prediction. In particular, the reference price for product $i$ is constructed by applying exponential smoothing to its own historical prices, where the memory parameter $\alpha \in [0, 1]$ governs the rate at which the reference price evolves. Starting with an initial reference price $r_0$, the reference price update for each product at period $t$ can be described as

$$r_i^{t+1} = \alpha \cdot r_i^t + (1 - \alpha) \cdot p_i^t, \quad \forall i \in \{H, L\}, \quad t \geq 0. \tag{4}$$

The exponential smoothing technique is among the most prevalent and empirically substantiated reference price update mechanism in the existing literature (see, e.g., [7, 1, 6, 34, 8]). We remark that the theories established in this work are readily generalizable to the scenario of time-varying memory parameters $\alpha$, and we present the static setting only for the sake of brevity.

## 3.2 Opaque market setup

In this study, we consider the partial information setting where the firms possess no knowledge of their competitors, their own reference prices, as well as the reference price update scheme. Each firm $i$ is only aware of its own sensitivity parameters $b_i, c_i$, and its previously posted price. Under this configuration, the firms cannot directly compute their demands from the expression in Eq. (2). However, it is legitimate to assume that each firm can access its own last-period demands through market feedback, i.e., determining the demand as the received revenue divided by the posted price. We point out that, in this non-transparent and non-cooperative market, the presence of feedback mechanisms is crucial for price optimization.

## 3.3 Market equilibrium and stability

The goal of our paper is to find a simple and intuitive pricing mechanism for firms so that the market equilibrium and stability can be achieved in the long-run while protecting firms' privacy. Before introducing the equilibrium notion considered in this work, we first define the *equilibrium pricing policy* denoted by $\mathbf{p}^\star(\mathbf{r}) = \big(p_H^\star(\mathbf{r}), p_L^\star(\mathbf{r})\big)$, which is a function that maps reference price to price and achieves the pure strategy Nash equilibrium in the single-period game. Mathematically, the equilibrium pricing policy satisfies that

$$p_i^\star(\mathbf{r}) = \underset{p_i \in \mathcal{P}}{\arg\max} \ p_i \cdot d_i\big((p_i, p_{-i}^\star(\mathbf{r})), \mathbf{r}\big), \quad \forall i \in \{H, L\}. \tag{5}$$

Next, we formally introduce the concept of stationary Nash equilibrium, which is utilized to jointly characterize the market equilibrium and stability.

**Definition 3.1 (Stationary Nash equilibrium)** *A point $\mathbf{p}^{\star\star}$ is considered a stationary Nash equilibrium (SNE) if $\mathbf{p}^\star(\mathbf{p}^{\star\star}) = \mathbf{p}^{\star\star}$, i.e., the equilibrium price is equal to its reference price.*

The notion of SNE has also been studied in [21, 22]. From Eq. (5) and Definition 3.1, we observe that an SNE possesses the following two properties:

- **Equilibrium.** The revenue function for each firm $i \in \{H, L\}$ satisfies $\Pi_i\big((p_i, p_{-i}^{\star\star}), \mathbf{p}^{\star\star}\big) \leq \Pi_i\big(\mathbf{p}^{\star\star}, \mathbf{p}^{\star\star}\big)$ for all $p_i \in \mathcal{P}$, i.e., when the reference price and firm $-i$'s price are equal to the SNE price, the best-response price for firm $i$ is the SNE price $p_i^{\star\star}$.

- **Stability.** If the price and the reference price attain the SNE at some period $t$, the reference price remains unchanged in the following period, i.e., $\mathbf{p}^t = \mathbf{r}^t = \mathbf{p}^{\star\star}$ implies that $\mathbf{r}^{t+1} = \mathbf{p}^{\star\star}$.

As a result, when the market reaches the SNE, the firms have no incentive to deviate, and the market remains stable in subsequent competitions. The following proposition states the uniqueness of the SNE and characterizes the boundedness of the SNE.

**Proposition 3.1** *There exists a unique stationary Nash equilibrium, denoted by $\mathbf{p}^{\star\star} = (p_H^{\star\star}, p_L^{\star\star})$. In addition, it holds that*

$$\frac{1}{b_i + c_i} < p_i^{\star\star} < \frac{1}{b_i + c_i} + \frac{1}{b_i} W\left(\frac{b_i}{b_i + c_i} \exp\left(a_i - \frac{b_i}{b_i + c_i}\right)\right), \quad \forall i \in \{H, L\}, \qquad (6)$$

*where $W(\cdot)$ is the Lambert W function (see definition in Eq. (22)).*

Without loss of generality, we assume that the feasible price range $\mathcal{P}^2 = [\underline{p}, \overline{p}]^2$ is sufficiently large to contain the unique SNE, i.e., $\mathbf{p}^{\star\star} \in [\underline{p}, \overline{p}]^2$. Proposition 3.1 provides a quantitative characterization for this assumption: it suffices to choose the price lower bound $\underline{p}$ to be any real number between $\big(0, \min_{i \in \{H,L\}} \{1/(b_i + c_i)\}\big]$, and the price upper bound $\overline{p}$ can be any value such that

$$\overline{p} \geq \max_{i \in \{H,L\}} \left\{\frac{1}{b_i + c_i} + \frac{1}{b_i} W\left(\frac{b_i}{b_i + c_i} \exp\left(a_i - \frac{b_i}{b_i + c_i}\right)\right)\right\}. \qquad (7)$$

This assumption is mild as the bound in Eq. (7) is independent of both the price and reference price, and it does not grow exponentially fast with respect to any parameters. Hence, there is no need for $\overline{p}$ to be excessively large. For example, when $a_H = a_L = 10$ and $b_H = b_L = c_H = c_L = 1$, Eq. (7) becomes $\overline{p} \geq 7.3785$. We refer the reader to Appendix B for further discussions on the structure and computation of the equilibrium pricing policy and SNE, as well as the proof of Proposition 3.1.

## 4 No-regret learning: Online Projected Gradient Ascent

In this section, we examine the long-term dynamics of the the price and reference price paths to determine if the market stabilizes over time. As studied in [22], under perfect information, the firms operating with full rationality will follow the equilibrium pricing policy in each period, while those functioning with bounded rationality will adhere to the best-response pricing policy throughout the planning horizon. In both situations, the market would stabilize in the long run. However, with partial information, the firms are incapable of computing either the equilibrium or the best-response policies due to the unavailability of reference prices and their competitor's price. Thus, one viable strategy to boost firms' revenues is to dynamically modify prices in response to market feedback.

In light of the success of gradient-based algorithms in the online learning literature (see, e.g., [43, 44, 45]), we propose the Online Projected Gradient Ascent (OPGA) method, as outlined in Algorithm 1. Specifically, in each period, both firms update their current prices using the first-order derivatives of their log-revenues with the same learning rate (see Eq. (9)). It is noteworthy that the difference between the derivatives of the log-revenue and the standard revenue is a scaling factor equal to the revenue itself, i.e.,

$$\frac{\partial \log\big(\Pi_i(\mathbf{p}, \mathbf{r})\big)}{\partial p_i} = \frac{1}{\Pi_i(\mathbf{p}, \mathbf{r})} \cdot \frac{\partial\big(\Pi_i(\mathbf{p}, \mathbf{r})\big)}{\partial p_i}, \quad \forall i \in \{H, L\}. \qquad (8)$$

Therefore, the price update in Algorithm 1 can be equivalently viewed as an adaptively regularized gradient ascent using the standard revenue function, where the regularizer at period $t$ is $\big(1/\Pi_H(\mathbf{p}^t, \mathbf{r}^t), 1/\Pi_L(\mathbf{p}^t, \mathbf{r}^t)\big)$.

We highlight that by leveraging the structure of MNL model, each firm $i$ can obtain the derivative of its log-revenue $D_i^t$ in Eq. (9) through the market feedback mechanism, i.e., last-period demand. The firms do not need to know their own reference price when executing the OPGA algorithm, and the reference price update in Line 6 is automatically performed by the market. In fact, computing the

---

**Algorithm 1** Online Projected Gradient Ascent (OPGA)

---

1: **Input:** Initial reference price $\mathbf{r}^0 = (r_H^0, r_L^0)$, initial price $\mathbf{p}^0 = (p_H^0, p_L^0)$, and step-sizes $\{\eta^t\}_{t \geq 0}$.
2: **for** $t = 0, 1, 2, \ldots$ **do**
3:     **for** $i \in \{H, L\}$ **do**
4:         Compute derivative $D_i^t$ from price and demand of firm $i$ at period $t$:

$$D_i^t \leftarrow \frac{\partial \log \left( \Pi_i(\mathbf{p}^t, \mathbf{r}^t) \right)}{\partial p_i} = \frac{1}{p_i^t} + (b_i + c_i) \cdot d_i(\mathbf{p}^t, \mathbf{r}^t) - (b_i + c_i). \tag{9}$$

5:         Update posted price: $p_i^{t+1} \leftarrow \mathrm{Proj}_{\mathcal{P}} \left( p_i^t + \eta^t D_i^t \right)$.
6:         Reference price update: $\mathbf{r}^{t+1} \leftarrow \alpha \mathbf{r}^t + (1 - \alpha)\mathbf{p}^t$.

---

derivative $D_i$ is identical to querying the first-order oracle, which is a common assumption in the optimization literature [46]. Further, this derivative can also be acquired through a minor perturbation of the posted price, even when the firms lack access to historical prices and market feedback, making the algorithm applicable in a variety of scenarios.

There are two potential ways to analyze Algorithm 1 using well-established theories. Below, we briefly introduce these methods and the underlying challenges, while referring readers to Appendix A for a more detailed discussion.

- First, by adding two virtual firms to represent the reference price, the game can be converted into a standard four-player online game, effectively eliminating the evolution of the underlying state (reference price). However, the challenge in analyzing this four-player game arises from the fact that, while real firms have the flexibility to dynamically adjust their step-sizes, the learning rate for virtual firms is fixed to the constant $(1 - \alpha)$, where $\alpha$ is the memory parameter for reference price updates. This disparity hinders the direct application of the existing results from multi-agent online learning literature, as these results typically require the step-sizes of all agents either diminish at comparable rates or remain as a small enough constant [47, 48, 14, 15].

- The second approach involves translating the OPGA algorithm into a discrete nonlinear system by treating $(\mathbf{p}^{t+1}, \mathbf{r}^{t+1})$ as a vector-valued function of $(\mathbf{p}^t, \mathbf{r}^t)$, i.e., $(\mathbf{p}^{t+1}, \mathbf{r}^{t+1}) = \mathbf{f}(\mathbf{p}^t, \mathbf{r}^t)$ for some function $\mathbf{f}(\cdot)$. In this context, analyzing the convergence of Algorithm 1 is equivalent to examining the stability of the fixed point of $\mathbf{f}(\cdot)$, which is related to the spectral radius of the Jacobian matrix $\nabla \mathbf{f}(\mathbf{p}^{\star\star}, \mathbf{p}^{\star\star})$ [49, 50]. However, the SNE lacks a closed-form expression, making it difficult to calculate the eigenvalues of $\nabla \mathbf{f}(\mathbf{p}^{\star\star}, \mathbf{p}^{\star\star})$. In addition, the function $\mathbf{f}(\cdot)$ is non-smooth due to the presence of the projection operator and can become non-stationary when the firms adopt time-varying step-sizes, such as diminishing step-sizes. Moreover, typical results in dynamical systems only guarantee local convergence [51], i.e., the asymptotic stability of the fixed point, whereas our goal is to establish the global convergence of both the price and reference price.

In the following section, we show that the OPGA algorithm with diminishing step-sizes converges to the unique SNE by exploiting characteristic properties of our model. This convergence result indicates that the OPGA algorithm provably achieves the no-regret learning, i.e., in the long run, the algorithm performs at least as well as the best fixed action in hindsight. However, it is essential to note that the reverse is not necessarily true: being no-regret does not guarantee the convergence at all, let alone the convergence to an equilibrium (see, e.g., [52, 53]). In fact, beyond the finite games, the agents may exhibit entirely unpredictable and chaotic behaviors under a no-regret policy [54].

## 5 Convergence results

In this section, we investigate the convergence properties of the OPGA algorithm. First, in Theorem 5.1, we establish the global convergence of the price path and reference price to the unique SNE under diminishing step-sizes. Subsequently, in Theorem 5.2, we further show that this convergence exhibits a rate of $\mathcal{O}(1/t)$, provided that the step-sizes are selected appropriately. The primary proofs for this section can be found in Appendices C and D, while supporting lemmas are located in Appendix F.

**Theorem 5.1 (Global convergence)** *Let the step-sizes $\{\eta^t\}_{t\geq 0}$ be a non-increasing sequence such that $\lim_{t\to\infty} \eta^t = 0$ and $\sum_{t=0}^{\infty} \eta^t = \infty$ hold. Then, the price paths and reference price paths generated by Algorithm 1 converge to the unique stationary Nash equilibrium.*

Theorem 5.1 demonstrates the global convergence of the OPGA algorithm to the SNE, thus ensuring the market equilibrium and stability in the long-run. Compared to [22] that establishes the convergence to SNE under the perfect information setting, Theorem 5.1 ensures that such convergence can also be achieved in an opaque market where the firms are reluctant to share the information with their competitors. The only condition we need for the convergence is diminishing step-sizes such that $\lim_{t\to\infty} \eta^t = 0$ and $\sum_{t=0}^{\infty} \eta^t = \infty$. This assumption is widely adopted in the study of online games (see, e.g., [15, 14, 40]). Since the firms will likely become more familiar with their competitors through repeated competitions, it is reasonable for the firms to gradually become more conservative in adjusting their prices and decrease their learning rates.

As discussed in Sections 2 and 4, the existing methods in the online game and dynamical system literatures are not applicable to our problem due to the absence of standard structural properties and the presence of the underlying dynamic state, i.e., reference price. Consequently, we develop a novel analysis to prove Theorem 5.1, leveraging the characteristic properties of the MNL demand model. We provide a proof sketch below, with the complete proof deferred to Appendix C. Our proof consists of two primary parts:

- In **Part 1** (Appendix C.1), we show that the price path $\{\mathbf{p}^t\}_{t\geq 0}$ would enter the neighborhood $N_\epsilon^1 := \{\mathbf{p} \in \mathcal{P}^2 \mid \varepsilon(\mathbf{p}) < \epsilon\}$ infinitely many times for any $\epsilon > 0$, where $\varepsilon(\cdot)$ is a weighted $\ell_1$ distance function defined as $\varepsilon(\mathbf{p}) := |p_H^{\star\star} - p_H|/(b_H + c_H) + |p_L^{\star\star} - p_L|/(b_L + c_L)$. To prove Part 1, we divide $\mathcal{P}^2$ into four quadrants with $\mathbf{p}^{\star\star}$ as the origin. Then, employing a contradiction-based argument, we suppose that the price path only visits $N_\epsilon^1$ finitely many times. Yet, we show that this forces the price path to oscillate between adjacent quadrants and ultimately converge to the SNE, which violates the initial assumption.

- In **Part 2** (Appendix C.2), we show that when the price path $\{\mathbf{p}^t\}_{t\geq 0}$ enters the $\ell_2$-neighborhood $N_\epsilon^2 := \{\mathbf{p} \in \mathcal{P}^2 \mid \|\mathbf{p} - \mathbf{p}^{\star\star}\|_2 < \epsilon\}$ for some sufficiently small $\epsilon > 0$ and with small enough step-sizes, the price path will remain in $N_\epsilon^2$ in subsequent periods. The proof of Part 2 relies on a local property of the MNL demand model around the SNE, with which we demonstrate that the price update provides adequate ascent to ensure the price path stays within the target neighborhood.

Owing to the equivalence between different distance functions in Euclidean spaces [55], these two parts jointly imply the convergence of the price path to the SNE. Since the reference price is equal to an exponential smoothing of the historical prices, the convergence of the reference price path follows that of the price path.

It is worth noting that [21, Theorem 5.1] also employs a two-part proof and shows the asymptotic convergence of online mirror descent [56, 57] to the SNE in the linear demand setting. However, their analysis heavily depends on the linear structure of the demand, which ensures that a property similar to the variational stability is globally satisfied [21, Lemma 9.1]. In contrast, such properties no longer hold for the MNL demand model considered in this paper. Therefore, we come up with two distinct distance functions for the two parts of the proof, respectively. Moreover, the proof by [21] relies on an assumption concerning the relationship between responsiveness parameters, whereas our convergence analysis only requires the minimum assumption that the responsiveness parameters are positive (otherwise, the model becomes counter-intuitive for substitutable products).

**Theorem 5.2 (Convergence rate)** *When both firms adopt Algorithm 1, there exists a sequence of step-sizes $\{\eta^t\}_{t\geq 0}$ with $\eta^t = \Theta(1/t)$ and constants $d_p, d_r > 0$ such that*

$$\left\|\mathbf{p}^{\star\star} - \mathbf{p}^t\right\|_2^2 \leq \frac{d_p}{t}, \quad \left\|\mathbf{p}^{\star\star} - \mathbf{r}^t\right\|_2^2 \leq \frac{d_r}{t}, \quad \forall t \geq 1. \tag{10}$$

Theorem 5.2 improves the result of Theorem 5.1 by demonstrating the non-asymptotic convergence rate for the price and reference price. Although non-concavity usually anticipates slower convergence, our rate of $\mathcal{O}(1/t)$ matches [21, Theorem 5.2] that assumes linear demand and exhibits concavity.

The proof of Theorem 5.2 is primarily based on the local convergence rate when the price vector remains in a specific neighborhood $N_{\epsilon_0}^2$ of the SNE after some period $T_{\epsilon_0}$. As the choice $\eta^t = \Theta(1/t)$

ensures that $\lim_{t\to\infty} \eta^t = 0$ and $\sum_{t=0}^{\infty} \eta^t = \infty$, the existence of such a period $T_{\epsilon_0}$ is guaranteed by Theorem 5.1. Utilizing an inductive argument, we first show that the difference between price and reference price decreases at a faster rate of $\mathcal{O}(1/t^2)$, i.e., $\|\mathbf{r}^t - \mathbf{p}^t\|_2^2 = \mathcal{O}(1/t^2)$, $\forall t \geq 1$ (see Eq. (67)). Then, by exploiting a local property around the SNE (Lemma F.3), we use another induction to establish the convergence rate of the price path after it enters the neighborhood $N_{\epsilon_0}^2$. In the meanwhile, the convergence rate of the reference price path can be determined through a triangular inequality: $\|\mathbf{p}^{\star\star} - \mathbf{r}^t\|_2^2 = \|\mathbf{p}^{\star\star} - \mathbf{p}^t + \mathbf{p}^t - \mathbf{r}^t\|_2^2 \leq 2\|\mathbf{p}^{\star\star} - \mathbf{p}^t\|_2^2 + 2\|\mathbf{p}^t - \mathbf{r}^t\|_2^2$. Finally, due to the boundedness of the feasible price range $\mathcal{P}^2$, we can obtain the global convergence rate for all $t \geq 1$ by choosing sufficiently large constants $d_p$ and $d_r$.

**Remark 5.3 (Constant step-sizes)** *The convergence analysis presented in this work can be readily extended to accommodate constant step-sizes. Particularly, our theoretical framework supports the selection of $\eta^t \equiv \mathcal{O}(\epsilon_0)$, where $\epsilon_0$ refers to the size of the neighborhood utilized in Theorem 5.2. When $\eta^t \equiv \mathcal{O}(\epsilon_0)$, an approach akin to **Part 1** can be employed to demonstrate that the price path enters the neighborhood $N_{\epsilon}^1$ infinitely many times for any $\epsilon \geq \mathcal{O}(\epsilon_0)$. Subsequently, by exploiting the local property in a manner similar to **Part 2**, we can establish that once the price path enters the neighborhood $N_{\epsilon}^2$ for some $\epsilon = \mathcal{O}(\epsilon_0)$, it will remain within that neighborhood.*

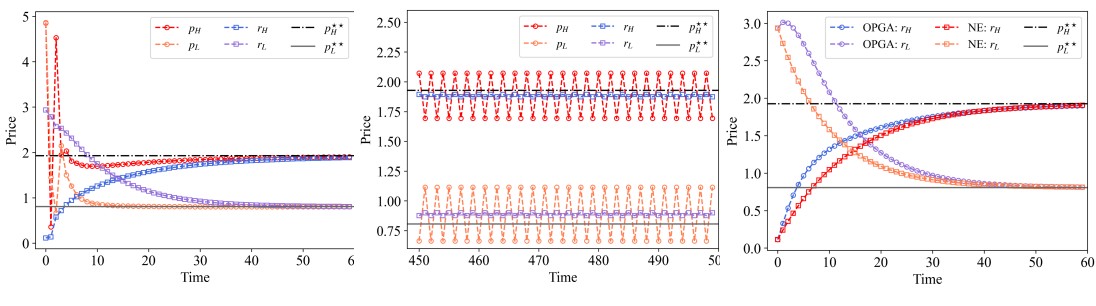

(a) Convergence with $\eta^t = 3/t$.   (b) Cyclic pattern with $\eta^t = 1$.   (c) OPGA vs. equilibrium policy.

Figure 1: Price and reference price paths for Examples 1, 2, and 3, where the parameters are $(a_H, b_H, c_H) = (8.70, 2.00, 0.82)$, $(a_L, b_L, c_L) = (4.30, 1.20, 0.32)$, $(r_H^0, r_L^0) = (0.10, 2.95)$, $(p_H^0, p_L^0) = (4.85, 4.86)$, and $\alpha = 0.90$.

We perform two numerical experiments, differentiated solely by their sequences of step-sizes, to highlight the significance of step-sizes in achieving the convergence. In particular, Example 1 (see Figure 1a) corroborates Theorem 5.1 by demonstrating that the price and reference price trajectories converge to the unique SNE when we choose diminishing step-sizes that fulfill the criteria specified in Theorem 5.1. By comparison, the over-large constant step-sizes employed in Example 2 (see Figure 1b) fails to ensure convergence, leading to cyclic patterns in the long run. Moreover, in Example 3 (see Figure 1c), we compare the OPGA algorithm and the repeated application of the equilibrium pricing policy in Eq. (5) by plotting the reference price trajectories produced by both approaches. Figure 1c conveys that the two algorithms reach the SNE at a comparable rate, which indicates that the OPGA algorithm allows firms to attain the market equilibrium and stability as though they operate under perfect information, while preventing excessive information disclosure.

# 6 Extensions

In previous sections, we assume that each firm $i$ possesses accurate information regarding its sensitivity parameters $(b_i, c_i)$ as well as its realized market share $d_i^t$. This is equivalent to having access to an exact first-order oracle. However, a more practical scenario to consider is one where the firm can only obtain a rough approximation for its market share and needs to estimate the sensitivities from historical data. This would bring extra noise to the computation of the first-order derivative $D_i^t$ in Eq. (9). In this section, we first elaborate on the feasibility of estimating the market share and sensitivities from realistic data. Then, we discuss the impact of an inexact first-order oracle on the convergence.

We consider the approximation of market share and the calibration of sensitivities under both *uncensored* and *censored* data cases. With uncensored data, both purchase and no-purchase data are

available, a scenario analogous to firms selling substitute goods on third-party online retail platforms like Amazon. The platform can track the non-purchase statistics by monitoring consumers who visited its website but did not purchase any product. Consequently, this facilitates a direct estimation of the total market size and, when complemented with sales quantities, allows for a precise computation of the firm's market share. Additionally, sensitivity parameters can be efficiently calibrated through the classical Maximum Likelihood Estimation (MLE) method. Due to the concavity of the log-likelihood function for the MNL model with respect to its parameters, many numerical algorithms (e.g., Newton-Raphson, BHHH-2 [58], and the steepest ascent) are guaranteed to improve at each iteration and achieve fast convergence [59, 60].

In many cases, the traditional transaction data only captures the realized demand, with non-purchases typically unrecorded. This scenario is known as the censored data case. In this case, the sensitives and the total market size of the MNL model can be estimated via the generalized expectation-maximization (GEM) gradient method proposed by [59], which is an iterative algorithm that is inspired by the expectation-maximization (EM) approach. The capability to approximate the sensitivity parameters, the market share, and thereby the gradient $D_i^t$ enhances the credibility and practicability of the OPGA algorithm, suggesting its great potential for broader applications in retailing.

In the presence of approximation errors, firms cannot precisely compute the derivative $D_i^t$. Hence, the convergence results in Section 5 are not directly applicable. Indeed, if the errors are disruptive enough, one can anticipate Algorithm 1 to possibly not show any convergent behavior. However, if the errors are uniformly bounded by some small number $\delta$, the following theorem demonstrates that both the price and reference price paths converge to a $\mathcal{O}(\delta)$-neighborhood of the unique SNE.

**Theorem 6.1 (Inexact first-order oracle)** *Suppose both firms only have access to an inexact first-order oracle such that $|D_i^t - \partial \log(\Pi_i(\mathbf{p}^t, \mathbf{r}^t))/\partial p_i| \leq \delta, \ \forall i \in \{H, L\}$ and $\forall t \geq 0$. Let the step-sizes $\{\eta^t\}_{t \geq 0}$ be a non-increasing sequence such that $\lim_{t \to \infty} \eta^t = 0$ and $\sum_{t=0}^{\infty} \eta^t = \infty$ hold. Then, the price paths reference price paths generated by Algorithm 1 converge to a neighborhood with radius $\mathcal{O}(\delta)$ of the unique SNE.*

We remark that the inexact first-order oracle studied in our setting is different from the stochastic gradient, which generally assumes a zero-mean noise with finite variance. In that case, it is possible to derive the convergence to a limiting point in expectation or with high probability. In contrast, the noise in $D_i^t$ is a kind of approximation error without any distributional properties. Therefore, with the step-sizes defined in Theorem 6.1, we expect the price and reference price paths to converge to the neighborhood of the SNE, but continue to fluctuate around the neighborhood without admitting a limiting point.

The proof of Theorem 6.1 is based upon Theorem 5.1. Specifically, we demonstrate that the inexact gradient still guides the price path toward the SNE if the magnitude of the true gradient dominates the noise. Conversely, if noise levels are comparable with the true gradient, we show that the price path is already close to the SNE. The formal proof of Theorem 6.1 can be found in Appendix E

## 7 Conclusion and future work

This article considers the price competition with reference effects in an opaque market, where we formulate the problem as an online game with an underlying dynamic state, known as reference price. We accomplish the goal of simultaneously capturing the market equilibrium and stability via proposing the no-regret algorithm named OPGA and providing the theoretical guarantee for its global convergence to the SNE. In addition, under appropriate step-sizes, we show that the OPGA algorithm has the convergence rate of $\mathcal{O}(1/t)$.

In this paper, we focus on the symmetric reference effects, where consumers exhibit equal sensitivity to the gains and losses. As empirical studies suggest that consumers may display asymmetric reactions [61, 32, 62], one possible extension involves accounting for asymmetric reference effects displayed by loss-averse and gain-seeking consumers. Additionally, while our study considers the duopoly competition between two firms, it would be valuable to explore the more general game involving $n$ players. Lastly, our research is based on a deterministic setting and pure strategy Nash equilibrium. Another intriguing direction for future work is to investigate mixed-strategy learning [63] in dynamic competition problems.

## Acknowledgement

We extend our gratitude to the anonymous reviewers for their insightful comments. This research was particularly supported by NSF China Grant 71991462.

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

# Appendix A    Discussion about alternative methods

As we briefly mentioned in Section 3, our problem can also be translated into a four-player online game or a dynamical system. In this section, we will discuss these alternative methods and explain why existing tools from the literature of online game and dynamical system cannot be applied.

## A.1    Four-player online game formulation

By viewing the underlying state variables $(r_H, r_L)$ as virtual players that undergo deterministic transitions, we are able to convert our problem into a general four-player game. Specifically, we can construct revenue functions $R_i(p_i, r_i)$ for $i \in \{H, L\}$ and a common sequence of step-sizes $\{\eta_r^t\}_{t \geq 0}$ for the two virtual players such that when firms $H, L$ and virtual players implement OPGA using the corresponding step-sizes, the resulting price path $\{\mathbf{p}^t\}_{t \geq 0}$ and reference price path $\{\mathbf{r}^t\}_{t \geq 0}$ recover those generated by Algorithm 1. The functions $R_i(\cdot, \cdot), \forall i \in \{H, L\}$ and step-sizes $\{\eta_r^t\}_{t \geq 0}$ are specified as follows

$$R_i(p_i, r_i) = -\frac{1}{2}r_i^2 + r_i \cdot p_i, \quad \forall i \in \{H, L\};$$
$$\eta_r^t \equiv 1 - \alpha, \quad \forall t \geq 0. \tag{11}$$

Then, given the initial reference price $\mathbf{r}^0$ and initial price $\mathbf{p}^0$, the four players respectively update their states through the online projected gradient ascent specified as follows, for $t \geq 0$:

$$\begin{cases} p_H^{t+1} = \text{Proj}_{\mathcal{P}}\left(p_H^t + \eta^t \cdot \frac{\partial \log\left(\Pi_H(\mathbf{p}^t, \mathbf{r}^t)\right)}{\partial p_H}\right) = \text{Proj}_{\mathcal{P}}\left(p_H^t + \eta^t D_H^t\right), \\[2mm] p_L^{t+1} = \text{Proj}_{\mathcal{P}}\left(p_L^t + \eta^t \cdot \frac{\partial \log\left(\Pi_L(\mathbf{p}^t, \mathbf{r}^t)\right)}{\partial p_L}\right) = \text{Proj}_{\mathcal{P}}\left(p_L^t + \eta^t D_L^t\right), \\[2mm] r_H^{t+1} = \text{Proj}_{\mathcal{P}}\left(r_H^t + \eta_r^t \cdot \frac{\partial R_H(p_H^t, r_H^t)}{\partial r_H}\right) = \text{Proj}_{\mathcal{P}}\left(\alpha r_H^t + (1-\alpha)p_H^t\right), \\[2mm] r_L^{t+1} = \text{Proj}_{\mathcal{P}}\left(r_L^t + \eta_r^t \cdot \frac{\partial R_L(p_L^t, r_L^t)}{\partial r_L}\right) = \text{Proj}_{\mathcal{P}}\left(\alpha r_L^t + (1-\alpha)p_L^t\right). \end{cases} \tag{12}$$

It can be easily seen that the pure strategy Nash equilibrium of this four-player static game is equivalent to the SNE of the original two-player dynamic game, as defined in Definition 3.1. However, even after converting to the static game, no general convergence results are readily applicable in this setting. One obstacle on the convergence analysis comes from the absence of critical properties, such as monotonicity of the static game [14, 16] or variational stability at its Nash equilibrium [26, 15].

Meanwhile, anther obstacle stems from the asynchronous updates for real firms (price players) and virtual firms (reference price players). While the real firms have the flexibility in adopting time-varying step-sizes, the updates for the two virtual firms in Eq. (12) stick to the constant step-size of $(1 - \alpha)$. As a result, this rigidity of virtual players perplexes the analysis, given that the typical convergence results of online games call for the step-sizes of multiple players to have the same pattern (all diminishing or constant step-sizes) [47, 48, 14, 15].

## A.2    Dynamical system formulation

The study of the limiting behavior of a competitive gradient-based learning algorithm is related to dynamical system theories [64]. In fact, the update of Algorithm 1 can be viewed as a nonlinear dynamical system. Assume a constant step-size is employed, i.e., $\eta^t \equiv \eta, \forall t \geq 0$. Then, Lines 5 and 6 in Algorithm 1 are equivalent to the dynamical system

$$(\mathbf{p}^{t+1}, \mathbf{r}^{t+1}) = \mathbf{f}(\mathbf{p}^t, \mathbf{r}^t), \quad \forall t \geq 0, \tag{13}$$

where $\mathbf{f}(\cdot)$ is a vector-valued function defined as

$$\mathbf{f}(\mathbf{p}, \mathbf{r}) := \begin{pmatrix} \mathrm{Proj}_{\mathcal{P}}\left(p_H + \eta\left(\dfrac{1}{p_H} + (b_H + c_H) \cdot d_H(\mathbf{p}, \mathbf{r}) - (b_H + c_H)\right)\right) \\ \mathrm{Proj}_{\mathcal{P}}\left(p_L + \eta\left(\dfrac{1}{p_L} + (b_L + c_L) \cdot d_L(\mathbf{p}, \mathbf{r}) - (b_L + c_L)\right)\right) \\ \mathrm{Proj}_{\mathcal{P}}\left(\alpha r_H + (1 - \alpha)p_H\right) \\ \mathrm{Proj}_{\mathcal{P}}\left(\alpha r_L + (1 - \alpha)p_L\right) \end{pmatrix}. \tag{14}$$

Under the assumption that $\mathbf{p}^{\star\star} \in \mathcal{P}^2$, it is evident that $\mathbf{p}^{\star\star}$ is the unique fixed point of the system in Eq. (13). Generally, fixed points can be categorized into three classes:

- *asymptotically stable*, when all nearby solutions converge to it,

- *stable*, when all nearby solutions remain in close proximity,

- *unstable*, when almost all nearby solutions diverge away from the fixed point.

Hence, if we can demonstrate the asymptotic stability of $\mathbf{p}^{\star\star}$, we can at least prove the local convergence of the price and reference price.

Standard dynamical systems theory [49, 50] states that $\mathbf{p}^{\star\star}$ is asymptotically stable if the spectral radius of the Jacobian matrix $\nabla \mathbf{f}(\mathbf{p}^{\star\star}, \mathbf{p}^{\star\star})$ is strictly less than one. However, computing the spectral radius is not straightforward. The primary challenge stems from the fact that, while the entries of $\nabla \mathbf{f}(\mathbf{p}^{\star\star}, \mathbf{p}^{\star\star})$ contain $\mathbf{p}^{\star\star}$ and $d_i(\mathbf{p}^{\star\star}, \mathbf{p}^{\star\star})$, there is no closed-form expression for $\mathbf{p}^{\star\star}$.

Apart from the above issue, it is worth noting that the function $f(\cdot)$ is not globally smooth due to the presence of the projection operator. Furthermore, the function $f(\cdot)$ also depends on the step-size $\eta$. When firms adopt time-varying step-sizes, the dynamical system in Eq. (13) becomes non-stationary, i.e., $(\mathbf{p}^{t+1}, \mathbf{r}^{t+1}) = \mathbf{f}^t(\mathbf{p}^t, \mathbf{r}^t)$. Although the sequence of functions $\mathbf{f}^t{}_{t \geq 0}$ shares the same fixed point, verifying the convergence (stability) of the system requires examining the spectral radius of $\nabla \mathbf{f}^t(\mathbf{p}^{\star\star}, \mathbf{p}^{\star\star})$ for all $t \geq 0$.

Most importantly, even if asymptotic stability holds, it can only guarantee local convergence of Algorithm 1. Our goal, however, is to prove global convergence, such that both the price and reference price converge to the SNE for arbitrary initializations.

## Appendix B  More details about SNE

### B.1  Connection between equilibrium pricing policy and SNE

First, we further elaborate on the connection between equilibrium pricing policy and SNE. The work [22] investigates the properties of equilibrium pricing policy under the full information setting. They demonstrate that the policy $\mathbf{p}^{\star}(\mathbf{r})$ is a well-defined function since it outputs a unique equilibrium price for every given reference price $\mathbf{r}$. Further, for any given reference price $\mathbf{r}$, the equilibrium price $\left(p_H^{\star}(\mathbf{r}), p_L^{\star}(\mathbf{r})\right)$ is the unique solution to the following system of first-order equations [22, Eq. (B.10)]:

$$\begin{cases} p_H^{\star}(\mathbf{r}) = \dfrac{1}{(b_H + c_H) \cdot \left(1 - d_H(\mathbf{p}^{\star}(\mathbf{r}), \mathbf{r})\right)}, \\ p_L^{\star}(\mathbf{r}) = \dfrac{1}{(b_L + c_L) \cdot \left(1 - d_L(\mathbf{p}^{\star}(\mathbf{r}), \mathbf{r})\right)}. \end{cases} \tag{15}$$

The intertemporal aspect of the memory-based reference effect prompts researchers to examine the long-term consequence of repeatedly applying the equilibrium pricing policy in conjunction with reference price updates. Theorem 4 in [22] confirms that in the complete information setting, the price paths and reference price paths generated by the equilibrium pricing policy will converge to the SNE in the long run. Moreover, based on Eq. (15) for the equilibrium pricing policy, the SNE $\mathbf{p}^{\star\star}$ is

the unique solution of the following system of equations:

$$\begin{cases} p_H^{\star\star} = \dfrac{1}{(b_H + c_H) \cdot \left(1 - d_H(\mathbf{p}^{\star\star}, \mathbf{p}^{\star\star})\right)} = \dfrac{1 + \exp(a_H - b_H \cdot p_H^{\star\star}) + \exp(a_L - b_L \cdot p_L^{\star\star})}{(b_H + c_H) \cdot (1 + \exp(a_L - b_L \cdot p_L^{\star\star}))}, \\[4mm] p_L^{\star\star} = \dfrac{1}{(b_L + c_L) \cdot \left(1 - d_L(\mathbf{p}^{\star\star}, \mathbf{p}^{\star\star})\right)} = \dfrac{1 + \exp(a_H - b_H \cdot p_H^{\star\star}) + \exp(a_L - b_L \cdot p_L^{\star\star})}{(b_L + c_L) \cdot (1 + \exp(a_H - b_H \cdot p_H^{\star\star}))}. \end{cases} \tag{16}$$

It can be seen from Eq. (16) that the value of SNE depends only on the model parameters.

## B.2 Proof of Proposition 3.1

*Proof.* Firstly, the uniqueness of SNE has been established in the discussion in Appendix B.1. Below, we show the boundedness of the SNE. By performing a transformation on Eq. (16), we obtain the following inequality for $\mathbf{p}^{\star\star}$:

$$\frac{\exp(a_i - b_i \cdot p_i^{\star\star})}{(b_i + c_i)p_i^{\star\star} - 1} = 1 + \exp(a_{-i} - b_{-i} \cdot p_{-i}^{\star\star}) > 1, \quad \forall i \in \{H, L\}. \tag{17}$$

To make the above inequality valid, it immediately follows that

$$\frac{1}{b_i + c_i} < p_i^{\star\star} < \frac{1 + \exp(a_i - b_i \cdot p_i^{\star\star})}{b_i + c_i}, \quad \forall i \in \{H, L\}. \tag{18}$$

Now, we derive the upper bound for $p_i^{\star\star}$ from the second inequality in Eq. (18). Since the quantity on the right-hand side of Eq. (18) is monotone decreasing in $p_i^{\star\star}$, the value of $p_i^{\star\star}$ must be upper-bounded by the unique solution to the following equation with respect to $p_i$:

$$p_i = \frac{1 + \exp(a_i - b_i \cdot p_i)}{b_i + c_i}. \tag{19}$$

Define $x := -b_i/(b_i + c_i) + b_i \cdot p_i$. Then, one can easily verify that the above equation can be converted into

$$x \exp(x) = \frac{b_i}{b_i + c_i} \exp\left(a_i - \frac{b_i}{b_i + c_i}\right), \tag{20}$$

which implies that

$$x = W\left(\frac{b_i}{b_i + c_i} \exp\left(a_i - \frac{b_i}{b_i + c_i}\right)\right), \tag{21}$$

where $W(\cdot)$ is known as the Lambert $W$ function [65]. For any value $y \geq 0$, $W(y)$ is defined as the unique real solution to the equation

$$W(y) \cdot \exp\left(W(y)\right) = y. \tag{22}$$

Hence, we have

$$p_i^{\star\star} < p_i = \frac{1}{b_i + c_i} + \frac{1}{b_i} W\left(\frac{b_i}{b_i + c_i} \exp\left(a_i - \frac{b_i}{b_i + c_i}\right)\right). \tag{23}$$

Together with the lower bound provided in Eq. (18), this completest the proof. $\qquad\square$

# Appendix C  Proof of Theorem 5.1

*Proof.* In this section, we present the proof of Theorem 5.1. By Lemma F.1, when the step-sizes in Algorithm 1 are non-increasing and $\lim_{t\to\infty} \eta^t = 0$, the difference between reference price and price converges to zero as $t$ goes to infinity, i.e., $\lim_{t\to\infty} \left(\mathbf{r}^t - \mathbf{p}^t\right) = 0$. Thus, we are left to verify that the price path $\{\mathbf{p}^t\}_{t\geq 0}$ converges to the unique SNE, denoted by $\mathbf{p}^{\star\star}$ (see Definition 3.1). Recall that the uniqueness of the SNE is established in Proposition 3.1.

Consider the following weighted $\ell_1$-distance between a point $\mathbf{p}$ and the SNE $\mathbf{p}^{\star\star}$:

$$\varepsilon(\mathbf{p}) := \frac{|p_H^{\star\star} - p_H|}{b_H + c_H} + \frac{|p_L^{\star\star} - p_L|}{b_L + c_L}. \tag{24}$$

In a nutshell, our proof contains the following two parts:

- In **Part 1** (Appendix C.1), we show that the price path $\{\mathbf{p}^t\}_{t\geq 0}$ would enter the $\ell_1$-neighborhood $N^1_{\epsilon_0} := \{\mathbf{p} \in \mathcal{P}^2 \mid \varepsilon(\mathbf{p}) < \epsilon_0\}$ infinitely many times for any $\epsilon_0 > 0$.

- In **Part 2** (Appendix C.2), we show that when the price path $\{\mathbf{p}^t\}_{t\geq 0}$ enters the $\ell_2$-neighborhood $N^2_{\epsilon_0} := \{\mathbf{p} \in \mathcal{P}^2 \mid \|\mathbf{p} - \mathbf{p}^{\star\star}\|_2 < \epsilon_0\}$ for some sufficiently small $\epsilon_0 > 0$ with small enough step-sizes, the price path will stay in $N^2_{\epsilon_0}$ during subsequent periods.

Due to the diminishing step-sizes and the equivalence between $\ell_1$ and $\ell_2$ norms, **Part 1** guarantees that the price path would enter any $\ell_2$-neighborhood $N^2_{\epsilon_0}$ of $\mathbf{p}^{\star\star}$ with arbitrarily small step-sizes. Together with **Part 2**, this proves that for any sufficiently small $\epsilon_0 > 0$, there exists some $T_{\epsilon_0} > 0$ such that $\|\mathbf{p}^t - \mathbf{p}^{\star\star}\|_2 \leq \epsilon_0$ for every $t \geq T_{\epsilon_0}$, which implies the convergence to the SNE.

### C.1 Proof of Part 1

We argue by contradiction. Suppose there eixsts $\epsilon_0 > 0$ such that $\{\mathbf{p}\}_{t\geq 0}$ only visits $N^1_{\epsilon_0}$ finitely many times. This is equivalent to say that $\exists T^{\epsilon_0}$ such that $\{\mathbf{p}^t\}_{t\geq 0}$ never visits $N^1_{\epsilon_0}$ if $t \geq T^{\epsilon_0}$.

Firstly, let $G_i(\mathbf{p}, \mathbf{r})$ be the scaled partial derivative of the log-revenue, defined as

$$G_i(\mathbf{p}, \mathbf{r}) := \frac{1}{b_i + c_i} \cdot \frac{\partial \log\left(\Pi_i(\mathbf{p}, \mathbf{r})\right)}{\partial p_i} = \frac{1}{(b_i + c_i)p_i} + d_i(\mathbf{p}, \mathbf{r}) - 1, \quad \forall i \in \{H, L\}. \quad (25)$$

For the ease of notation, we denote $\mathcal{P}_i := \{p/(b_i + c_i) \mid p \in \mathcal{P}\}$ as the scaled price range. Then, the price update in Line 5 of Algorithm 1 is equivalent to

$$\frac{p_i^{t+1}}{b_i + c_i} = \mathrm{Proj}_{\mathcal{P}_i}\left(\frac{p_i^t}{b_i + c_i} + \eta^t \frac{D_i^t}{b_i + c_i}\right) = \mathrm{Proj}_{\mathcal{P}_i}\left(\frac{p_i^t}{b_i + c_i} + \eta^t G_i(\mathbf{p}^t, \mathbf{r}^t)\right). \quad (26)$$

Let $\mathrm{sign}(\cdot)$ be the sign function defined as

$$\mathrm{sign}(x) := \begin{cases} 1, & \text{if } x > 0, \\ 0, & \text{if } x = 0, \\ -1, & \text{if } x < 0. \end{cases} \quad (27)$$

Then, an essential observation from Eq. (26) is that: if $\mathrm{sign}(p_i^{\star\star} - p_i^t) \cdot G_i(\mathbf{p}^t, \mathbf{r}^t) > 0$, we have that $\mathrm{sign}(p_i^{t+1} - p_i^t) = \mathrm{sign}\left(G_i(\mathbf{p}^t, \mathbf{r}^t)\right) = \mathrm{sign}(p_i^{\star\star} - p_i^t)$, i.e., the update from $p_i^t$ to $p_i^{t+1}$ is toward the direction of the SNE price $p_i^{\star\star}$. Conversely, if $\mathrm{sign}(p_i^{\star\star} - p_i^t) \cdot G_i(\mathbf{p}^t, \mathbf{r}^t) < 0$, the update from $p_i^t$ to $p_i^{t+1}$ is deviating from $p_i^{\star\star}$.

We separate the whole feasible price range into four quadrants with $\mathbf{p}^{\star\star}$ being the origin:

$$N_1 := \{\mathbf{p} \in \mathcal{P}^2 \mid p_H > p_H^{\star\star} \text{ and } p_L \geq p_L^{\star\star}\}; \ N_2 := \{\mathbf{p} \in \mathcal{P}^2 \mid p_H \leq p_H^{\star\star} \text{ and } p_L > p_L^{\star\star}\};$$
$$N_3 := \{\mathbf{p} \in \mathcal{P}^2 \mid p_H < p_H^{\star\star} \text{ and } p_L \leq p_L^{\star\star}\}; \ N_4 := \{\mathbf{p} \in \mathcal{P}^2 \mid p_H \geq p_H^{\star\star} \text{ and } p_L < p_L^{\star\star}\}. \quad (28)$$

Below, we first show that *when $t$ is sufficiently large, prices in consecutive periods cannot always stay within a same region.* We prove it by a contradiction argument. Suppose that there exists some period $T_1^{\epsilon_0} > T^{\epsilon_0}$, after which the price path $\{\mathbf{p}^t\}_{t\geq T_1^{\epsilon_0}}$ stays within a same region, i.e.,

$\operatorname{sign}\left(p_i^{\star\star} - p_i^{t_1}\right) = \operatorname{sign}\left(p_i^{\star\star} - p_i^{t_2}\right), \forall t_1, t_2 \geq T_1^{\epsilon_0}, \forall i \in \{H, L\}$. Then, for $t \geq T_1^{\epsilon_0}$, it holds that

$$\varepsilon(\mathbf{p}^{t+1}) = \frac{|p_H^{\star\star} - p_H^{t+1}|}{b_H + c_H} + \frac{|p_L^{\star\star} - p_L^{t+1}|}{b_L + c_L}$$

$$= \left| \frac{p_H^{\star\star}}{b_H + c_H} - \operatorname{Proj}_{\mathcal{P}_H}\left( \eta^t G_H(\mathbf{p}^t, \mathbf{r}^t) + \frac{p_H^t}{b_H + c_H} \right) \right|$$
$$+ \left| \frac{p_L^{\star\star}}{b_L + c_L} - \operatorname{Proj}_{\mathcal{P}_L}\left( \eta^t G_L(\mathbf{p}^t, \mathbf{r}^t) + \frac{p_L^t}{b_L + c_L} \right) \right|$$

$$\overset{(\Delta_1)}{\leq} \left| \frac{p_H^{\star\star}}{b_H + c_H} - \eta^t G_H(\mathbf{p}^t, \mathbf{r}^t) - \frac{p_H^t}{b_H + c_H} \right| + \left| \frac{p_L^{\star\star}}{b_L + c_L} - \eta^t G_L(\mathbf{p}^t, \mathbf{r}^t) - \frac{p_L^t}{b_L + c_L} \right|$$

$$\overset{(\Delta_2)}{=} \operatorname{sign}\left(p_H^{\star\star} - p_H^t\right) \cdot \left( \frac{p_H^{\star\star}}{b_H + c_H} - \eta^t G_H(\mathbf{p}^t, \mathbf{r}^t) - \frac{p_H^t}{b_H + c_H} \right)$$
$$+ \operatorname{sign}\left(p_L^{\star\star} - p_L^t\right) \cdot \left( \frac{p_L^{\star\star}}{b_L + c_L} - \eta^t G_L(\mathbf{p}^t, \mathbf{r}^t) - \frac{p_L^t}{b_L + c_L} \right)$$

$$= \operatorname{sign}\left(p_H^{\star\star} - p_H^t\right) \cdot \left( \frac{p_H^{\star\star}}{b_H + c_H} - \eta^t G_H(\mathbf{p}^t, \mathbf{p}^t) - \frac{p_H^t}{b_H + c_H} \right)$$
$$+ \operatorname{sign}\left(p_L^{\star\star} - p_L^t\right) \cdot \left( \frac{p_L^{\star\star}}{b_L + c_L} - \eta^t G_L(\mathbf{p}^t, \mathbf{p}^t) - \frac{p_L^t}{b_L + c_L} \right)$$
$$+ \eta^t \left( \left| G_H(\mathbf{p}^t, \mathbf{r}^t) - G_H(\mathbf{p}^t, \mathbf{p}^t) \right| + \left| G_L(\mathbf{p}^t, \mathbf{r}^t) - G_L(\mathbf{p}^t, \mathbf{p}^t) \right| \right)$$

$$\overset{(\Delta_3)}{\leq} \operatorname{sign}\left(p_H^{\star\star} - p_H^t\right) \cdot \left( \frac{p_H^{\star\star} - p_H^t}{b_H + c_H} \right) + \operatorname{sign}\left(p_L^{\star\star} - p_L^t\right) \cdot \left( \frac{p_L^{\star\star} - p_L^t}{b_L + c_L} \right)$$
$$- \eta^t \left( \underbrace{\operatorname{sign}\left(p_H^{\star\star} - p_H^t\right) \cdot G_H(\mathbf{p}^t, \mathbf{p}^t) + \operatorname{sign}\left(p_L^{\star\star} - p_L^t\right) \cdot G_L(\mathbf{p}^t, \mathbf{p}^t)}_{\mathcal{G}(\mathbf{p}^t)} \right)$$
$$+ 2\eta^t \cdot \ell_r \|\mathbf{r}^t - \mathbf{p}^t\|_2$$

$$\overset{(\Delta_4)}{=} \frac{|p_H^{\star\star} - p_H^t|}{b_H + c_H} + \frac{|p_L^{\star\star} - p_L^t|}{b_L + c_L} - \eta^t \mathcal{G}(\mathbf{p}^t) + 2\eta^t \cdot \ell_r \|\mathbf{r}^t - \mathbf{p}^t\|_2$$
$$= \varepsilon(\mathbf{p}^t) - \eta^t \left( \mathcal{G}(\mathbf{p}^t) - 2\ell_r \|\mathbf{r}^t - \mathbf{p}^t\|_2 \right)$$
$$\overset{(\Delta_5)}{\leq} \varepsilon(\mathbf{p}^t) - \eta^t \left( M_{\epsilon_0} - 2\ell_r \|\mathbf{r}^t - \mathbf{p}^t\|_2 \right), \quad \forall t \geq T_1^{\epsilon_0},$$

$$(29)$$

where we use the property of the projection operator in $(\Delta_1)$. The equality $(\Delta_2)$ follows since $\operatorname{sign}\left(p_i^{\star\star} - p_i^{t+1}\right) = \operatorname{sign}\left(p_i^{\star\star} - p_i^t\right)$ for $i \in \{H, L\}$, and the projection does not change the sign, then we have

$$\operatorname{sign}\left( \frac{p_i^{\star\star}}{b_i + c_i} - \eta^t G_i(\mathbf{p}^t, \mathbf{r}^t) - \frac{p_i^t}{b_i + c_i} \right) = \operatorname{sign}\left(p_i^{\star\star} - p_i^t\right), \quad \forall i \in \{H, L\}.$$

The inequality $(\Delta_3)$ results from Lemma F.5. In particular, since $\left\| \nabla_{\mathbf{r}} G_i(\mathbf{p}, \mathbf{r}) \right\|_2 \leq \ell_r$, where $\ell_r = (1/4)\sqrt{c_H^2 + c_L^2}$, it follows that

$$\left| G_i(\mathbf{p}^t, \mathbf{r}^t) - G_i(\mathbf{p}^t, \mathbf{p}^t) \right| \leq \left\| \nabla_{\mathbf{r}} G_i(\mathbf{p}, \mathbf{r}) \right\|_2 \cdot \left\| \mathbf{r}^t - \mathbf{p}^t \right\|_2 \leq \ell_r \left\| \mathbf{r}^t - \mathbf{p}^t \right\|_2, \quad \forall i \in \{H, L\}. \quad (30)$$

In step $(\Delta_4)$, we introduce the function $\mathcal{G}(\mathbf{p})$, which is defined as

$$\mathcal{G}(\mathbf{p}) := \operatorname{sign}(p_H^{\star\star} - p_H) \cdot G_H(\mathbf{p}, \mathbf{p}) + \operatorname{sign}(p_L^{\star\star} - p_L) \cdot G_L(\mathbf{p}, \mathbf{p}), \quad \forall \mathbf{p} \in \mathcal{P}^2. \quad (31)$$

Finally, the last step $(\Delta_5)$ is due to Lemma F.2, which states that $\exists M_{\epsilon_0} > 0$ such that $\mathcal{G}(\mathbf{p}) \geq M_{\epsilon_0}$ for every $\mathbf{p} \in \mathcal{P}^2$ with $\varepsilon(\mathbf{p}) > \epsilon_0$. Since for $t \geq T_1^{\epsilon_0} > T^{\epsilon_0}$, we have $\varepsilon(\mathbf{p}^t) \geq \epsilon_0$ by the premise, it follows that $\mathcal{G}(\mathbf{p}^t) \geq M_{\epsilon_0}$.

By Lemma F.1, the difference $\{\mathbf{r}^t - \mathbf{p}^t\}_{t \geq 0}$ converges to 0 as $t$ approaches to infinity . As a result, we can always find $T_2^{\epsilon_0} > 0$ to ensure $\|\mathbf{r}^t - \mathbf{p}^t\|_2 \leq M_{\epsilon_0}/(4\ell_r)$ for all $t \geq T_2^{\epsilon_0}$. Then, the last line in Eq. (29) can be further upper-bounded as

$$
\begin{aligned}
\varepsilon(\mathbf{p}^{t+1}) &\leq \varepsilon(\mathbf{p}^t) - \eta^t \left( M_{\epsilon_0} - 2\ell_r \|\mathbf{r}^t - \mathbf{p}^t\|_2 \right) \\
&\leq \varepsilon(\mathbf{p}^t) - \eta^t \left( M_{\epsilon_0} - \frac{1}{2} M_{\epsilon_0} \right) \\
&\leq \varepsilon(\mathbf{p}^t) - \frac{1}{2} \eta^t M_{\epsilon_0}, \quad \forall t \geq \widehat{T}^{\epsilon_0} := \max\{T_1^{\epsilon_0}, T_2^{\epsilon_0}\}.
\end{aligned}
\tag{32}
$$

By a telescoping sum from any period $T - 1 \geq \widehat{T}^{\epsilon_0}$ down to period $\widehat{T}^{\epsilon_0}$, we have that

$$
\varepsilon(\mathbf{p}^T) \leq \varepsilon(\mathbf{p}^{\widehat{T}^{\epsilon_0}}) - \frac{M_{\epsilon_0}}{2} \sum_{t=\widehat{T}^{\epsilon_0}}^{T-1} \eta^t.
\tag{33}
$$

Since the step-sizes satisfy $\sum_{t=0}^{\infty} \eta^t = \infty$, we have that $\lim_{T \to \infty} (M_{\epsilon_0}/2) \sum_{t=\widehat{T}^{\epsilon_0}}^{T-1} \eta^t = \infty$, which further implies the contradiction that $\lim_{T \to \infty} \varepsilon(\mathbf{p}^T) \to -\infty$. Thus, *the prices in consecutive period will not always stay in the same region, i.e., the price path $\{\mathbf{p}^t\}_{t \geq 0}$ keeps oscillating between different regions $N_j$ for $j \in \{1, 2, 3, 4\}$.*

Next, we prove that $\exists T_3^{\epsilon_0} \geq T^{\epsilon_0}$ such that the price path $\{\mathbf{p}^t\}_{t \geq T_3^{\epsilon_0}}$ *only oscillates between adjacent quadrants.* Arguing by contradiction, we assume $\eta^{T_3^{\epsilon_0}} < \epsilon_0/(2M_G)$, where $M_G$ is the upper bound of $|G_i(\mathbf{p}, \mathbf{r})|$ defined in Eq. (100) in Lemma F.5. If $\mathbf{p}^t \in N_1$ (or $N_2$) and $\mathbf{p}^{t+1} \in N_3$ (or $N_4$), i.e., $\mathrm{sign}\left(p_i^{\star\star} - p_i^t\right) \neq \mathrm{sign}\left(p_i^{\star\star} - p_i^{t+1}\right)$ for both $i \in \{H, L\}$, then it automatically follows from the update rule (see Eq. (26)) that

$$
\begin{aligned}
\varepsilon(\mathbf{p}^{t+1}) &\leq \eta^t \cdot \left( \left| G_H(\mathbf{p}^t, \mathbf{r}^t) \right| + \left| G_L(\mathbf{p}^t, \mathbf{r}^t) \right| \right) \\
&\leq \frac{\epsilon_0}{2M_G} \cdot 2M_G = \epsilon_0,
\end{aligned}
\tag{34}
$$

which again contradicts the premise that $\{\mathbf{p}^t\}_{t \geq T^{\epsilon_0}}$ never visits $N_{\epsilon_0}^1$. Thus, *when $t \geq T_3^{\epsilon_0}$, the price path $\{\mathbf{p}^t\}_{t \geq T_3^{\epsilon_0}}$ only oscillates between adjacent quadrants.*

Because of the following two established facts: $(i)$ when $t \geq \max\{T_2^{\epsilon_0}, T_3^{\epsilon_0}\}$, the price path will not remain in the same quadrant but can only oscillate between adjacent quadrants, and $(ii)$ the step-sizes $\{\eta^t\}_{t \geq 0}$ decrease to 0 as $t \to \infty$, we conclude that the price path $\{\mathbf{p}\}_{t \geq 0}$ must visit the set

$$
\widehat{N}_{\epsilon_1} := \left\{ \mathbf{p} \in \mathcal{P}^2 \,\middle|\, \frac{|p_H^{\star\star} - p_H|}{b_H + c_H} < \epsilon_1 \text{ or } \frac{|p_L^{\star\star} - p_L|}{b_L + c_L} < \epsilon_1 \right\}
\tag{35}
$$

infinitely many times for any $\epsilon_1 > 0$. Intuitively, set $\widehat{N}_{\epsilon_1}$ can be understood as the boundary regions between adjacent quadrants. We define $\mathcal{T}_{\epsilon_1}$ as the set of "jumping periods" from $(N_1 \cup N_3) \cap \widehat{N}_{\epsilon_1}$ to $(N_2 \cup N_4) \cap \widehat{N}_{\epsilon_1}$ after $T^{\epsilon_0}$, i.e.,

$$
\mathcal{T}_{\epsilon_1} := \left\{ t \geq T^{\epsilon_0} \mid \mathbf{p}^t \in (N_1 \cup N_3) \cap \widehat{N}_{\epsilon_1} \text{ and } \mathbf{p}^{t+1} \in (N_2 \cup N_4) \cap \widehat{N}_{\epsilon_1} \right\}.
\tag{36}
$$

Then, the above fact $(i)$ implies that $|\mathcal{T}_{\epsilon_1}| = \infty$ for any $\epsilon_1 > 0$.

The key idea of the following proof is that *if $\{\mathbf{p}^t\}_{t \geq 0}$ visits $\widehat{N}_{\epsilon_1}$ with some sufficiently small $\epsilon_1$ and a relatively smaller step-size, it will stay in $\widehat{N}_{\epsilon_1}$ and converge to $N_{\epsilon_0}^1$ concurrently.* This results in a contradiction with our initial assumption that $\{\mathbf{p}^t\}_{t \geq 0}$ never visits $N_{\epsilon_0}^1$ when $t \geq T^{\epsilon_0}$.

Let $T_1^{\epsilon_1} \in \mathcal{T}_{\epsilon_1}$ be a jumping period with $\epsilon_1 \ll \epsilon_0$. Without loss of generality, suppose that $\left| p_H^{\star\star} - p_H^{T_1^{\epsilon_1}} \right|/(b_H + c_H) < \epsilon_1$. Then, since $\mathbf{p}^{T_1^{\epsilon_1}} \notin N_{\epsilon_0}^1$, we accordingly have that $\left| p_L^{\star\star} - p_L^{T_1^{\epsilon_1}} \right|/(b_L + c_L) >$

$\epsilon_0 - \epsilon_1$. By the update rule of price in Eq. (26), it holds that

$$
\frac{\left|p_L^{\star\star} - p_L^{T_1^{\epsilon_1}+1}\right|}{b_L + c_L} = \text{sign}\left(p_L^{\star\star} - p_L^{T_1^{\epsilon_1}+1}\right) \cdot \frac{p_L^{\star\star} - p_L^{T_1^{\epsilon_1}+1}}{b_L + c_L}
$$

$$
= \text{sign}\left(p_L^{\star\star} - p_L^{T_1^{\epsilon_1}+1}\right) \cdot \left(\frac{p_L^{\star\star}}{b_L + c_L} - \text{Proj}_{\mathcal{P}_L}\left(\frac{p_L^{T_1^{\epsilon_1}}}{b_L + c_L} + \eta^{T_1^{\epsilon_1}} G_L\left(\mathbf{p}^{T_1^{\epsilon_1}}, \mathbf{r}^{T_1^{\epsilon_1}}\right)\right)\right)
$$

$$
\leq \text{sign}\left(p_L^{\star\star} - p_L^{T_1^{\epsilon_1}+1}\right) \cdot \left(\frac{p_L^{\star\star} - p_L^{T_1^{\epsilon_1}}}{b_L + c_L} - \eta^{T_1^{\epsilon_1}} G_L\left(\mathbf{p}^{T_1^{\epsilon_1}}, \mathbf{r}^{T_1^{\epsilon_1}}\right)\right)
$$

$$
= \text{sign}\left(p_L^{\star\star} - p_L^{T_1^{\epsilon_1}}\right) \cdot \frac{p_L^{\star\star} - p_L^{T_1^{\epsilon_1}}}{b_L + c_L} - \text{sign}\left(p_L^{\star\star} - p_L^{T_1^{\epsilon_1}}\right) \cdot \left(\eta^{T_1^{\epsilon_1}} G_L\left(\mathbf{p}^{T_1^{\epsilon_1}}, \mathbf{r}^{T_1^{\epsilon_1}}\right)\right).
$$
(37)

We note that, the last step above is due to the premise that the step-size is sufficiently small: since $\left|p_L^{\star\star} - p_L^{T_1^{\epsilon_1}}\right|/(b_L + c_L) > \epsilon_0 - \epsilon_1$, the equality $\text{sign}\left(p_L^{\star\star} - p_L^{T_1^{\epsilon_1}+1}\right) = \text{sign}\left(p_L^{\star\star} - p_L^{T_1^{\epsilon_1}}\right)$ holds if $\eta^{T_1^{\epsilon_1}} < (\epsilon_0 - \epsilon_1)/M_G$. Next, we show that the second term on the right-hand side of Eq. (37) is upper-bounded away from zero:

$$
\text{sign}\left(p_L^{\star\star} - p_L^{T_1^{\epsilon_1}}\right) \cdot G_L\left(\mathbf{p}^{T_1^{\epsilon_1}}, \mathbf{r}^{T_1^{\epsilon_1}}\right)
$$

$$
\geq \text{sign}\left(p_L^{\star\star} - p_L^{T_1^{\epsilon_1}}\right) \cdot G_L\left(\mathbf{p}^{T_1^{\epsilon_1}}, \mathbf{p}^{T_1^{\epsilon_1}}\right) - \left|G_L\left(\mathbf{p}^{T_1^{\epsilon_1}}, \mathbf{r}^{T_1^{\epsilon_1}}\right) - G_L\left(\mathbf{p}^{T_1^{\epsilon_1}}, \mathbf{p}^{T_1^{\epsilon_1}}\right)\right|
$$

$$
\overset{(\Delta_1)}{\geq} \text{sign}\left(p_L^{\star\star} - p_L^{T_1^{\epsilon_1}}\right) \cdot G_L\left(\mathbf{p}^{T_1^{\epsilon_1}}, \mathbf{p}^{T_1^{\epsilon_1}}\right) - \ell_r \left\|\mathbf{r}^{T_1^{\epsilon_1}} - \mathbf{p}^{T_1^{\epsilon_1}}\right\|_2
$$

$$
= \text{sign}\left(p_L^{\star\star} - p_L^{T_1^{\epsilon_1}}\right) \cdot \left[\frac{1}{(b_L + c_L)p_L^{T_1^{\epsilon_1}}} + d_L(\mathbf{p}^{T_1^{\epsilon_1}}, \mathbf{p}^{T_1^{\epsilon_1}}) - 1\right] - \ell_r \left\|\mathbf{r}^{T_1^{\epsilon_1}} - \mathbf{p}^{T_1^{\epsilon_1}}\right\|_2
$$

$$
\geq \text{sign}\left(p_L^{\star\star} - p_L^{T_1^{\epsilon_1}}\right) \cdot \left[\frac{1}{(b_L + c_L)p_L^{T_1^{\epsilon_1}}} + d_L\left((p_H^{\star\star}, p_L^{T_1^{\epsilon_1}}), (p_H^{\star\star}, p_L^{T_1^{\epsilon_1}})\right) - 1\right]
$$
$$
- \left|d_L(\mathbf{p}^{T_1^{\epsilon_1}}, \mathbf{p}^{T_1^{\epsilon_1}}) - d_L\left((p_H^{\star\star}, p_L^{T_1^{\epsilon_1}}), (p_H^{\star\star}, p_L^{T_1^{\epsilon_1}})\right)\right| - \ell_r \left\|\mathbf{r}^{T_1^{\epsilon_1}} - \mathbf{p}^{T_1^{\epsilon_1}}\right\|_2
$$

$$
\overset{(\Delta_2)}{\geq} \text{sign} \cdot \left(p_L^{\star\star} - p_L^{T_1^{\epsilon_1}}\right) \cdot \left[\frac{1}{(b_L + c_L)p_L^{T_1^{\epsilon_1}}} + d_L\left((p_H^{\star\star}, p_L^{T_1^{\epsilon_1}}), (p_H^{\star\star}, p_L^{T_1^{\epsilon_1}})\right) - 1\right] - \ell_r \left\|\mathbf{r}^{T_1^{\epsilon_1}} - \mathbf{p}^{T_1^{\epsilon_1}}\right\|_2
$$
$$
- \max_{\mathbf{p} \in \mathcal{P}}\left\{\left|\frac{\partial d_L(\mathbf{p}, \mathbf{p})}{\partial p_H}\right|\right\} \cdot \left|p_H^{\star\star} - p_H^{T_1^{\epsilon_1}}\right|
$$

$$
\overset{(\Delta_3)}{\geq} \text{sign}\left(p_L^{\star\star} - p_L^{T_1^{\epsilon_1}}\right) \cdot \left[\frac{1}{(b_L + c_L)p_L^{T_1^{\epsilon_1}}} + d_L\left((p_H^{\star\star}, p_L^{T_1^{\epsilon_1}}), (p_H^{\star\star}, p_L^{T_1^{\epsilon_1}})\right) - 1\right] - \ell_r \left\|\mathbf{r}^{T_1^{\epsilon_1}} - \mathbf{p}^{T_1^{\epsilon_1}}\right\|_2
$$
$$
- \frac{1}{4}b_H(b_H + c_H)\epsilon_1
$$

$$
= \mathcal{G}\left((p_H^{\star\star}, p_L^{T_1^{\epsilon_1}})\right) - \ell_r \left\|\mathbf{r}^{T_1^{\epsilon_1}} - \mathbf{p}^{T_1^{\epsilon_1}}\right\|_2 - \frac{1}{4}b_H(b_H + c_H)\epsilon_1,
$$
(38)

where $(\Delta_1)$ holds because of the mean value theorem and the fact that $\|\nabla_{\mathbf{r}} G_L(\mathbf{p}, \mathbf{r})\|_2 \leq \ell_r$ for $\mathbf{p}, \mathbf{r} \in \mathcal{P}^2$ by Lemma F.5, inequality $(\Delta_2)$ applies the mean value theorem again to the demand function. In $(\Delta_3)$, we use the assumption that $\left|p_H^{\star\star} - p_H^{T_1^{\epsilon_1}}\right| < (b_H + c_H)\epsilon_1$ and apply a similar

argument as Eq. (101) to derive that

$$\left|\frac{\partial d_L(\mathbf{p},\mathbf{p})}{\partial p_H}\right| = |b_H \cdot d_L(\mathbf{p},\mathbf{p}) \cdot d_H(\mathbf{p},\mathbf{p})| \leq \frac{1}{4}b_H, \quad \forall \mathbf{p} \in \mathcal{P}^2. \tag{39}$$

Since $\varepsilon\big((p_H^{\star\star}, p_L^{T_1^{\epsilon_1}})\big) > \epsilon_0 - \epsilon_1$, by Lemma F.2, there exists a constant $M_{\epsilon_0 - \epsilon_1} > 0$, such that $\mathcal{G}\big((p_H^{\star\star}, p_L^{T_1^{\epsilon_1}})\big) \geq M_{\epsilon_0 - \epsilon_1}$. Meanwhile, we can choose $\epsilon_1$ sufficiently smaller than $\epsilon_0$ to ensure that $(1/2)M_{\epsilon_0 - \epsilon_1} - (1/4)b_H(b_H + c_H)\epsilon_1 > 0$. Furthermore, recall that $T_1^{\epsilon_1}$ can be arbitrarily chosen from $\mathcal{T}_{\epsilon_1}$, where $|\mathcal{T}_{\epsilon_1}| = \infty$. Thus, we can always find a sufficiently large $T_1^{\epsilon_1} \in \mathcal{T}_{\epsilon_1}$ to guarantee that $\ell_r \|\mathbf{r}^{T_1^{\epsilon_1}} - \mathbf{p}^{T_1^{\epsilon_1}}\|_2 \leq (1/2)M_{\epsilon_0 - \epsilon_1} - (1/4)b_H(b_H + c_H)\epsilon_1$ by Lemma F.1. Back to Eq. (38), it follows that

$$\begin{aligned}
&\text{sign}\big(p_L^{\star\star} - p_L^{T_1^{\epsilon_1}}\big) \cdot G_L\big(\mathbf{p}^{T_1^{\epsilon_1}}, \mathbf{r}^{T_1^{\epsilon_1}}\big) \\
&\geq \mathcal{G}\big((p_H^{\star\star}, p_L^{T_1^{\epsilon_1}})\big) - \ell_r\|\mathbf{r}^{T_1^{\epsilon_1}} - \mathbf{p}^{T_1^{\epsilon_1}}\|_2 - \frac{1}{4}b_H(b_H + c_H)\epsilon_1 \\
&\geq M_{\epsilon_0 - \epsilon_1} - \left(\frac{1}{2}M_{\epsilon_0 - \epsilon_1} - \frac{1}{4}b_H(b_H + c_H)\epsilon_1\right) - \frac{1}{4}b_H(b_H + c_H)\epsilon_1 \\
&= \frac{1}{2}M_{\epsilon_0 - \epsilon_1}.
\end{aligned} \tag{40}$$

By substituting Eq. (40) into Eq. (37), we further derive that

$$\begin{aligned}
\frac{\left|p_L^{\star\star} - p_L^{T_1^{\epsilon_1}+1}\right|}{b_L + c_L} &\leq \text{sign}\big(p_L^{\star\star} - p_L^{T_1^{\epsilon_1}}\big) \cdot \frac{p_L^{\star\star} - p_L^{T_1^{\epsilon_1}}}{b_L + c_L} - \text{sign}\big(p_L^{\star\star} - p_L^{T_1^{\epsilon_1}}\big) \cdot \left(\eta^{T_1^{\epsilon_1}} G_L\big(\mathbf{p}^{T_1^{\epsilon_1}}, \mathbf{r}^{T_1^{\epsilon_1}}\big)\right) \\
&\leq \frac{\left|p_L^{\star\star} - p_L^{T_1^{\epsilon_1}}\right|}{b_L + c_L} - \frac{1}{2}\eta^{T_1^{\epsilon_1}} M_{\epsilon_0 - \epsilon_1}.
\end{aligned} \tag{41}$$

This indicates that the price update from $p_L^{T_1^{\epsilon_1}}$ to $p_L^{T_1^{\epsilon_1}+1}$ is towards the SNE price $p_L^{\star\star}$ when $\left|p_H^{\star\star} - p_H^{T_1^{\epsilon_1}}\right| < (b_H + c_H)\epsilon_1$.

Now, we use a strong induction to show that $\left|p_H^{\star\star} - p_H^t\right|/(b_H + c_H) < \epsilon_1$ *holds for all* $t \geq T_1^{\epsilon_1}$ *provided that* $\epsilon_1$ *is sufficiently small and* $T_1^{\epsilon_1}$ *is sufficiently large.* Suppose there exists $T_2^{\epsilon_1} \geq T_1^{\epsilon_1}$ such that $\left|p_H^{\star\star} - p_H^t\right|/(b_H + c_H) < \epsilon_1$ holds for all $t = T_1^{\epsilon_1}, T_1^{\epsilon_1} + 1, \ldots, T_2^{\epsilon_1}$, we aim to show that $\left|p_H^{\star\star} - p_H^{T_2^{\epsilon_1}+1}\right|/(b_H + c_H) < \epsilon_1$. Following the same derivation from Eq. (37) to Eq. (41), it holds that

$$\frac{\left|p_L^{\star\star} - p_L^{t+1}\right|}{b_L + c_L} \leq \frac{\left|p_L^{\star\star} - p_L^t\right|}{b_L + c_L} - \frac{1}{2}\eta^t M_{\epsilon_0 - \epsilon_1}, \quad t = T_1^{\epsilon_1}, T_1^{\epsilon_1} + 1, \ldots, T_2^{\epsilon_1}. \tag{42}$$

By telescoping the above inequality from $T_2^{\epsilon_1} - 1$ down to $T_1^{\epsilon_1}$, we have that

$$\frac{\left|p_L^{\star\star} - p_L^{T_2^{\epsilon_1}}\right|}{b_L + c_L} \leq \frac{\left|p_L^{\star\star} - p_L^{T_1^{\epsilon_1}}\right|}{b_L + c_L} - \frac{M_{\epsilon_0 - \epsilon_1}}{2}\sum_{t=T_1^{\epsilon_1}}^{T_2^{\epsilon_1}-1} \eta^t \leq \frac{\left|p_L^{\star\star} - p_L^{T_1^{\epsilon_1}}\right|}{b_L + c_L}. \tag{43}$$

Additionally, we note that $\left|p_H^{\star\star} - p_H^t\right|/(b_H + c_H) < \epsilon_1$ and $\mathbf{p}^t \notin N_{\epsilon_0}^1$ together imply that $\left|p_L^{\star\star} - p_L^t\right|/(b_L + c_L) > (\epsilon_0 - \epsilon_1)$ for all $t = T_1^{\epsilon_1}, T_1^{\epsilon_1} + 1, \ldots, T_2^{\epsilon_1}$. As the step-size is sufficiently small, we conclude that the price path $\{p_L^t\}_{t=T_1^{\epsilon_1}}^{T_2^{\epsilon_1}}$ lies on the same side of the SNE price $p_L^{\star\star}$, and thus $\text{sign}\big(p_L^{\star\star} - p_L^{T_2^{\epsilon_1}}\big) = \text{sign}\big(p_L^{\star\star} - p_L^{T_1^{\epsilon_1}}\big)$. Below, we discuss two cases based on the value of $\left|p_H^{\star\star} - p_H^{T_2^{\epsilon_1}}\right|$:

**Case 1:** $\left|p_H^{\star\star} - p_H^{T_2^{\epsilon_1}}\right|/(b_H + c_H) < \epsilon_1 - \eta^{T_1^{\epsilon_1}} M_G$.

Following a similar derivation as Eq.(37), we have that

$$\frac{\left|p_H^{\star\star} - p_H^{T_2^{\epsilon_1}+1}\right|}{b_H + c_H} \leq \text{sign}\left(p_H^{\star\star} - p_H^{T_2^{\epsilon_1}}\right) \cdot \frac{p_H^{\star\star} - p_H^{T_2^{\epsilon_1}}}{b_H + c_H} - \eta^{T_2^{\epsilon_1}} \text{sign}\left(p_H^{\star\star} - p_H^{T_2^{\epsilon_1}}\right) \cdot G_H\left(\mathbf{p}^{T_2^{\epsilon_1}}, \mathbf{r}^{T_2^{\epsilon_1}}\right).$$

$$\leq \frac{\left|p_H^{\star\star} - p_H^{T_2^{\epsilon_1}}\right|}{b_H + c_H} + \eta^{T_2^{\epsilon_1}} \left|G_H\left(\mathbf{p}^{T_2^{\epsilon_1}}, \mathbf{r}^{T_2^{\epsilon_1}}\right)\right|$$

$$\overset{(\Delta_1)}{<} \left(\epsilon_1 - \eta^{T_1^{\epsilon_1}} M_G\right) + \eta^{T_2^{\epsilon_1}} M_G$$

$$\overset{(\Delta_2)}{\leq} \epsilon_1,$$

(44)

where $(\Delta_1)$ is from Lemma F.5 that $\left|G_H(\mathbf{p}, \mathbf{r})\right| \leq M_G$ for $\mathbf{p}, \mathbf{r} \in \mathcal{P}^2$, and $(\Delta_2)$ is because the sequence of step-sizes is non-increasing, hence $\eta^{T_1^{\epsilon_1}} \geq \eta^{T_2^{\epsilon_1}}$.

**Case 2:** $\left|p_H^{\star\star} - p_H^{T_2^{\epsilon_1}}\right|/(b_H + c_H) \in \left[\epsilon_1 - \eta^{T_1^{\epsilon_1}} M_G, \epsilon_1\right)$.

It suffices to show that $\left|p_H^{\star\star} - p_H^{T_2^{\epsilon_1}+1}\right| \leq \left|p_H^{\star\star} - p_H^{T_2^{\epsilon_1}}\right|$. According to the observation below Eq. (27), this is equivalent to showing that

$$\text{sign}\left(p_H^{\star\star} - p_H^{T_2^{\epsilon_1}}\right) \cdot G_H\left(\mathbf{p}^{T_2^{\epsilon_1}}, \mathbf{r}^{T_2^{\epsilon_1}}\right) \geq 0.$$

(45)

We further discuss two scenarios depending on whether $\mathbf{p}^{T_2^{\epsilon_1}} \in N_1 \cup N_3$ or $\mathbf{p}^{T_2^{\epsilon_1}} \in N_2 \cup N_4$.

1. Suppose that $\mathbf{p}^{T_2^{\epsilon_1}} \in N_1 \cup N_3$. According to the definition of set $\mathcal{T}_{\epsilon_1}$ in Eq. (36), we have that $\mathbf{p}^{T_1^{\epsilon_1}} \in N_1 \cup N_3$ since $T_1^{\epsilon_1} \in \mathcal{T}_{\epsilon_1}$ is a jumping period. Together with the established fact that $\text{sign}\left(p_L^{\star\star} - p_L^{T_2^{\epsilon_1}}\right) = \text{sign}\left(p_L^{\star\star} - p_L^{T_1^{\epsilon_1}}\right)$ (see the argument below Eq. (43)), we conclude that $\mathbf{p}^{T_2^{\epsilon_1}}$ and $\mathbf{p}^{T_1^{\epsilon_1}}$ are in the same quadrant, and $\text{sign}\left(p_H^{\star\star} - p_H^{T_2^{\epsilon_1}}\right) = \text{sign}\left(p_H^{\star\star} - p_H^{T_1^{\epsilon_1}}\right)$. Then, we can write that

$$\text{sign}\left(p_H^{\star\star} - p_H^{T_2^{\epsilon_1}}\right) \cdot G_H\left(\mathbf{p}^{T_2^{\epsilon_1}}, \mathbf{r}^{T_2^{\epsilon_1}}\right)$$

$$\geq \text{sign}\left(p_H^{\star\star} - p_H^{T_2^{\epsilon_1}}\right) \cdot G_H\left(\mathbf{p}^{T_2^{\epsilon_1}}, \mathbf{p}^{T_2^{\epsilon_1}}\right) - \ell_r \|\mathbf{r}^{T_2^{\epsilon_1}} - \mathbf{p}^{T_2^{\epsilon_1}}\|_2$$

$$\overset{(\Delta)}{\geq} \text{sign}\left(p_H^{\star\star} - p_H^{T_2^{\epsilon_1}}\right) \cdot G_H\left((p_H^{T_2^{\epsilon_1}}, p_L^{T_1^{\epsilon_1}}), (p_H^{T_2^{\epsilon_1}}, p_L^{T_1^{\epsilon_1}})\right) - \ell_r \|\mathbf{r}^{T_2^{\epsilon_1}} - \mathbf{p}^{T_2^{\epsilon_1}}\|_2$$

(46)

$$= \text{sign}\left(p_H^{\star\star} - p_H^{T_1^{\epsilon_1}}\right) \cdot G_H\left(\mathbf{p}^{T_1^{\epsilon_1}}, \mathbf{p}^{T_1^{\epsilon_1}}\right) - \ell_r \|\mathbf{r}^{T_2^{\epsilon_1}} - \mathbf{p}^{T_2^{\epsilon_1}}\|_2$$

$$+ \text{sign}\left(p_H^{\star\star} - p_H^{T_1^{\epsilon_1}}\right) \cdot \left(G_H\left((p_H^{T_2^{\epsilon_1}}, p_L^{T_1^{\epsilon_1}}), (p_H^{T_2^{\epsilon_1}}, p_L^{T_1^{\epsilon_1}})\right) - G_H\left(\mathbf{p}^{T_1^{\epsilon_1}}, \mathbf{p}^{T_1^{\epsilon_1}}\right)\right),$$

where we apply Lemma F.4 in $(\Delta)$ by utilizing the relation $\left|p_L^{\star\star} - p_L^{T_2^{\epsilon_1}}\right| \leq \left|p_L^{\star\star} - p_L^{T_1^{\epsilon_1}}\right|$ proved in Eq. (43). Next, as $T_1^{\epsilon_1}$ is a jumping period, it follows from the price update rule in Eq. (26) that $\left|p_H^{\star\star} - p_H^{T_1^{\epsilon_1}}\right|/(b_H + c_H) \leq \left|\eta^{T_1^{\epsilon_1}} G_H\left(\mathbf{p}^{T_1^{\epsilon_1}}, \mathbf{p}^{T_1^{\epsilon_1}}\right))\right| \leq \eta^{T_1^{\epsilon_1}} M_G$. Thus, when the chosen jumping period $T_1^{\epsilon_1}$ is sufficiently large so that the step-size $\eta^{T_1^{\epsilon_1}}$ is considerably smaller than $\epsilon_1$, we further derive that

$$\left|p_H^{\star\star} - p_H^{T_2^{\epsilon_1}}\right| \geq (b_H + c_H) \cdot \left(\epsilon_1 - \eta^{T_1^{\epsilon_1}} M_G\right) \geq (b_H + c_H) \cdot \eta^{T_1^{\epsilon_1}} M_G \geq \left|p_H^{\star\star} - p_H^{T_1^{\epsilon_1}}\right|. \quad (47)$$

With Eq. (47), we use Lemma F.4 again to upper-bound the last term on the right-hand side of Eq. (46):

$$\text{sign}\left(p_H^{\star\star} - p_H^{T_1^{\epsilon_1}}\right) \cdot \left(G_H\left((p_H^{T_2^{\epsilon_1}}, p_L^{T_1^{\epsilon_1}}), (p_H^{T_2^{\epsilon_1}}, p_L^{T_1^{\epsilon_1}})\right) - G_H\left(\mathbf{p}^{T_1^{\epsilon_1}}, \mathbf{p}^{T_1^{\epsilon_1}}\right)\right)$$

$$= \text{sign}\left(p_H^{\star\star} - p_H^{T_2^{\epsilon_1}}\right) \cdot G_H\left((p_H^{T_2^{\epsilon_1}}, p_L^{T_1^{\epsilon_1}}), (p_H^{T_2^{\epsilon_1}}, p_L^{T_1^{\epsilon_1}})\right) - \text{sign}\left(p_H^{\star\star} - p_H^{T_1^{\epsilon_1}}\right) \cdot G_H\left(\mathbf{p}^{T_1^{\epsilon_1}}, \mathbf{p}^{T_1^{\epsilon_1}}\right)$$

$$\overset{(\Delta_1)}{=} \left| \left( \frac{1}{(b_H + c_H) p_H^{T_2^{\epsilon_1}}} + d_H\left((p_H^{T_2^{\epsilon_1}}, p_L^{T_1^{\epsilon_1}}), (p_H^{T_2^{\epsilon_1}}, p_L^{T_1^{\epsilon_1}})\right) - 1 \right) \right.$$

$$\left. - \left( \frac{1}{(b_H + c_H) p_H^{T_1^{\epsilon_1}}} + d_H\left(\mathbf{p}^{T_1^{\epsilon_1}}, \mathbf{p}^{T_1^{\epsilon_1}}\right) - 1 \right) \right|$$

$$= \left| \underbrace{\frac{1}{b_H + c_H}\left( \frac{1}{p_H^{T_2^{\epsilon_1}}} - \frac{1}{p_H^{T_1^{\epsilon_1}}} \right)}_{\text{diff}_1} + \underbrace{\left[ d_H\left((p_H^{T_2^{\epsilon_1}}, p_L^{T_1^{\epsilon_1}}), (p_H^{T_2^{\epsilon_1}}, p_L^{T_1^{\epsilon_1}})\right) - d_H\left(\mathbf{p}^{T_1^{\epsilon_1}}, \mathbf{p}^{T_1^{\epsilon_1}}\right) \right]}_{\text{diff}_2} \right|$$

$$\overset{(\Delta_2)}{\geq} \frac{1}{b_H + c_H} \left| \frac{1}{p_H^{T_2^{\epsilon_1}}} - \frac{1}{p_H^{T_1^{\epsilon_1}}} \right|,$$

(48)

where $(\Delta_1)$ holds since the difference in the previous step is non-negative by Lemma F.4. Furthermore, it is straightforward to observe that the terms $\text{diff}_1$ and $\text{diff}_2$ have the same sign, which results in the final inequality $(\Delta_2)$. We substitute Eq. (48) back into Eq. (46):

$$\text{sign}\left(p_H^{\star\star} - p_H^{T_2^{\epsilon_1}}\right) \cdot G_H\left(\mathbf{p}^{T_2^{\epsilon_1}}, \mathbf{r}^{T_2^{\epsilon_1}}\right)$$

$$\geq \text{sign}\left(p_H^{\star\star} - p_H^{T_1^{\epsilon_1}}\right) \cdot G_H\left(\mathbf{p}^{T_1^{\epsilon_1}}, \mathbf{p}^{T_1^{\epsilon_1}}\right) - \ell_r \|\mathbf{r}^{T_2^{\epsilon_1}} - \mathbf{p}^{T_2^{\epsilon_1}}\|_2 + \frac{1}{b_H + c_H} \left| \frac{1}{p_H^{T_2^{\epsilon_1}}} - \frac{1}{p_H^{T_1^{\epsilon_1}}} \right|$$

$$\geq \text{sign}\left(p_H^{\star\star} - p_H^{T_1^{\epsilon_1}}\right) \cdot G_H\left(\mathbf{p}^{T_1^{\epsilon_1}}, \mathbf{r}^{T_1^{\epsilon_1}}\right) - \ell_r \left( \|\mathbf{r}^{T_2^{\epsilon_1}} - \mathbf{p}^{T_2^{\epsilon_1}}\|_2 + \|\mathbf{r}^{T_1^{\epsilon_1}} - \mathbf{p}^{T_1^{\epsilon_1}}\|_2 \right)$$

$$+ \frac{1}{b_H + c_H} \left| \frac{1}{p_H^{T_2^{\epsilon_1}}} - \frac{1}{p_H^{T_1^{\epsilon_1}}} \right|$$

$$\overset{(\Delta_1)}{\geq} - \ell_r \left( \|\mathbf{r}^{T_2^{\epsilon_1}} - \mathbf{p}^{T_2^{\epsilon_1}}\|_2 + \|\mathbf{r}^{T_1^{\epsilon_1}} - \mathbf{p}^{T_1^{\epsilon_1}}\|_2 \right) + \frac{\left| p_H^{T_2^{\epsilon_1}} - p_H^{T_1^{\epsilon_1}} \right|}{(b_H + c_H) \cdot \left( p_H^{T_2^{\epsilon_1}} \cdot p_H^{T_1^{\epsilon_1}} \right)}$$

$$\overset{(\Delta_2)}{\geq} - \ell_r \left( \|\mathbf{r}^{T_2^{\epsilon_1}} - \mathbf{p}^{T_2^{\epsilon_1}}\|_2 + \|\mathbf{r}^{T_1^{\epsilon_1}} - \mathbf{p}^{T_1^{\epsilon_1}}\|_2 \right) + \frac{\left| (\epsilon_1 - \eta^{T_1^{\epsilon_1}} M_G) - \eta^{T_1^{\epsilon_1}} M_G \right|}{(b_H + c_H) \cdot \bar{p}^2}.$$

(49)

In inequality $(\Delta_1)$, we use the fact that $\text{sign}\left(p_H^{\star\star} - p_H^{T_1^{\epsilon_1}}\right) \cdot G_H\left(\mathbf{p}^{T_1^{\epsilon_1}}, \mathbf{r}^{T_1^{\epsilon_1}}\right) > 0$ since $T_1^{\epsilon_1}$ is a jumping period in $\mathcal{T}_{\epsilon_1}$. Then, inequality $(\Delta_2)$ is implied by the properties that $\text{sign}\left(p_H^{\star\star} - p_H^{T_2^{\epsilon_1}}\right) = \text{sign}\left(p_H^{\star\star} - p_H^{T_1^{\epsilon_1}}\right)$, $\left| p_H^{\star\star} - p_H^{T_2^{\epsilon_1}} \right| / (b_H + c_H) \geq \epsilon_1 - \eta^{T_1^{\epsilon_1}} M_G$, and $\left| p_H^{\star\star} - p_H^{T_1^{\epsilon_1}} \right| / (b_H + c_H) \leq \eta^{T_1^{\epsilon_1}} M_G$. Since both the step-size $\eta^t$ and the difference $\|\mathbf{r}^t - \mathbf{p}^t\|_2$ decrease to 0 as $t \to \infty$, by choosing a sufficiently large jumping period $T_1^{\epsilon_1}$ from $\mathcal{T}_{\epsilon_1}$, the right-hand side of Eq. (46) is guaranteed to be non-negative, i.e., $\text{sign}\left(p_H^{\star\star} - p_H^{T_2^{\epsilon_1}}\right) \cdot G_H\left(\mathbf{p}^{T_2^{\epsilon_1}}, \mathbf{r}^{T_2^{\epsilon_1}}\right) \geq 0$. This completes the proof of scenario 1.

2. Suppose $\mathbf{p}^{T_2^{\epsilon_1}} \in N_2 \cup N_4$. We deduce that

$$
\text{sign}\left(p_H^{\star\star} - p_H^{T_2^{\epsilon_1}}\right) \cdot G_H\left(\mathbf{p}^{T_2^{\epsilon_1}}, \mathbf{r}^{T_2^{\epsilon_1}}\right)
$$

$$
\geq \ \text{sign}\left(p_H^{\star\star} - p_H^{T_2^{\epsilon_1}}\right) \cdot G_H\left(\mathbf{p}^{T_2^{\epsilon_1}}, \mathbf{p}^{T_2^{\epsilon_1}}\right) - \ell_r \|\mathbf{r}^{T_2^{\epsilon_1}} - \mathbf{p}^{T_2^{\epsilon_1}}\|_2
$$

$$
\overset{(\Delta_1)}{\geq} \ \text{sign}\left(p_H^{\star\star} - p_H^{T_2^{\epsilon_1}}\right) \cdot G_H\left((p_H^{T_2^{\epsilon_1}}, p_L^{\star\star}), (p_H^{T_2^{\epsilon_1}}, p_L^{\star\star})\right) - \ell_r \|\mathbf{r}^{T_2^{\epsilon_1}} - \mathbf{p}^{T_2^{\epsilon_1}}\|_2 \qquad (50)
$$

$$
\overset{(\Delta_2)}{=} \mathcal{G}\left((p_H^{T_2^{\epsilon_1}}, p_L^{\star\star})\right) - \ell_r \|\mathbf{r}^{T_2^{\epsilon_1}} - \mathbf{p}^{T_2^{\epsilon_1}}\|_2
$$

$$
\overset{(\Delta_3)}{\geq} M_{\widehat{\epsilon}_1} - \ell_r \|\mathbf{r}^{T_2^{\epsilon_1}} - \mathbf{p}^{T_2^{\epsilon_1}}\|_2, \quad \text{where } \widehat{\epsilon}_1 := \epsilon_1 - \eta^{T_2^{\epsilon_1}} M_G.
$$

The inequality $(\Delta_1)$ follows from Lemma F.4, where $\text{sign}\left(p_H^{\star\star} - p_H\right) \cdot G_H\left(\mathbf{p}, \mathbf{p}\right)$ is deceasing when $|p_L^{\star\star} - p_L|$ decreases . The equality $(\Delta_2)$ is due to the definition of $\mathcal{G}(\mathbf{p})$ in Eq. (31) and the fact that $\text{sign}(p_L^{\star\star} - p_L^{\star\star}) = 0$. Finally, in the last line $(\Delta_3)$, we leverage the initial assumption of **Case 2**, which implies that $\varepsilon\left((p_H^{T_2^{\epsilon_1}}, p_L^{\star\star})\right) = \left|p_H^{\star\star} - p_H^{T_2^{\epsilon_1}}\right|/(b_H + c_H) \in \left[\epsilon_1 - \eta^{T_1^{\epsilon_1}} M_G, \epsilon_1\right)$. Then, by Lemma F.2, there must exist $M_{\widehat{\epsilon}_1} > 0$ with $\widehat{\epsilon}_1 := \epsilon_1 - \eta^{T_2^{\epsilon_1}} M_G$ such that $\mathcal{G}\left((p_H^{T_2^{\epsilon_1}}, p_L^{\star\star})\right) \geq M_{\widehat{\epsilon}_1}$. As $|\mathcal{T}_{\epsilon_1}| = \infty$, we can pick a sufficiently large $T_1^{\epsilon_1} \in \mathcal{T}_{\epsilon_1}$ to guarantee that $\ell_r \|\mathbf{r}^t - \mathbf{p}^t\|_2 \leq M_{\widehat{\epsilon}_1}$ for $t \geq T_1^{\epsilon_1}$ by Lemma F.1. Consequently, we obtain the desired conclusion that $\text{sign}\left(p_H^{\star\star} - p_H^{T_2^{\epsilon_1}}\right) \cdot G_H\left(\mathbf{p}^{T_2^{\epsilon_1}}, \mathbf{r}^{T_2^{\epsilon_1}}\right) > 0$ when $\mathbf{p}^{T_2^{\epsilon_1}} \in N_2 \cup N_4$, which completes the proof of scenario 2.

Combining **Case 1** and **Case 2**, we have shown that $\left|p_H^{\star\star} - p_H^{T_2^{\epsilon_1}+1}\right|/(b_H + c_H) < \epsilon_1$, which completes the strong induction, i.e., $\left|p_H^{\star\star} - p_H^t\right|/(b_H + c_H) < \epsilon_1$ for all $t \geq T_1^{\epsilon_1}$. Under the assumption that $\{\mathbf{p}^t\}_{t \geq 0}$ never visits $N_{\epsilon_0}^1$ if $t \geq T^{\epsilon_0}$, we accordingly have that $\left|p_L^{\star\star} - p_L^t\right|/(b_L + c_L) > (\epsilon_0 - \epsilon_1)$ for all $t \geq T_1^{\epsilon_1}$. Therefore, following the derivations from Eq. (37) to Eq. (41), we deduce that Eq. (42) holds for all $t \geq T_1^{\epsilon_1}$. Using a telescoping sum, it holds for any period $T > T_1^{\epsilon_1}$ that

$$
\frac{\left|p_L^{\star\star} - p_L^T\right|}{b_L + c_L} \leq \frac{\left|p_L^{\star\star} - p_L^{T_1^{\epsilon_1}}\right|}{b_L + c_L} - \frac{M_{\epsilon_0 - \epsilon_1}}{2} \sum_{t=T_1^{\epsilon_1}}^{T-1} \eta^t. \qquad (51)
$$

Since $M_{\epsilon_0 - \epsilon_1} > 0$ is a constant and $\lim_{T \to \infty} \sum_{t=T_1^{\epsilon_1}}^{T-1} \eta^t = \infty$, we arrive at the contradiction that $\left|p_L^{\star\star} - p_L^T\right|/(b_L + c_L) \to -\infty$ as $T \to \infty$. Consequently, our initial assumption is incorrect and the price path $\{\mathbf{p}^t\}_{t \geq 0}$ would visit the $\ell_1$-neighborhood $N_{\epsilon_0}^1$ infinitely many times for any $\epsilon_0 > 0$, which completes the proof of **Part 1**.

## C.2 Proof of Part 2

In particular, we show that when $\mathbf{p}^t \in N_{\epsilon_0}^2$ for some sufficiently large $t$, then it also holds $\mathbf{p}^{t+1} \in N_{\epsilon_0}^2$. The value of $\epsilon_0$ will be specified later in the proof.

By the update rule of Algorithm 1, it holds that

$$
\begin{aligned}
\left| p_i^{\star\star} - p_i^{t+1} \right|^2 &= \left| p_i^{\star\star} - \text{Proj}_{\mathcal{P}} \left( p_i^t + \eta^t D_i^t \right) \right|^2 \\
&\leq \left| p_i^{\star\star} - \left( p_i^t + \eta^t D_i^t \right) \right|^2 \\
&= \left| \left( p_i^{\star\star} - p_i^t \right) - \eta^t (b_i + c_i) \cdot G_i(\mathbf{p}^t, \mathbf{r}^t) \right|^2 \\
&= \left| p_i^{\star\star} - p_i^t \right|^2 - 2\eta^t (b_i + c_i) \cdot G_i(\mathbf{p}^t, \mathbf{r}^t) \cdot \left( p_i^{\star\star} - p_i^t \right) + \left( \eta^t (b_i + c_i) \cdot G_i(\mathbf{p}^t, \mathbf{r}^t) \right)^2 \\
&= \left| p_i^{\star\star} - p_i^t \right|^2 - 2\eta^t (b_i + c_i) \cdot G_i(\mathbf{p}^t, \mathbf{p}^t) \cdot \left( p_i^{\star\star} - p_i^t \right) + \left( \eta^t (b_i + c_i) \cdot G_i(\mathbf{p}^t, \mathbf{r}^t) \right)^2 \\
&\quad + 2\eta^t (b_i + c_i) \cdot \left( G_i(\mathbf{p}^t, \mathbf{p}^t) - G_i(\mathbf{p}^t, \mathbf{r}^t) \right) \cdot \left( p_i^{\star\star} - p_i^t \right) \\
&\leq \left| p_i^{\star\star} - p_i^t \right|^2 - 2\eta^t (b_i + c_i) \cdot G_i(\mathbf{p}^t, \mathbf{p}^t) \cdot \left( p_i^{\star\star} - p_i^t \right) + \left( \eta^t M_G (b_i + c_i) \right)^2 \\
&\quad + 2\eta^t (b_i + c_i) \cdot \left| p_i^{\star\star} - p_i^t \right| \cdot \ell_r \| \mathbf{r}^t - \mathbf{p}^t \|_2,
\end{aligned}
\tag{52}
$$

where we use $|G_i(\mathbf{p}, \mathbf{r})| \leq M_G$ and the mean value theorem in the last inequality (see Lemma F.5). Let $\mathcal{H}(\mathbf{p})$ be a function defined as

$$
\mathcal{H}(\mathbf{p}) := (b_H + c_H) \cdot G_H(\mathbf{p}, \mathbf{p}) \cdot (p_H^{\star\star} - p_H) + (b_L + c_L) \cdot G_L(\mathbf{p}, \mathbf{p}) \cdot (p_L^{\star\star} - p_L).
\tag{53}
$$

Then, by summing Eq. (52) over both products $i \in \{H, L\}$, we have that

$$
\begin{aligned}
\left\| \mathbf{p}^{\star\star} - \mathbf{p}^{t+1} \right\|_2^2 &\leq \left\| \mathbf{p}^{\star\star} - \mathbf{p}^t \right\|_2^2 - 2\eta^t \sum_{i \in \{H,L\}} (b_i + c_i) \cdot G_i(\mathbf{p}^t, \mathbf{p}^t) \cdot \left( p_i^{\star\star} - p_i^t \right) \\
&\quad + (\eta^t M_G)^2 \sum_{i \in \{H,L\}} (b_i + c_i)^2 + 2\eta^t \ell_r \| \mathbf{r}^t - \mathbf{p}^t \|_2 \sum_{i \in \{H,L\}} (b_i + c_i) \cdot \left| p_i^{\star\star} - p_i^t \right| \\
&= \left\| \mathbf{p}^{\star\star} - \mathbf{p}^t \right\|_2^2 - 2\eta^t \mathcal{H}(\mathbf{p}^t) + (\eta^t M_G)^2 \sum_{i \in \{H,L\}} (b_i + c_i)^2 \\
&\quad + 2\eta^t \ell_r \| \mathbf{r}^t - \mathbf{p}^t \|_2 \sum_{i \in \{H,L\}} (b_i + c_i) \cdot \left| p_i^{\star\star} - p_i^t \right| \\
&\leq \left\| \mathbf{p}^{\star\star} - \mathbf{p}^t \right\|_2^2 - \eta^t \left( 2\mathcal{H}(\mathbf{p}^t) - \eta^t C_1 - C_2 \| \mathbf{r}^t - \mathbf{p}^t \|_2 \right)
\end{aligned}
\tag{54}
$$

where we denote $C_1 := (M_G)^2 \cdot \sum_{i \in \{H,L\}} (b_i + c_i)^2$ and $C_2 = 2\ell_r |\overline{p} - \underline{p}| \cdot \sum_{i \in \{H,L\}} (b_i + c_i)$.

By Lemma F.3, there exist $\gamma > 0$ and a open set $U_\gamma \ni \mathbf{p}^{\star\star}$ such that $\mathcal{H}(\mathbf{p}) \geq \gamma \cdot \| \mathbf{p} - \mathbf{p}^{\star\star} \|_2^2$, $\forall \mathbf{p} \in U_\gamma$. Consider $\epsilon_0 > 0$ such that the $\ell_2$-neighborhood $N_{\epsilon_0}^2 = \{ \mathbf{p} \in \mathcal{P}^2 \mid \| \mathbf{p} - \mathbf{p}^{\star\star} \|_2 < \epsilon_0 \} \subset U_\gamma$. Furthermore, let $T_\gamma$ be some period such that

$$
\eta^t \left( \eta^t C_1 + \sqrt{2} C_2 (\overline{p} - \underline{p}) \right) \leq \frac{\epsilon_0^2}{4}, \text{ and } \eta^t C_1 + C_2 \| \mathbf{r}^t - \mathbf{p}^t \|_2 \leq \frac{\gamma (\epsilon_0)^2}{2}, \quad \forall t > T_\gamma.
\tag{55}
$$

The existence of such a number $T_\gamma$ follows from the fact that $\lim_{t \to \infty} \eta^t = 0$ and $\lim_{t \to \infty} \| \mathbf{r}^t - \mathbf{p}^t \|_2 = 0$ (see Lemma F.1). Below, we discuss two cases depending on the location of $\mathbf{p}^t$ in $N_{\epsilon_0}^2$.

**Case 1:** $\mathbf{p}^t \in N_{\epsilon_0/2}^2 \subset N_{\epsilon_0}^2$, i.e., $\left\| \mathbf{p}^{\star\star} - \mathbf{p}^t \right\|_2 < \epsilon_0/2$.

Since $\mathcal{H}(\mathbf{p}) \geq 0, \forall \mathbf{p} \in U_\gamma$ by Lemma F.3, it follows from Eq. (54) and Eq. (55) that

$$
\begin{aligned}
\left\|\mathbf{p}^{\star\star} - \mathbf{p}^{t+1}\right\|_2^2 &\leq \left\|\mathbf{p}^{\star\star} - \mathbf{p}^t\right\|_2^2 + \eta^t \left(\eta^t C_1 + C_2 \|\mathbf{r}^t - \mathbf{p}^t\|_2\right) \\
&\overset{(\Delta)}{\leq} \frac{(\epsilon_0)^2}{4} + \eta^t \left(\eta^t C_1 + \sqrt{2} C_2 (\bar{p} - \underline{p})\right) \\
&\leq \frac{(\epsilon_0)^2}{4} + \frac{(\epsilon_0)^2}{4} \\
&< (\epsilon_0)^2,
\end{aligned}
\tag{56}
$$

where inequality $(\Delta)$ is due to $\|\mathbf{r}^t - \mathbf{p}^t\|_2 \leq \sqrt{2}(\bar{p} - \underline{p})$. Eq. (56) implies that $\mathbf{p}^{t+1} \in N_{\epsilon_0}^2$.

**Case 2:** $\mathbf{p}^t \in N_{\epsilon_0}^2 \setminus N_{\epsilon_0/2}^2$, i.e., $\left\|\mathbf{p}^{\star\star} - \mathbf{p}^t\right\|_2 \in [\epsilon_0/2, \epsilon_0)$.

By Lemma F.3, we have that $\mathcal{H}(\mathbf{p}^t) \geq \gamma \left\|\mathbf{p}^{\star\star} - \mathbf{p}^t\right\|_2^2 \geq \gamma(\epsilon_0)^2/4$. Thus, again by Eq. (54) and Eq. (55), we have that

$$
\begin{aligned}
\left\|\mathbf{p}^{\star\star} - \mathbf{p}^{t+1}\right\|_2^2 &\leq \left\|\mathbf{p}^{\star\star} - \mathbf{p}^t\right\|_2^2 - \eta^t \left(2\mathcal{H}(\mathbf{p}^t) - \eta^t C_1 - C_2 \|\mathbf{r}^t - \mathbf{p}^t\|_2\right) \\
&\leq \left\|\mathbf{p}^{\star\star} - \mathbf{p}^t\right\|_2^2 - \eta^t \left(\frac{\gamma(\epsilon_0)^2}{2} - \eta^t C_1 - C_2 \|\mathbf{r}^t - \mathbf{p}^t\|_2\right) \\
&\leq \left\|\mathbf{p}^{\star\star} - \mathbf{p}^t\right\|_2^2 \\
&\leq (\epsilon_0)^2,
\end{aligned}
\tag{57}
$$

which implies $\mathbf{p}^{t+1} \in N_{\epsilon_0}^2$. Therefore, we conclude by induction that the price path will stay in the $\ell_2$-neighborhood $N_{\epsilon_0}^2$. This completes the proof of Theorem 5.1. $\qquad\square$

## Appendix D   Proof of Theorem 5.2

*Proof.* Recall the function $\mathcal{H}(\mathbf{p})$ defined in Eq. (53). By Lemma F.3, there exist $\gamma > 0$ and a open set $U_\gamma \ni \mathbf{p}^{\star\star}$ such that $\mathcal{H}(\mathbf{p}) \geq \gamma \cdot \|\mathbf{p} - \mathbf{p}^{\star\star}\|_2^2, \forall \mathbf{p} \in U_\gamma$. Consider $\epsilon_0 > 0$ such that the $\ell_2$-neighborhood $N_{\epsilon_0}^2 = \left\{\mathbf{p} \in \mathcal{P}^2 \mid \|\mathbf{p} - \mathbf{p}^{\star\star}\|_2 < \epsilon_0\right\} \subset U_\gamma$. Below, we first show that the price path $\{\mathbf{p}^t\}_{t \geq 0}$ enjoys the sublinear convergence rate in $N_{\epsilon_0}^2$ when $t$ is greater than some constant $T_{\epsilon_0}$. Then, we will show that this convergence rate also holds for any $t \leq T_{\epsilon_0}$.

By **Part 2** in the proof of Theorem 5.1, there exists $T_{\epsilon_0} > 0$ such that $\mathbf{p}^t \in N_{\epsilon_0}^2$ for every $t \geq T_{\epsilon_0}$. Following a similar argument as Eq. (52) and Eq. (54), we have that

$$
\begin{aligned}
&\left\|\mathbf{p}^{\star\star} - \mathbf{p}^{t+1}\right\|_2^2 \\
&\overset{(\Delta_1)}{\leq} \left\|\mathbf{p}^{\star\star} - \mathbf{p}^t\right\|_2^2 - 2\eta^t \mathcal{H}(\mathbf{p}^t) + C_1 (\eta^t)^2 + 2\eta^t \ell_r \|\mathbf{r}^t - \mathbf{p}^t\|_2 \sum_{i \in \{H, L\}} (b_i + c_i) \cdot \left|p_i^{\star\star} - p_i^t\right| \\
&\overset{(\Delta_2)}{\leq} \left\|\mathbf{p}^{\star\star} - \mathbf{p}^t\right\|_2^2 - 2\eta^t \gamma \left\|\mathbf{p}^{\star\star} - \mathbf{p}^t\right\|_2^2 + C_1 (\eta^t)^2 + 2\eta^t \ell_r \|\mathbf{r}^t - \mathbf{p}^t\|_2 \cdot \hat{k} \left\|\mathbf{p}^{\star\star} - \mathbf{p}^t\right\|_2 \\
&\overset{(\Delta_3)}{\leq} \left\|\mathbf{p}^{\star\star} - \mathbf{p}^t\right\|_2^2 - 2\eta^t \gamma \left\|\mathbf{p}^{\star\star} - \mathbf{p}^t\right\|_2^2 + C_1 (\eta^t)^2 + \eta^t \ell_r \hat{k} \left[\frac{\gamma}{\ell_r \hat{k}} \left\|\mathbf{p}^{\star\star} - \mathbf{p}^t\right\|_2^2 + \frac{\ell_r \hat{k}}{\gamma} \left\|\mathbf{r}^t - \mathbf{p}^t\right\|_2^2\right] \\
&\overset{(\Delta_4)}{\leq} \left\|\mathbf{p}^{\star\star} - \mathbf{p}^t\right\|_2^2 - \eta^t \gamma \left\|\mathbf{p}^{\star\star} - \mathbf{p}^t\right\|_2^2 + C_1 (\eta^t)^2 + \eta^t k \left\|\mathbf{r}^t - \mathbf{p}^t\right\|_2^2.
\end{aligned}
\tag{58}
$$

In step $(\Delta_1)$, $\ell_r$ is the Lipschitz constant defined in Eq. (100) and the constant $C_1$ is defined in Eq. (54). In step $(\Delta_2)$, we utilize Lemma F.3 and the following inequality

$$
\sum_{i \in \{H,L\}} (b_i + c_i)|p_i^{\star\star} - p_i^t| \leq \max_{i \in \{H,L\}} \{b_i + c_i\} \|\mathbf{p}^{\star\star} - \mathbf{p}^t\|_1
$$

$$
\leq \sqrt{2} \max_{i \in \{H,L\}} \{b_i + c_i\} \|\mathbf{p}^{\star\star} - \mathbf{p}^t\|_2 \tag{59}
$$

$$
= \hat{k} \|\mathbf{p}^{\star\star} - \mathbf{p}^t\|_2,
$$

where we define $\hat{k} := \sqrt{2} \max_{i \in \{H,L\}}\{b_i + c_i\}$. Step $(\Delta_3)$ in Eq. (58) follows from the inequality of arithmetic and geometric means, i.e., $2xy \leq Ax^2 + (1/A)y^2$ for any constant $A > 0$. The value of constant $k$ in $(\Delta_4)$ is given by $k := (\ell_r \hat{k})^2/\gamma = 2\big(\ell_r \max_{i \in \{H,L\}}\{b_i + c_i\}\big)^2/\gamma$.

To upper-bound the right-hand side of Eq. (58), we first focus on the term $\|\mathbf{r}^t - \mathbf{p}^t\|_2^2$ and inductively show that

$$
\|\mathbf{r}^t - \mathbf{p}^t\|_2^2 = \mathcal{O}\left(\frac{1}{t^2}\right), \quad \forall t \geq T_\lambda := \frac{\sqrt{\lambda+1}}{\sqrt{\lambda+1} - \sqrt{2\lambda}}, \tag{60}
$$

where $\lambda := (1+\alpha^2)/2 < 1$. We use the notation $\mathbf{D}^t := \big(D_H^t, D_L^t\big) = \big((b_H + c_H)G_H(\mathbf{p}^t, \mathbf{r}^t), (b_L + c_L)G_L(\mathbf{p}^t, \mathbf{r}^t)\big)$, where we recall that $D_i^t$ is the partial derivative specified in Eq. (9) and function $G_i(\cdot, \cdot)$ is defined in Eq. (25). Then, the term $\|\mathbf{r}^t - \mathbf{p}^t\|_2^2$ can be upper-bounded as follows

$$
\begin{aligned}
&\|\mathbf{r}^t - \mathbf{p}^t\|_2^2 \\
&= \|\alpha \mathbf{r}^{t-1} + (1-\alpha)\mathbf{p}^{t-1} - \mathbf{p}^t\|_2^2 \\
&= \|\alpha(\mathbf{r}^{t-1} - \mathbf{p}^{t-1}) + (\mathbf{p}^{t-1} - \mathbf{p}^t)\|_2^2 \\
&= \alpha^2 \|\mathbf{r}^{t-1} - \mathbf{p}^{t-1}\|_2^2 + \|\mathbf{p}^{t-1} - \mathbf{p}^t\|_2^2 + 2\alpha(\mathbf{r}^{t-1} - \mathbf{p}^{t-1})^\top(\mathbf{p}^{t-1} - \mathbf{p}^t) \\
&\overset{(\Delta_1)}{\leq} \alpha^2 \|\mathbf{r}^{t-1} - \mathbf{p}^{t-1}\|_2^2 + \|\eta^{t-1}\mathbf{D}^{t-1}\|_2^2 + 2\alpha \|\mathbf{r}^{t-1} - \mathbf{p}^{t-1}\|_2 \|\eta^{t-1}\mathbf{D}^{t-1}\|_2 \\
&\overset{(\Delta_2)}{\leq} \alpha^2 \|\mathbf{r}^{t-1} - \mathbf{p}^{t-1}\|_2^2 + \|\eta^{t-1}\mathbf{D}^{t-1}\|_2^2 + \frac{1-\alpha^2}{2}\|\mathbf{r}^{t-1} - \mathbf{p}^{t-1}\|_2^2 + \frac{2\alpha^2}{1-\alpha^2}\|\eta^{t-1}\mathbf{D}^{t-1}\|_2^2 \\
&= \frac{1+\alpha^2}{2}\|\mathbf{r}^{t-1} - \mathbf{p}^{t-1}\|_2^2 + \frac{1+\alpha^2}{1-\alpha^2}\|\eta^{t-1}\mathbf{D}^{t-1}\|_2^2 \\
&= \lambda \|\mathbf{r}^{t-1} - \mathbf{p}^{t-1}\|_2^2 + \frac{1+\alpha^2}{1-\alpha^2} \cdot \left(\sum_{i \in \{H,L\}} (b_i + c_i)^2 \left[G_i(\mathbf{p}^{t-1}, \mathbf{r}^{t-1})\right]^2\right) \cdot (\eta^{t-1})^2 \\
&\overset{(\Delta_3)}{\leq} \lambda \|\mathbf{r}^{t-1} - \mathbf{p}^{t-1}\|_2^2 + \underbrace{\left(\frac{1+\alpha^2}{1-\alpha^2} \cdot (M_G)^2 \sum_{i \in \{H,L\}} (b_i + c_i)^2\right)}_{\lambda_0} \cdot (\eta^{t-1})^2,
\end{aligned}
$$

$$\tag{61}$$

where $(\Delta_1)$ holds due to the Cauchy-Schwarz inequality and the property of the projection operator (see Line 5 in Algorithm 1). Step $(\Delta_2)$ is derived from the inequality of arithmetic and geometric means. Lastly, step $(\Delta_3)$ applies the upper bound on function $G_i(\cdot, \cdot)$ in Lemma F.5 and the definition of $C_1$ in Eq. (54). For the simplicity of notation, we denote the coefficient of $(\eta^{t-1})^2$ in the last line as $\lambda_0$.

Let the step-size be $\eta^t = d_\eta/(t+1)$ for $t \geq 0$, where $d_\eta$ is some constant that will be determined later. Suppose that there exists a constant $d_{rp}$ such that

$$
\|\mathbf{r}^{t-1} - \mathbf{p}^{t-1}\|_2^2 \leq \frac{d_{rp}}{(t-1)^2}, \tag{62}
$$

for some $t \geq T_\lambda + 1$. Then, together with Eq. (61), we have that

$$
\begin{aligned}
\left\|\mathbf{r}^t - \mathbf{p}^t\right\|_2^2 &\leq \lambda \left\|\mathbf{r}^{t-1} - \mathbf{p}^{t-1}\right\|_2^2 + \lambda_0 (\eta^{t-1})^2 \\
&\leq \frac{\lambda d_{rp}}{(t-1)^2} + \frac{\lambda_0 (d_\eta)^2}{t^2} \\
&= \frac{\lambda d_{rp}}{t^2} \cdot \frac{t^2}{(t-1)^2} + \frac{\lambda_0 (d_\eta)^2}{t^2} \\
&\overset{(\Delta)}{\leq} \frac{\lambda d_{rp}}{t^2} \cdot \frac{\lambda + 1}{2\lambda} + \frac{\lambda_0 (d_\eta)^2}{t^2} \\
&= \frac{0.5(\lambda + 1) \cdot d_{rp} + \lambda_0 (d_\eta)^2}{t^2},
\end{aligned}
\tag{63}
$$

where the inequality $(\Delta)$ results from the choice of $T_\lambda$. Hence, the induction follows if $0.5(\lambda + 1)d_{rp} + \lambda_0 (d_\eta)^2 \leq d_{rp}$, which is further equivalent to

$$
d_{rp} \geq \frac{2\lambda_0 (d_\eta)^2}{1 - \lambda}.
\tag{64}
$$

Lastly, the base case of the induction requires that $\left\|\mathbf{r}^{T_\lambda} - \mathbf{p}^{T_\lambda}\right\|_2^2 \leq \frac{d_{rp}}{(T_\lambda)^2}$. Therefore, by Eq. (64) and the definition of feasible price range $\mathcal{P} = [\underline{p}, \overline{p}]$, it suffices to choose

$$
d_{rp} = \max \left\{ \frac{2\lambda_0 (d_\eta)^2}{1 - \lambda}, \ 2(T_\lambda)^2 (\overline{p} - \underline{p})^2 \right\},
\tag{65}
$$

where the constants $\lambda_0$ and $T_\lambda$ are respectively defined in Eq. (61) and Eq. (60). Note that, under this choice of constant $d_{rp}$, it also holds that

$$
\left\|\mathbf{r}^t - \mathbf{p}^t\right\|_2^2 \leq 2(\overline{p} - \underline{p})^2 < \frac{2(T_\lambda)^2 (\overline{p} - \underline{p})^2}{t^2} \leq \frac{d_{rp}}{t^2}, \quad \forall 1 \leq t < T_\lambda.
\tag{66}
$$

Together with Eq. (60), we derive that

$$
\left\|\mathbf{r}^t - \mathbf{p}^t\right\|_2^2 \leq \frac{d_{rp}}{t^2}, \quad \forall t \geq 1.
\tag{67}
$$

For every $t \geq T_{\epsilon_0}$, we can further upper-bound the right-hand side of Eq. (58) by exploiting the upper bound of $\left\|\mathbf{r}^t - \mathbf{p}^t\right\|_2^2$ in Eq. (67) and the choice of $\eta_t = d_\eta / (t + 1)$:

$$
\begin{aligned}
\left\|\mathbf{p}^{\star\star} - \mathbf{p}^{t+1}\right\|_2^2 &\leq \left(1 - \frac{\gamma d_\eta}{t+1}\right) \left\|\mathbf{p}^{\star\star} - \mathbf{p}^t\right\|_2^2 + \frac{C_1 (d_\eta)^2}{(t+1)^2} + \frac{d_\eta k \cdot d_{rp}}{(t+1) t^2} \\
&\leq \left(1 - \frac{\gamma d_\eta}{t+1}\right) \left\|\mathbf{p}^{\star\star} - \mathbf{p}^t\right\|_2^2 + \frac{1}{t(t+1)} \underbrace{\left(C_1 (d_\eta)^2 + \frac{d_\eta k d_{rp}}{T_{\epsilon_0}}\right)}_{C_3}.
\end{aligned}
\tag{68}
$$

Now, we inductively show that

$$
\left\|\mathbf{p}^{\star\star} - \mathbf{p}^t\right\|_2^2 = \mathcal{O}\left(\frac{1}{t}\right), \quad \forall t \geq T_{\epsilon_0}.
\tag{69}
$$

Suppose there exists a constant $d_p$ such that for a fixed period $t \geq T_{\epsilon_0}$

$$
\left\|\mathbf{p}^{\star\star} - \mathbf{p}^t\right\|_2^2 \leq \frac{d_p}{t}.
\tag{70}
$$

To establish the induction, it suffices for the following inequality to hold

$$
\left\|\mathbf{p}^{\star\star} - \mathbf{p}^{t+1}\right\|_2^2 \leq \left(1 - \frac{\gamma d_\eta}{t+1}\right) \frac{d_p}{t} + \frac{C_3}{t(t+1)} \leq \frac{d_p}{t+1},
\tag{71}
$$

which is further equivalent to

$$(\gamma d_\eta - 1)d_p \geq C_3. \tag{72}$$

To satisfy the base case of the induction, we can select $d_p$ such that $d_p \geq T_{\epsilon_0} \cdot \left\| \mathbf{p}^{\star\star} - \mathbf{p}^{T_{\epsilon_0}} \right\|_2^2$. In summary, one possible set of constants $(d_\eta, d_{rp}, d_p)$ that satisfies all the requirements can be

$$d_\eta = \frac{2}{\gamma}, \quad d_{rp} = \max \left\{ \frac{2\lambda_0(d_\eta)^2}{1 - \lambda}, \, 2(T_\lambda)^2(\overline{p} - \underline{p})^2 \right\}, \quad d_p = \max\{C_3, 2T_{\epsilon_0}(\overline{p} - \underline{p})^2\}, \tag{73}$$

where the constants $\lambda_0$, $T_\lambda$, and $C_3$ are respectively defined in Eq. (61), Eq. (60), and Eq. (68). Now, for any period $1 \leq t < T_{\epsilon_0}$, it follows that

$$\left\| \mathbf{p}^{\star\star} - \mathbf{p}^t \right\|_2^2 \leq 2(\overline{p} - \underline{p})^2 < \frac{2T_{\epsilon_0}(\overline{p} - \underline{p})^2}{t} \leq \frac{d_p}{t}. \tag{74}$$

Together with Eq. (69), this proves the convergence rate for the price path, i.e.,

$$\left\| \mathbf{p}^{\star\star} - \mathbf{p}^t \right\|_2^2 \leq \frac{d_p}{t}, \quad \forall t \geq 1. \tag{75}$$

Finally, the convergence rate of the reference price path can be deduced from the following triangular inequality:

$$
\begin{aligned}
\left\| \mathbf{p}^{\star\star} - \mathbf{r}^t \right\|_2^2 &= \left\| \mathbf{p}^{\star\star} - \mathbf{p}^t + \mathbf{p}^t - \mathbf{r}^t \right\|_2^2 \\
&\leq 2\left\| \mathbf{p}^{\star\star} - \mathbf{p}^t \right\|_2^2 + 2\left\| \mathbf{p}^t - \mathbf{r}^t \right\|_2^2 \\
&\overset{(\Delta)}{\leq} \frac{2d_p}{t} + \frac{2d_{rp}}{t^2} \\
&\leq \frac{2d_p + 2d_{rp}}{t}, \quad \forall t \geq 1,
\end{aligned}
\tag{76}
$$

where $(\Delta)$ follows from Eq. (67) and Eq. (75). Therefore, it suffices to choose $d_r := 2d_p + 2d_{rp}$, and this completes the proof.

$\square$

## Appendix E  Proof of Theorem 6.1

*Proof.* Recall the definition of $G_i(\mathbf{p}, \mathbf{r})$ from Eq. (25). We can write the inexact derivative as $D_i^t = (b_i + c_i)G_i(\mathbf{p}^t, \mathbf{r}^t) + n_i^t$, where $n_i^t$ represents the error satisfying $|n_i^t| < \delta$, $\forall i \in \{H, L\}$ and $\forall t \geq 0$.

We also recall the two-part proof for Theorem 5.1: in Part 1, we show that the price path and reference price paths converge towards the SNE and visit the neighborhood $N_\epsilon^1$ infinitely many times, for any $\epsilon > 0$. In Part 2, we establish that, given $\epsilon$ is below a certain threshold $\epsilon_0$, the price vector would remain in the neighborhood $N_\epsilon^2$ after entering it with a sufficiently small step-size. Since the relative scale of $\delta$ and $\epsilon_0$ is unsure, our proof below is mainly based on Part 1.

In Part 1, the original proof assuming the exact gradient oracle employs a contradiction-based argument. Suppose the price path does not converge to the SNE, the proof consists of the following major steps (different from Appendix C, we use $\epsilon$ in lieu of $\epsilon_0$ to avoid confusion in the following proof):

- In Eq. (29), it is demonstrated that the price path steadily approaches the SNE if it stays in the same quadrant defined in Eq. (28). The main technique we use is $\mathcal{G}(\mathbf{p}) > M_\epsilon$ when $\varepsilon(\mathbf{p}) > \epsilon$. The definition of $\mathcal{G}(\mathbf{p})$ and the validation for the technique are stated in Lemma F.2.

- In Eq. (34), we show that the price path only oscillates between adjacent quadrants provided the step-sizes are sufficiently small.

- From Eqs (37) to (41), we prove that even when the price path does not stay in the same quadrant, it will still converge toward the SNE if it is at the boundary regions between two quadrants. The pivotal inequality employed here is again $\mathcal{G}((p_H^{\star\star}, p_L^{T_1^{\epsilon_1}})) \geq M_{\epsilon - \epsilon_1}$, provided $\varepsilon((p_H^{\star\star}, p_L^{T_1^{\epsilon_1}})) > \epsilon - \epsilon_1$ (see the end of Eqs. (38) and (40)).

- Finally, in Eq. (44) and subsequent equations, we provide supplementary justification for the above bullet that the price path remains adjacent to the boundaries given that the step-size is small. A crucial consideration here is opting for a much smaller $\epsilon_1$ relative to $\epsilon$ when defining the boundary region. In addition, the inequality $\mathcal{G}((p_H^{T_2^{\epsilon_1}}, p_L^{\star\star})) \geq M_{\widehat{\epsilon}_1}$ is also utilized in Eq. (50).

To summarize, the key to the proof is Lemma F.2., i.e., $\mathcal{G}(\mathbf{p}) > M_\epsilon > 0$ as long as $\varepsilon(\mathbf{p}) > \epsilon$. By definition, $\mathcal{G}(\mathbf{p}^t)$ approximately characterizes the difference $\varepsilon(\mathbf{p}^t) - \varepsilon(\mathbf{p}^{t+1})$ as seen in Eq. (29). As a result, the effect of the lower bound $M_\epsilon$ is that, if other terms can be upper-bounded by $M_\epsilon$, we can show that the updated price is closer to the SNE. For example, in the right-hand side of Eq. (29), once the term $2\ell_r \|\mathbf{r}^t - \mathbf{p}^t\|_2$ is below $M_\epsilon$, it suggests that the price vector is heading towards the SNE.

With inexact first-order oracles, the immediate consequence is that there will always be an error term accompanying $\mathcal{G}(\mathbf{p})$. For instance, given $D_i^t = (b_i + c_i)G_i(\mathbf{p}^t, \mathbf{r}^t) + n_i^t$, we observe that Eq. (29) evolves to

$$\varepsilon(\mathbf{p}^{t+1}) \leq \varepsilon(\mathbf{p}^t) - \eta^t \left( M_\epsilon - 2\ell_r \|\mathbf{r}^t - \mathbf{p}^t\|_2 - \sum_{i \in \{H,L\}} \frac{n_i^t}{(b_i + c_i)} \right). \tag{77}$$

This observation is consistent across the proof. Thus, if the error $n_i^t$ is substantially smaller than $M_\epsilon$, the proof of Theorem 5.1 remains valid, implying that the price vector converges towards the SNE. The subtlety arises when the size of $n_i^t$ is comparable with $M_\epsilon$. Under such circumstances, the analysis in both Eq. (29) and Eqs. (37) to (41) becomes invalid, as the errors are substantial enough to negate any assurance that the price path strictly approaches the SNE.

However, if the error $n_i^t$ is similar in magnitude to $\mathcal{G}(\mathbf{p})$, we can show that the price vector $\mathbf{p}$ is already close to the SNE $\mathbf{p}^{\star\star}$. More precisely, since the errors are bounded by $\delta$, it is equivalent to show that $\mathcal{G}(\mathbf{p}) = \mathcal{O}(\delta)$ also implies $\|\mathbf{p} - \mathbf{p}^{\star\star}\|_2 = \mathcal{O}(\delta)$. This can be proved by a refined version of Lemma F.2.

**Lemma E.1 (Refined Lemma F.2)** *Let $\mathcal{G}(\mathbf{p})$ be the function defined in Eq. (86). Then, it holds that*

$$\mathcal{G}(\mathbf{p}) \geq \sum_i \frac{|p_i - p_i^{\star\star}|}{(b_i + c_i)p_i^{\star\star}\overline{p}}, \tag{78}$$

*i.e., $\mathcal{G}(\mathbf{p})$ is lower-bounded by a weighted $\ell_1$ distance between $\mathbf{p}$ and $\mathbf{p}^{\star\star}$. By the equivalence of norms in the Euclidean space, there exists some constant $C$ such that $\mathcal{G}(\mathbf{p}) \geq C \cdot \|\mathbf{p} - \mathbf{p}^{\star\star}\|_2$.*

*Proof.* Similar to the proof of the original Lemma F.2, we separately consider the four possible scenarios where $\mathbf{p}$ belongs to one of the quadrant defined in Eq. (28).

1. Suppose $p_H > p_H^{\star\star}$ and $p_L \geq p_L^{\star\star}$, i.e., $\mathbf{p} \in N_1$. Since $G_i(\mathbf{p}^{\star\star}) = 0$, we have that

$$\mathcal{G}(\mathbf{p}) = \mathcal{G}(\mathbf{p}) - \mathcal{G}(\mathbf{p}^{\star\star}) = \sum_i \frac{1}{b_i + c_i} \left( \frac{1}{p_i^{\star\star}} - \frac{1}{p_i} \right) + d_0(\mathbf{p}, \mathbf{p}) - d_0(\mathbf{p}^{\star\star}, \mathbf{p}^{\star\star}). \tag{79}$$

By definition of the non-purchase probability, we observe that $d_0(\mathbf{p}, \mathbf{p}) - d_0(\mathbf{p}^{\star\star}, \mathbf{p}^{\star\star}) > 0$. Hence, the above equation implies that

$$\mathcal{G}(\mathbf{p}) > \sum_i \frac{1}{b_i + c_i} \cdot \frac{p_i - p_i^{\star\star}}{p_i^{\star\star}p_i} \geq \sum_i \frac{1}{b_i + c_i} \cdot \frac{p_i - p_i^{\star\star}}{p_i^{\star\star}\overline{p}} = \sum_i \frac{|p_i - p_i^{\star\star}|}{(b_i + c_i)p_i^{\star\star}\overline{p}}, \tag{80}$$

where we use the fact that $p_i \in \mathcal{P} = [\underline{p}, \overline{p}]$ and the presumption $\mathbf{p} \in N_1$.

2. Suppose $p_H \leq p_H^{\star\star}$ and $p_L > p_L^{\star\star}$, i.e., $\mathbf{p} \in N_2$. Again, using the fact that $G_i(\mathbf{p}^{\star\star}) = 0$, we derive that

$$\mathcal{G}(\mathbf{p}) = \frac{1}{b_H + c_H} \left( \frac{1}{p_H} - \frac{1}{p_H^{\star\star}} \right) + \frac{1}{b_L + c_L} \left( \frac{1}{p_L^{\star\star}} - \frac{1}{p_L} \right) + [d_H(\mathbf{p}, \mathbf{p}) - d_H(\mathbf{p}^{\star\star}, \mathbf{p}^{\star\star})] + [d_L(\mathbf{p}^{\star\star}, \mathbf{p}^{\star\star}) - d_L(\mathbf{p}, \mathbf{p})]. \tag{81}$$

Since $\mathbf{p} \in N_2$, we have that $d_H(\mathbf{p}, \mathbf{p}) - d_H(\mathbf{p}^{\star\star}, \mathbf{p}^{\star\star}) > 0$ and $d_L(\mathbf{p}^{\star\star}, \mathbf{p}^{\star\star}) - d_L(\mathbf{p}, \mathbf{p}) > 0$. Hence, similar to the first case, it follows that

$$\mathcal{G}(\mathbf{p}) > \frac{1}{b_H + c_H} \left( \frac{1}{p_H} - \frac{1}{p_H^{\star\star}} \right) + \frac{1}{b_L + c_L} \left( \frac{1}{p_L^{\star\star}} - \frac{1}{p_L} \right) \geq \sum_i \frac{|p_i - p_i^{\star\star}|}{(b_i + c_i)p_i^{\star\star}\overline{p}}. \tag{82}$$

The same conclusion can be drawn when $\mathbf{p} \in N_3 \cup N_4$, and the proof is intrinsically the same. Hence, we conclude the proof of this lemma. □

Finally, due to Lemma E.1, we have that $\|\mathbf{p} - \mathbf{p}^{\star\star}\|_2 \leq 1/C \cdot \mathcal{G}(\mathbf{p})$. Therefore, when $\mathcal{G}(\mathbf{p}) = \mathcal{O}(\delta)$, we also have $\|\mathbf{p} - \mathbf{p}^{\star\star}\|_2 = \mathcal{O}(\delta)$, i.e., the price vector is already in a $\mathcal{O}(\delta)$ neighborhood of the SNE. This completes the proof of Theorem 6.1. □

## Appendix F Supporting lemmas

**Lemma F.1 (Convergence of price to reference price)** *Let $\{\mathbf{p}^t\}_{t \geq 0}$ and $\{\mathbf{r}^t\}_{t \geq 0}$ be the price path and reference path generated by Algorithm 1 with non-increasing step-sizes $\{\eta^t\}_{t \geq 0}$ such that $\lim_{t \to \infty} \eta^t = 0$. Then, their difference $\{\mathbf{r}^t - \mathbf{p}^t\}_{t \geq 0}$ converges to 0 as $t$ goes to infinity.*

*Proof.* First, we recall that $D_i^t = (b_i + c_i) \cdot G_i(\mathbf{p}^t, \mathbf{r}^t)$, where $G_i(\mathbf{p}, \mathbf{r})$ is the scaled partial derivative defined in (25). Thus, it follows from Lemma F.5 that $|D_i^t| \leq (b_i + c_i)M_G$. Since $\{\eta^t\}_{t \geq 0}$ is a non-increasing sequence with $\lim_{t \to \infty} \eta^t = 0$, for any constant $\eta > 0$, there exists $T_\eta \in \mathbb{N}$ such that $|\eta^t D_i^t| \leq \eta$ for every $t \geq T_\eta$ and for every $i \in \{H, L\}$. Therefore, it holds that

$$\left| p_i^{t+1} - p_i^t \right| = \left| \mathrm{Proj}_{\mathcal{P}} \left( p_i^t + \eta^t D_i^t \right) - p_i^t \right| \leq \left| \eta^t D_i^t \right| \leq \eta, \quad \forall t \geq T_\eta, \tag{83}$$

where the first inequality is due to the property of the projection operator. Then, by the reference price update, we have for $t \geq T_\eta$ and for $i \in \{H, L\}$ that

$$\begin{aligned} \left| r_i^{t+1} - p_i^{t+1} \right| &= \left| \alpha r_i^t + (1 - \alpha)p_i^t - p_i^{t+1} \right| \\ &= \left| \alpha \left( r_i^t - p_i^t \right) + \left( p_i^{t+1} - p_i^t \right) \right| \\ &\leq \alpha \left| r_i^t - p_i^t \right| + \left| p_i^{t+1} - p_i^t \right| \\ &\leq \alpha \left| r_i^t - p_i^t \right| + \eta, \end{aligned} \tag{84}$$

where the last line follows from the upper bound in Eq. (83). Applying Eq. (84) recursively from $t$ to $T_\eta$, we further derive that

$$\begin{aligned} \left| r_i^{t+1} - p_i^{t+1} \right| &\leq \alpha^{t+1-T_\eta} \cdot \left| r_i^{T_\eta} - p_i^{T_\eta} \right| + \eta \sum_{\tau = T_\eta}^{t} \alpha^{\tau - T_\eta} \\ &\leq \alpha^{t+1-T_\eta} \cdot (\overline{p} - \underline{p}) + \frac{\eta}{1 - \alpha}, \quad \forall i \in \{H, L\}. \end{aligned} \tag{85}$$

Since $\eta$ can be arbitrarily close to 0, we have that $|r_i^t - p_i^t| \to 0$ as $t \to \infty$, which completes the proof of the convergence. □

**Lemma F.2** *Define the function $\mathcal{G}(\mathbf{p})$ as*

$$\mathcal{G}(\mathbf{p}) := \mathrm{sign}(p_H^{\star\star} - p_H) \cdot G_H(\mathbf{p}, \mathbf{p}) + \mathrm{sign}(p_L^{\star\star} - p_L) \cdot G_L(\mathbf{p}, \mathbf{p}), \tag{86}$$

*where $G_i(\mathbf{p}, \mathbf{r})$ is the scaled partial derivative introduced in Eq. (25). Then, it always holds that $\mathcal{G}(\mathbf{p}) > 0, \forall \mathbf{p} \in \mathcal{P}^2 \backslash \{\mathbf{p}^{\star\star}\}$, where $\mathbf{p}^{\star\star} = (p_H^{\star\star}, p_L^{\star\star})$ denotes the unique SNE, and the function $\mathrm{sign}(\cdot)$ is defined in Eq. (27).*

*Furthermore, for every $\epsilon > 0$, there exists $M_\epsilon > 0$ such that $\mathcal{G}(\mathbf{p}) \geq M_\epsilon$ if $\varepsilon(\mathbf{p}) \geq \epsilon$, where $\varepsilon(\mathbf{p})$ is the weighted $\ell_1$-distance function defined in Eq. (24).*

*Proof.* Firstly, by the first-order condition at the SNE (see Eq. (16)), we have $G_H(\mathbf{p}^{\star\star}, \mathbf{p}^{\star\star}) = G_L(\mathbf{p}^{\star\star}, \mathbf{p}^{\star\star}) = 0$, which also implies $\mathcal{G}(\mathbf{p}^{\star\star}) = 0$. Then, we recall the definition of four regions $N_1, N_2, N_3$, and $N_4$ in Eq. (28). We show that $\mathcal{G}(\mathbf{p}) > 0$ when $\mathbf{p}$ belongs to either one of the four regions.

1. When $\mathbf{p} \in N_1$, i.e., $p_H > p_H^{\star\star}$ and $p_L \geq p_L^{\star\star}$, the function $\mathcal{G}(\mathbf{p})$ becomes

$$
\begin{aligned}
\mathcal{G}(\mathbf{p}) &= -G_H(\mathbf{p}, \mathbf{p}) - G_L(\mathbf{p}, \mathbf{p}) \\
&= -\frac{1}{(b_H + c_H)p_H} - \frac{1}{(b_L + c_L)p_L} - \big(d_H(\mathbf{p}, \mathbf{p}) + d_L(\mathbf{p}, \mathbf{p})\big) + 2 \\
&= -\frac{1}{(b_H + c_H)p_H} - \frac{1}{(b_L + c_L)p_L} + d_0(\mathbf{p}, \mathbf{p}) + 1 \\
&= -\frac{1}{(b_H + c_H)p_H} - \frac{1}{(b_L + c_L)p_L} + \frac{1}{1 + \exp(a_H - b_H p_H) + \exp(a_L - b_L p_L)} + 1,
\end{aligned}
$$
(87)

where $d_0(\mathbf{p}, \mathbf{r}) = 1 - d_H(\mathbf{p}, \mathbf{r}) - d_L(\mathbf{p}, \mathbf{r})$ denotes the no purchase probability. We observe from Eq. (87) that $\mathcal{G}(\mathbf{p})$ is strictly increasing in $p_H$ and $p_L$. Together with the fact that $p_H$ and $p_L$ are lowered bounded by $p_H^{\star\star}$ and $p_L^{\star\star}$, respectively, and $\mathcal{G}(\mathbf{p}^{\star\star}) = 0$, we verify that $\mathcal{G}(\mathbf{p}) > 0$ when $\mathbf{p} \in N_1$. With the similar approach, we show that when $\mathbf{p} \in N_3$, i.e., $p_H < p_H^{\star\star}$ and $p_L \leq p_L^{\star\star}$, it also follows that $\mathcal{G}(\mathbf{p}) > 0$.

2. When $\mathbf{p} \in N_2$, i.e., $p_H \leq p_H^{\star\star}$ and $p_L > p_L^{\star\star}$, the function $\mathcal{G}(\mathbf{p})$ becomes

$$
\begin{aligned}
\mathcal{G}(\mathbf{p}) &= G_H(\mathbf{p}, \mathbf{p}) - G_L(\mathbf{p}, \mathbf{p}) \\
&= \frac{1}{(b_H + c_H)p_H} - \frac{1}{(b_L + c_L)p_L} + d_H(\mathbf{p}, \mathbf{p}) - d_L(\mathbf{p}, \mathbf{p}) \\
&= \frac{1}{(b_H + c_H)p_H} - \frac{1}{(b_L + c_L)p_L} + \frac{\exp(a_H - b_H p_H) - \exp(a_L - b_L p_L)}{1 + \exp(a_H - b_H p_H) + \exp(a_L - b_L p_L)}.
\end{aligned}
$$
(88)

By Eq. (88), we notice that $\mathcal{G}(\mathbf{p})$ under region $N_2$ is strictly decreasing in $p_H$ and strictly increasing in $p_L$. Meanwhile, since $p_H \leq p_H^{\star\star}$, $p_L > p_L^{\star\star}$, and $\mathcal{G}(\mathbf{p}^{\star\star}) = 0$, it implies that $\mathcal{G}(\mathbf{p}) > 0$ when $\mathbf{p} \in N_2$. Moreover, by similar reasoning, we show that when $\mathbf{p} \in N_4$, i.e., $p_H \geq p_H^{\star\star}$ and $p_L < p_L^{\star\star}$, the inequality $\mathcal{G}(\mathbf{p}) > 0$ also holds.

Finally, we are left to establish the existence of $M_\epsilon$. It suffices to show that

$$
\min_{\varepsilon(\mathbf{p})=\epsilon_1} \mathcal{G}(\mathbf{p}) > \min_{\varepsilon(\mathbf{p})=\epsilon_2} \mathcal{G}(\mathbf{p}).
$$
(89)

for every $\epsilon_1 > \epsilon_2 > 0$. Suppose $\mathbf{p}^{\epsilon_1} := \arg\min_{\varepsilon(\mathbf{p})=\epsilon_1} \mathcal{G}(\mathbf{p})$. Define $\mathbf{p}^{\epsilon_2} := (\epsilon_2/\epsilon_1)\mathbf{p}^{\epsilon_1} + (1 - \epsilon_2/\epsilon_1)\mathbf{p}^{\star\star}$, which satisfies that $\varepsilon(\mathbf{p}^{\epsilon_2}) = \epsilon_2$. Then, we have that

$$
\begin{aligned}
\mathcal{G}(\mathbf{p}^{\epsilon_2}) &= \text{sign}(p_H^{\star\star} - p_H^{\epsilon_2}) \cdot G_H(\mathbf{p}^{\epsilon_2}, \mathbf{p}^{\epsilon_2}) + \text{sign}(p_L^{\star\star} - p_L^{\epsilon_2}) \cdot G_L(\mathbf{p}^{\epsilon_2}, \mathbf{p}^{\epsilon_2}) \\
&\overset{(\Delta_1)}{=} \text{sign}\left(\frac{\epsilon_2}{\epsilon_1}(p_H^{\star\star} - p_H^{\epsilon_1})\right) \cdot G_H(\mathbf{p}^{\epsilon_2}, \mathbf{p}^{\epsilon_2}) + \text{sign}\left(\frac{\epsilon_2}{\epsilon_1}(p_L^{\star\star} - p_L^{\epsilon_1})\right) \cdot G_L(\mathbf{p}^{\epsilon_2}, \mathbf{p}^{\epsilon_2}) \\
&= \text{sign}(p_H^{\star\star} - p_H^{\epsilon_1}) \cdot G_H(\mathbf{p}^{\epsilon_2}, \mathbf{p}^{\epsilon_2}) + \text{sign}(p_L^{\star\star} - p_L^{\epsilon_1}) \cdot G_L(\mathbf{p}^{\epsilon_2}, \mathbf{p}^{\epsilon_2}) \\
&\overset{(\Delta_2)}{\leq} \text{sign}(p_H^{\star\star} - p_H^{\epsilon_1}) \cdot G_H(\mathbf{p}^{\epsilon_1}, \mathbf{p}^{\epsilon_1}) + \text{sign}(p_L^{\star\star} - p_L^{\epsilon_1}) \cdot G_L(\mathbf{p}^{\epsilon_1}, \mathbf{p}^{\epsilon_1}) = \mathcal{G}(\mathbf{p}^{\epsilon_1}),
\end{aligned}
$$
(90)

where $(\Delta_1)$ follows from substituting $p_i^{\epsilon_2}$ in $\text{sign}(p_i^{\star\star} - p_i^{\epsilon_2})$ with $p_i^{\epsilon_2} = (\epsilon_2/\epsilon_1)p_i^{\epsilon_1} + (1 - \epsilon_2/\epsilon_1)p_i^{\star\star}$. To see why $(\Delta_2)$ holds, recall that we have shown in Eq. (87) and Eq. (88) that when two prices are in the same region (see four regions defined in Eq. (28)), the price closer to $\mathbf{p}^{\star\star}$ in terms of the metric $\varepsilon(\cdot)$ has a greater value of $\mathcal{G}(\mathbf{p})$. Since $\mathbf{p}^{\epsilon_1}$ and $\mathbf{p}^{\epsilon_2}$ are from the same region and $\epsilon_1 > \epsilon_2$, we conclude that $\min_{\varepsilon(\mathbf{p})=\epsilon_1} \mathcal{G}(\mathbf{p}) = \mathcal{G}(\mathbf{p}^{\epsilon_1}) > \mathcal{G}(\mathbf{p}^{\epsilon_2}) \geq \min_{\varepsilon(\mathbf{p})=\epsilon_2} \mathcal{G}(\mathbf{p})$. $\square$

**Lemma F.3** *Define function $\mathcal{H}(\mathbf{p})$ as follows*

$$
\mathcal{H}(\mathbf{p}) := (b_H + c_H) \cdot G_H(\mathbf{p}, \mathbf{p}) \cdot (p_H^{\star\star} - p_H) + (b_L + c_L) \cdot G_L(\mathbf{p}, \mathbf{p}) \cdot (p_L^{\star\star} - p_L)
$$
(91)

*Then, there exist $\gamma > 0$ and a open set $U_\gamma \ni \mathbf{p}^{\star\star}$ such that*

$$
\mathcal{H}(\mathbf{p}) \geq \gamma \cdot \|\mathbf{p} - \mathbf{p}^{\star\star}\|_2^2, \quad \forall \mathbf{p} \in U_\gamma.
$$
(92)

*Proof.* According to the partial derivatives in Eq. (98a) and Eq. (98b) from Lemma F.5, we have

$$
\begin{aligned}
\frac{\partial G_i(\mathbf{p}, \mathbf{p})}{\partial p_i} &= -\frac{1}{(b_i + c_i)p_i^2} - b_i \cdot d_i(\mathbf{p}, \mathbf{p}) \cdot \big(1 - d_i(\mathbf{p}, \mathbf{p})\big); \\
\frac{\partial G_i(\mathbf{p}, \mathbf{p})}{\partial p_{-i}} &= b_{-i} \cdot d_i(\mathbf{p}, \mathbf{p}) \cdot d_{-i}(\mathbf{p}, \mathbf{p}).
\end{aligned}
$$
(93)

Then, to compute the gradient $\nabla \mathcal{H}(\mathbf{p}) = [\partial \mathcal{H}(\mathbf{p})/\partial p_H, \partial \mathcal{H}(\mathbf{p})/\partial p_L]$, we utilize partial derivatives of $G_i(\mathbf{p}, \mathbf{p})$ in Eq. (93) and obtain the partial derivatives of $\mathcal{H}(\mathbf{p})$ for $i \in \{H, L\}$:

$$
\begin{aligned}
\frac{\partial \mathcal{H}(\mathbf{p})}{\partial p_i} =& (b_i + c_i) \left[ \frac{\partial G_i(\mathbf{p}, \mathbf{p})}{\partial p_i} (p_i^{\star\star} - p_i) - G_i(\mathbf{p}, \mathbf{p}) \right] + (b_{-i} + c_{-i})(p_{-i}^{\star\star} - p_{-i}) \frac{\partial G_{-i}(\mathbf{p}, \mathbf{p})}{\partial p_i} \\
=& -\frac{p_i^{\star\star}}{p_i^2} - (b_i + c_i)\big(d_i(\mathbf{p}, \mathbf{p}) - 1\big) - b_i(b_i + c_i) \cdot d_i(\mathbf{p}, \mathbf{p}) \cdot \big(1 - d_i(\mathbf{p}, \mathbf{p})\big)(p_i^{\star\star} - p_i) \\
& + b_i(b_{-i} + c_{-i}) \cdot d_i(\mathbf{p}, \mathbf{p}) \cdot d_{-i}(\mathbf{p}, \mathbf{p}) \cdot (p_{-i}^{\star\star} - p_{-i}).
\end{aligned}
\tag{94}
$$

From the definition of $G_i(\cdot, \cdot)$ in Eq. (31) and the system equations of first-order condition in Eq. (16), we have $G_i(\mathbf{p}^{\star\star}, \mathbf{p}^{\star\star}) = 1/\big[(b_i + c_i)p_i^{\star\star}\big] + (d_i^{\star\star} - 1) = 0$, where $d_i^{\star\star} := d_i(\mathbf{p}^{\star\star}, \mathbf{p}^{\star\star})$ denotes the market share of product $i$ at the SNE. Thereby, it follows that $\nabla \mathcal{H}(\mathbf{p}^{\star\star}) = 0$.

Next, the Hessian matrix $\nabla^2 \mathcal{H}(\mathbf{p})$ evaluated at $\mathbf{p}^{\star\star}$ can be computed as

$$
\nabla^2 \mathcal{H}(\mathbf{p}^{\star\star})
$$

$$
= \begin{bmatrix}
\dfrac{\partial^2 \mathcal{H}(\mathbf{p}^{\star\star})}{\partial p_H{}^2}, & \dfrac{\partial^2 \mathcal{H}(\mathbf{p}^{\star\star})}{\partial p_H \partial p_L} \\[2ex]
\dfrac{\partial^2 \mathcal{H}(\mathbf{p}^{\star\star})}{\partial p_L \partial p_H}, & \dfrac{\partial^2 \mathcal{H}(\mathbf{p}^{\star\star})}{\partial p_L{}^2}
\end{bmatrix}
$$

$$
= \begin{bmatrix}
\dfrac{2p_H^{\star\star}}{(p_H^{\star\star})^3} + 2b_H(b_H + c_H) \cdot d_H^{\star\star}\big(1 - d_H^{\star\star}\big), & -\big[b_H(b_L + c_L) + b_L(b_H + c_H)\big] \cdot d_H^{\star\star} d_L^{\star\star} \\[2ex]
-\big[b_H(b_L + c_L) + b_L(b_H + c_H)\big] \cdot d_H^{\star\star} d_L^{\star\star}, & \dfrac{2p_L^{\star\star}}{(p_L^{\star\star})^3} + 2b_L(b_L + c_L) \cdot d_L^{\star\star}\big(1 - d_L^{\star\star}\big)
\end{bmatrix}
$$

$$
= \begin{bmatrix}
2(b_H + c_H) \cdot \big(1 - d_H^{\star\star}\big) \cdot \big[(b_H + c_H) - c_H d_H^{\star\star}\big], & -\big[b_H(b_L + c_L) + b_L(b_H + c_H)\big] \cdot d_H^{\star\star} d_L^{\star\star} \\[2ex]
-\big[b_H(b_L + c_L) + b_L(b_H + c_H)\big] \cdot d_H^{\star\star} d_L^{\star\star}, & 2(b_L + c_L) \cdot \big(1 - d_L^{\star\star}\big) \cdot \big[(b_L + c_L) - c_L d_L^{\star\star}\big]
\end{bmatrix}.
$$

Note that the last equality results again from substituting in the identity $G_i(\mathbf{p}^{\star\star}, \mathbf{p}^{\star\star}) = 1/\big[(b_i + c_i) \cdot p_i^{\star\star}\big] + (d_i^{\star\star} - 1) = 0$.

The diagonal entries of $\nabla^2 \mathcal{H}(\mathbf{p}^{\star\star})$ are clearly positive. For $i \in \{H, L\}$, define $k_i := b_i/(b_i + c_i) \in [0, 1]$. Then, the determinant of $\nabla^2 \mathcal{H}(\mathbf{p}^{\star\star})$ can be computed as follows:

$$
\begin{aligned}
&\det\left(\nabla^2 \mathcal{H}(\mathbf{p}^{\star\star})\right) \\
=\ & 4(b_H + c_H)(b_L + c_L) \cdot \left(1 - d_H^{\star\star}\right)\left(1 - d_L^{\star\star}\right) \cdot \left[(b_H + c_H) - c_H d_H^{\star\star}\right]\left[(b_L + c_L) - c_L d_L^{\star\star}\right] \\
& - \left(\left[b_H(b_L + c_L) + b_L(b_H + c_H)\right] \cdot d_H^{\star\star} d_L^{\star\star}\right)^2 \\
=\ & (b_H + c_H)^2(b_L + c_L)^2 \cdot \left(4\left(1 - d_H^{\star\star}\right)\left(1 - d_L^{\star\star}\right) \cdot \left[1 - (1 - k_H)d_H^{\star\star}\right]\left[1 - (1 - k_L)d_L^{\star\star}\right]\right. \\
& \qquad\qquad\qquad\qquad \left. - (k_H + k_L)^2 \left(d_H^{\star\star} d_L^{\star\star}\right)^2\right) \\
\overset{(\Delta_1)}{\geq}\ & 4(b_H + c_H)^2(b_L + c_L)^2 \cdot \left(\left(1 - d_H^{\star\star}\right)\left(1 - d_L^{\star\star}\right) \cdot \left[1 - (1 - k_H)d_H^{\star\star}\right]\left[1 - (1 - k_L)d_L^{\star\star}\right] - \left(d_H^{\star\star} d_L^{\star\star}\right)^2\right) \\
\overset{(\Delta_2)}{\geq}\ & 4(b_H + c_H)^2(b_L + c_L)^2 \cdot d_H^{\star\star} d_L^{\star\star}\left(\left[1 - (1 - k_H)d_H^{\star\star}\right]\left[1 - (1 - k_L) \cdot d_L^{\star\star}\right] - d_H^{\star\star} d_L^{\star\star}\right) \\
\overset{(\Delta_3)}{\geq}\ & 4(b_H + c_H)^2(b_L + c_L)^2 \cdot d_H^{\star\star} d_L^{\star\star}\left[\left(1 - d_H^{\star\star}\right)\left(1 - d_L^{\star\star}\right) - d_H^{\star\star} d_L^{\star\star}\right] \\
=\ & 4(b_H + c_H)^2(b_L + c_L)^2 \cdot d_H^{\star\star} d_L^{\star\star}\left(1 - d_H^{\star\star} - d_L^{\star\star}\right) \\
\overset{(\Delta_4)}{>}\ & 0,
\end{aligned}
$$
(95)

where inequalities respectively result from the following facts $(\Delta_1)$: $k_H + k_L \leq 2$; $(\Delta_2)$: $1 - d_i^{\star\star} > d_{-i}^{\star\star}$ for $i \in \{H, L\}$; $(\Delta_3)$: $1 - k_i \leq 1$ for $i \in \{H, L\}$; $(\Delta_4)$: $d_H^{\star\star} + d_L^{\star\star} < 1$. Thus, we conclude that $\nabla^2 \mathcal{H}(\mathbf{p}^{\star\star})$ is positive definite.

By the continuity of $\nabla^2 \mathcal{H}(\mathbf{p})$, there exists some constant $\gamma > 0$ and a open set $U_\gamma \ni \mathbf{p}^{\star\star}$ such that $\nabla^2 \mathcal{H}(\mathbf{p}) \succeq 2\gamma I_2, \forall \mathbf{p} \in U_\gamma$, where $I_2$ is the $2 \times 2$ identity matrix. Using the second-order Taylor expansion at $\mathbf{p}^{\star\star}$, for all $\mathbf{p} \in U_\gamma$, there exists $\widetilde{\mathbf{p}} \in U_\gamma$ such that

$$
\begin{aligned}
\mathcal{H}(\mathbf{p}) &= \mathcal{H}(\mathbf{p}^{\star\star}) + \nabla\mathcal{H}(\mathbf{p}^{\star\star}) \cdot (\mathbf{p} - \mathbf{p}^{\star\star}) + \frac{1}{2}(\mathbf{p} - \mathbf{p}^{\star\star})^\top \cdot \nabla^2\mathcal{H}(\widetilde{\mathbf{p}}) \cdot (\mathbf{p} - \mathbf{p}^{\star\star}) \\
&= \frac{1}{2}(\mathbf{p} - \mathbf{p}^{\star\star})^\top \cdot \nabla^2\mathcal{H}(\widetilde{\mathbf{p}}) \cdot (\mathbf{p} - \mathbf{p}^{\star\star}) \\
&\geq \frac{1}{2}(\mathbf{p} - \mathbf{p}^{\star\star})^\top \cdot 2\gamma I_2 \cdot (\mathbf{p} - \mathbf{p}^{\star\star}) \\
&= \gamma\|\mathbf{p} - \mathbf{p}^{\star\star}\|_2^2,
\end{aligned}
$$
(96)

where the second equality arises from that $\mathcal{H}(\mathbf{p}^{\star\star}) = 0$ and $\nabla\mathcal{H}(\mathbf{p}^{\star\star}) = 0$. $\qquad\square$

**Lemma F.4** *For any product $i \in \{H, L\}$, let*

$$
\widehat{G}_i(\mathbf{p}) := \mathrm{sign}\left(p_i^{\star\star} - p_i\right) \cdot G_i(\mathbf{p}, \mathbf{p}),
$$
(97)

*where $G_i(\mathbf{p}, \mathbf{r})$ is the scaled partial derivative defined in Eq. (25). Then, $\widehat{G}_i(\mathbf{p})$ is always increasing as $|p_i^{\star\star} - p_i|$ increases, and*

1. *when $\mathbf{p} \in N_1 \cup N_3$ (see the definition in Eq. (28)), $\widehat{G}_i(\mathbf{p})$ is decreasing as $|p_{-i}^{\star\star} - p_{-i}|$ increases;*

2. *when $\mathbf{p} \in N_2 \cup N_4$, $\widehat{G}_i(\mathbf{p})$ is increasing as $|p_{-i}^{\star\star} - p_{-i}|$ increases.*

*Proof.* Without loss of generality, consider the case that product $i = H$ and product $-i = L$.

Apparently, we have $\mathrm{sign}\left(p_H^{\star\star} - p_H\right) \leq 0$ in $N_1 \cup N_4$, and $\mathrm{sign}\left(p_H^{\star\star} - p_H\right) \geq 0$ in $N_2 \cup N_3$ by definitions (see Eq. (28)). On the other hand, by Eq. (98b) in Lemma F.5, it holds that $\partial G_H(\mathbf{p}, \mathbf{p})/\partial p_H < 0$ and $\partial G_H(\mathbf{p}, \mathbf{p})/\partial p_L > 0, \forall \mathbf{p} \in \mathcal{P}^2$. Thus,

1. in $N_1 \cup N_4$, $G_H(\mathbf{p}, \mathbf{p})$ is decreasing as $|p_H^{\star\star} - p_H|$ increases; in $N_2 \cup N_3$, $G_H(\mathbf{p}, \mathbf{p})$ is increasing as $|p_H^{\star\star} - p_H|$ increases;

2. in $N_1 \cup N_2$, $G_H(\mathbf{p}, \mathbf{p})$ is increasing as $|p_L^{\star\star} - p_L|$ increases; conversely, $G_H(\mathbf{p}, \mathbf{p})$ is decreasing as $|p_L^{\star\star} - p_L|$ increases in $N_3 \cup N_4$.

The final results directly follows by combining the above pieces together. □

**Lemma F.5** *Let $G_i(\mathbf{p}, \mathbf{r})$ be the scaled partial derivative defined in Eq. (25), then partial derivatives of $G_i(\mathbf{p}, \mathbf{r})$ with respect to $\mathbf{p}$ and $\mathbf{r}$ are given as*

$$\frac{\partial G_i(\mathbf{p}, \mathbf{r})}{\partial p_i} = -\frac{1}{(b_i + c_i)p_i^2} - (b_i + c_i) \cdot d_i(\mathbf{p}, \mathbf{r}) \cdot \big(1 - d_i(\mathbf{p}, \mathbf{r})\big); \tag{98a}$$

$$\frac{\partial G_i(\mathbf{p}, \mathbf{r})}{\partial p_{-i}} = (b_{-i} + c_{-i}) \cdot d_i(\mathbf{p}, \mathbf{r}) \cdot d_{-i}(\mathbf{p}, \mathbf{r}); \tag{98b}$$

$$\frac{\partial G_i(\mathbf{p}, \mathbf{r})}{\partial r_i} = c_i \cdot d_i(\mathbf{p}, \mathbf{r}) \cdot \big(1 - d_i(\mathbf{p}, \mathbf{r})\big); \tag{98c}$$

$$\frac{\partial G_i(\mathbf{p}, \mathbf{r})}{\partial r_{-i}} = -c_{-i} \cdot d_i(\mathbf{p}, \mathbf{r}) \cdot d_{-i}(\mathbf{p}, \mathbf{r}). \tag{98d}$$

*Meanwhile, $G_i(\mathbf{p}, \mathbf{r})$ and its gradient are bounded as follows*

$$\big|G_i(\mathbf{p}, \mathbf{r})\big| \leq M_G, \quad \big\|\nabla_{\mathbf{r}} G_i(\mathbf{p}, \mathbf{r})\big\|_2 \leq \ell_r, \quad \forall \mathbf{p}, \mathbf{r} \in \mathcal{P}^2 \text{ and } \forall i \in \{H, L\}, \tag{99}$$

*where the upper bound constant $M_G$ and the Lipschitz constant $\ell_r$ are defined as*

$$M_G := \max\left\{ \frac{1}{(b_H + c_L)\underline{p}}, \frac{1}{(b_L + c_L)\underline{p}} \right\} + 1, \quad \ell_r := \frac{1}{4}\sqrt{c_H^2 + c_L^2}. \tag{100}$$

*Proof.* We first verify the partial derivatives from Eq. (98a) to Eq. (98d):

$$\frac{\partial G_i(\mathbf{p}, \mathbf{r})}{\partial p_i} = \frac{1}{(b_i + c_i)p_i^2} + \frac{\partial d_i(\mathbf{p}, \mathbf{r})}{\partial p_i}$$

$$= \frac{1}{(b_i + c_i)p_i^2} - \frac{(b_i + c_i) \cdot \exp\big(u_i(p_i, r_i)\big) \cdot \Big(1 + \exp\big(u_{-i}(p_{-i}, r_{-i})\big)\Big)}{\Big(1 + \exp\big(u_i(p_i, r_i)\big) + \exp\big(u_{-i}(p_{-i}, r_{-i})\big)\Big)^2}$$

$$= \frac{1}{(b_i + c_i)p_i^2} - (b_i + c_i) \cdot d_i(\mathbf{p}, \mathbf{r}) \cdot \big(1 - d_i(\mathbf{p}, \mathbf{r})\big). \tag{101}$$

$$\frac{\partial G_i(\mathbf{p}, \mathbf{r})}{\partial p_{-i}} = \frac{\partial d_i(\mathbf{p}, \mathbf{r})}{\partial p_{-i}}$$

$$= \frac{(b_{-i} + c_{-i}) \cdot \exp\big(u_i(p_i, r_i)\big) \cdot \exp\big(u_{-i}(p_{-i}, r_{-i})\big)}{\Big(1 + \exp\big(u_i(p_i, r_i)\big) + \exp\big(u_{-i}(p_{-i}, r_{-i})\big)\Big)^2}$$

$$= (b_{-i} + c_{-i}) \cdot d_i(\mathbf{p}, \mathbf{r}) \cdot d_{-i}(\mathbf{p}, \mathbf{r}).$$

Then, the partial derivatives with respect to $\mathbf{r}$, as shown in Eq. (98c) and Eq. (98d), can be similarly computed.

In the next part, we show that $G_i(\mathbf{p}, \mathbf{r})$ is bounded for $\mathbf{p}, \mathbf{r} \in \mathcal{P}^2$:

$$\big|G_i(\mathbf{p}, \mathbf{r})\big| = \left| \frac{1}{(b_i + c_i)p_i} + d_i(\mathbf{p}, \mathbf{r}) - 1 \right|$$

$$\leq \left| \frac{1}{(b_i + c_i)p_i} \right| + \big|d_i(\mathbf{p}, \mathbf{r}) - 1\big| \tag{102}$$

$$\leq \frac{1}{(b_i + c_i)\underline{p}} + 1.$$

Hence, it follows that $\left|G_i(\mathbf{p}, \mathbf{r})\right|$ for every $i \in \{H, L\}$ is upper bounded by

$$\left|G_i(\mathbf{p}, \mathbf{r})\right| \leq \max \left\{ \frac{1}{(b_H + c_L)\underline{p}}, \frac{1}{(b_L + c_L)\underline{p}} \right\} + 1 =: M_G, \quad \forall \mathbf{p}, \mathbf{r} \in \mathcal{P}^2, \forall i \in \{H, L\}. \quad (103)$$

Finally, we demonstrate that $\|\nabla_{\mathbf{r}} G_i(\mathbf{p}, \mathbf{r})\|$ is also bounded $\forall \mathbf{p}, \mathbf{r} \in \mathcal{P}^2$ and $\forall i \in \{H, L\}$. From Eq. (98c) and Eq. (98d), we have

$$\begin{aligned}
\left\|\nabla_{\mathbf{r}} G_i(\mathbf{p}, \mathbf{r})\right\|_2^2 &= \left( c_i \cdot d_i(\mathbf{p}, \mathbf{r}) \cdot \left(1 - d_i(\mathbf{p}, \mathbf{r})\right) \right)^2 + \left( - c_{-i} \cdot d_i(\mathbf{p}, \mathbf{r}) \cdot d_{-i}(\mathbf{p}, \mathbf{r}) \right)^2 \\
&= c_i^2 \cdot \left( d_i(\mathbf{p}, \mathbf{r}) \cdot \left(1 - d_i(\mathbf{p}, \mathbf{r})\right) \right)^2 + c_{-i}^2 \cdot \left( d_i(\mathbf{p}, \mathbf{r}) \cdot d_{-i}(\mathbf{p}, \mathbf{r}) \right)^2 \qquad (104) \\
&\leq \frac{1}{16} \left( c_i^2 + c_{-i}^2 \right),
\end{aligned}$$

where the inequality follows from the fact that $x \cdot y \leq 1/4$ for any two numbers such that $x, y > 0$ and $x + y \leq 1$. Thus, it follows that $\left\|\nabla_{\mathbf{r}} G_i(\mathbf{p}, \mathbf{r})\right\|_2 \leq (1/4)\sqrt{c_H^2 + c_L^2} := l_r, \forall \mathbf{p}, \mathbf{r} \in \mathcal{P}^2$ and $\forall i \in \{H, L\}$. $\qquad \square$

## Limitaions

This is a theoretical work that concerns with algorithm design in a competitive market, with the primary goal of learning a stable equilibrium. The consumer demand follows the multinomial logit model, and we assume that their price/reference price sensitivities are positive, i.e., $b_i, c_i > 0$. The feasible price range $\mathcal{P} = [\underline{p}, \overline{p}]$ is assumed to contain the unique SNE, with the lower bound $\underline{p} > 0$. We have justified both assumptions in the main body of the paper (see the discussion below Eq. (3) and Proposition 3.1).

The convergence results in the paper assume the firms take diminishing step-sizes $\{\eta^t\}_{t \geq 0}$ with $\sum_{t=0}^{\infty} \eta^t = \infty$, which is common in the literature of online games (see, e.g., [15, 14, 40]). We also discuss the extension to constant step-sizes in Remark 5.3.

