# OpenReview forum: "No-Regret Learning in Dynamic Competition with Reference Effects Under Logit Demand"
_NeurIPS.cc/2023/Conference — NeurIPS 2023 poster_

### Official Review · Reviewer_aAW7 · 2023-07-01

**Soundness:** 3 good
**Presentation:** 3 good
**Contribution:** 2 fair
**Rating:** 6
**Confidence:** 3

**Summary:**

The paper studies gradient descent dynamics in duopoly competitions with reference effects and logit demand.
Convergence results are proven to show that Online Projected Gradient Ascent (OPGA) with decreasing step size converges to a stationary Nash equilibrium. This is a novel result requiring new analysis because the considered game is neither convex nor strongly monotone or variationally stable. The setup and results extend the ones of (Golrezaei et al., 2020) which however considered linear demands and a uniform reference price.

**Strengths:**

- The paper is well written and the results are sound and original
- The authors did a good job of connecting the considered games with the existing literature and explaining why existing results do not apply
- The proven theorems are novel and the analysis seems somewhat original.

**Weaknesses:**

- The analysis is very specific to the considered duopoly games. Although they are widely studied in the Marketing and Management literature, they may not be of high interest to the machine learning community. That said, online learning and convergence to NE is a relevant topic within NeurIPS.
- The authors repeatedly refer to no-regret (even in the title). However, the notion of regret is never defined and -- moreover -- bounds on the regret are not provided. I agree that last-iterate convergence to NE is a stronger guarantee, but it is not clear to me that this implies sublinear regret.


**Questions:**

How realistic is it for the competing firms to compute exact gradients? Which set of information is required to do so at each round?
Also, as pointed out by the authors, one-point gradient estimates could be obtained. Does the current analysis allow draw conclusions on the resulting convergence results?

**Limitations:**

Limitations are discussed in combination with future directions.

---

> ### Author Rebuttal · Authors · 2023-08-06
>
> We thank the reviewer for the positive feedback. Please find our responses below. We hope they address your concerns and provide further clarity.
> > W1: The analysis is ... relevant topic within NeurIPS.
>
> A: We understand your concern and agree that deriving convergence results for general online games is a meaningful research direction. Meanwhile, we believe that analyzing specific instances with practical importance is also a great contribution to the community. In our competitive framework, we consider a popular demand model, namely the MNL, which is a famous choice model from [R1] and has been empirically validated for its good representation of consumer purchasing behavior. Additionally, the general convergence results for online games are often built on certain assumptions, such as concavity [R2], strong monotonicity [R3], or variational stability [R4]. Hence, the general results usually fail or become weaker when applied to problems without corresponding properties, such as our problem. Yet, by leveraging the distinctive properties of the MNL model and developing a new line of analysis, we demonstrate that the global convergence and a $\Theta(1/t)$ rate still hold for our problem under minimal assumptions (the market feedback mechanism). For the above reasons, we believe our work aligns well with the scope of NeurIPS in terms of both practical significance and theoretical analysis.
> > W2: The authors repeatedly refer to no-regret ... sublinear regret.
>
> A: We appreciate the reviewer for pointing out the source of confusion. We will include the following definition of regret in our revised paper.
>
> In a competitive framework, the ***regret*** of player $i$ is a quantity used to measure the performance of an online learning algorithm, which is the difference between the total reward of the best fixed action in hindsight and the realized total reward of player $i$ over $T$ periods. Formally, the regret of player $i$ at period $T$ is defined as $$R_i^T:= \max_{p_i \in \mathcal{P}} \bigg\\{\sum_{t = 1}^T \Pi_i\big((p_i,p_{-i}^t), \mathbf{r}^t\big)\bigg\\} -\sum_{t=1}^T \Pi_i\big((p_i^t,p_{-i}^t), \mathbf{r}^t\big).$$
>
> Then, an algorithm is said to be ***no-regret*** if $R_i^T$ grows sub-linearly with respect to $T$ for every player $i$. The established results in our paper imply that OPGA is no-regret, with reasoning as follows:
> * The convergence of the OPGA algorithm to the SNE (see Theorem 5.1 of the paper) guarantees that $\lim_{t \rightarrow \infty} (\mathbf{p}^t, \mathbf{r}^t) = (\mathbf{p}^{\star\star}, \mathbf{r}^{\star\star})$. Hence, when $T$ is sufficiently large, by the definition of SNE, the best fixed price in hindsight for firm $i$ is the SNE price $p_i^{\star\star}$ (note that when $p_{-i}^{t}\rightarrow p_{-i}^{\star\star}$ and $\mathbf{r}^t\rightarrow \mathbf{p}^{\star\star}$, then $p_i^{\star\star}$ is the unique optimal price for firm $i$).
> *  In the meanwhile, since the revenue functions are Lipschitz continuous in prices and reference prices, we have that $\lim_{t\rightarrow\infty}\big\\{\Pi_i(\mathbf{p}^{\star\star},\mathbf{r}^{\star\star})-\Pi_i(\mathbf{p}^t,\mathbf{r}^t)\big\\}\rightarrow 0$. Therefore, it follows that $\lim_{T\rightarrow \infty} {R_i^T}/{T}  = 0$, as $T\rightarrow \infty, \ \forall i\in \\{H, L\\},$ which indicates that $R_i^T = o(T)$, i.e., the regret is sublinear.
>
> Finally, we remark that our result is stronger than merely being no-regret, as the no-regret alone does not guarantee convergence at all, let alone the convergence to SNE. In fact, the players may exhibit entirely unpredictable and chaotic behaviors under a no-regret policy [R5], with the only exception being the finite game, where players only compete for finitely many rounds (note that our problem does NOT fall under this category).
>
> > Q1: How realistic ... compute exact gradients? Which set of info ... round?
>
> A: Thank you for your comments. Computing the gradient $D_i^t$ in Eq. (9) of the paper is feasible and realistic in practice. For firm $i$, the required information includes its previously posted price $p_i^t$, its own sensitivities to price and reference price $(b_i, c_i)$ (which can be estimated from historical data), and its demand from the last period. We note it is reasonable to assume that the firm has access to the aforementioned information, as it exclusively pertains to its internal data, and not that of its competitor. This also aligns with the opaque market setup in our paper, where both computing gradient $D_i^t$ and implementing the OPGA do not require firms to have any knowledge of their competitors.
> > Q2: Also, as pointed out ... convergence results?
>
> A: Our current analysis is based on the exact gradient, and we recognize that it may be more practical to use noisy first-order feedback or even zeroth-order feedback. In terms of difficulties, we believe that the generalization to the noisy first-order oracle is relatively straightforward, where the main difference would be an extra error term in the analysis, causing the price path to converge to a neighborhood of SNE, whose size is determined by the magnitude of noises. Yet, the extension to the zeroth-order oracle might be a meaningful future work. In addition, we hope to highlight that the core of this paper is to show that firms can achieve a stable equilibrium while safeguarding their private data. Our convergence results, even with an exact first-order oracle, are highly non-trivial. The techniques developed for the convergence analysis are both original and innovative.
>
> ### Reference
> [R1] Conditional logit analysis of qualitative choice behavior.
>
> [R2] Bandit learning in concave N-person games.
>
> [R3] Optimal no-regret learning in strongly monotone games with bandit feedback.
>
> [R4] Learning in games with continuous action sets and unknown payoff functions.
>
> [R5] Multiplicative weights update with constant step-size in congestion games: Convergence, limit cycles and chaos.

---

> > ### Comment · Reviewer_aAW7 · 2023-08-13
> >
> > Thank you for your rebuttal and explanations.
> > About the regret, I was hoping that a sublinear *bound* could also be derived as a function of $T$ (and not only asymptotically), e.g., $R(T) \leq C\sqrt{T}$ for some appropriate problem-dependent constant $C$. This is typical in the existing online learning literature, where last-iterate convergence is instead harder to guarantee.
> >
> > I will keep my score.

---

> > > ### Author Response · Authors · 2023-08-13
> > > **Replying to Further Comment by Reviewer aAW7**
> > >
> > > Thank you very much for your reply and your extremely helpful comment! Upon checking the proof, we realize that our sublinear regret bound can also be derived as a function of $T$ (and not only asymptotically) as you indicated. Indeed, our Theorem 5.2 (last-iterate convergence rate in terms of price and reference price) already guarantees the $O(\sqrt{T})$ regret bound. We attach the proof below and will add this as a corollary to the revised paper.
> > >
> > > For the ease of notation, we denote $f^t(\cdot)=\Pi_i((\cdot, p_{-i}^t), \mathbf{r}^t)$. Then, we have that $R_i^T=\sum_{t=1}^T\left[f^t(p_i^{\star\star})-f^t(p_i^t)\right]$. By Theorem 5.2 in our paper, we have $|p_i^{\star\star}-p_i^t|=O(1/\sqrt{t})$. Hence, if $f^t(\cdot)$ is $\ell$-Lipschitz continuous $\forall t\geq 0$ for some $\ell>0$, we can derive that
> > > $$R_i^T\leq\sum_{t=1}^T\ell |p_i^{\star\star}-p_i^t|=O\left(\sum_{t=1}^T\ell/\sqrt{t} \right)=O(\sqrt{T}),$$which is exactly the regret bound suggested by the reviewer.
> > >
> > > The only point remaining is justifying the Lipschitz continuity of $f^t(\cdot)$ for all $t$. This can be directly done by computing the derivative of the revenue function: $$\dfrac{\partial\Pi_i(\mathbf{p},\mathbf{r})}{\partial p_i}=\dfrac{\partial (p_i\cdot d_i(\mathbf{p},\mathbf{r}))}{\partial p_i} = d_i(\mathbf{p},\mathbf{r})-p_i\cdot (b_i+c_i)\cdot d_i(\mathbf{p},\mathbf{r})\cdot (1-d_i(\mathbf{p},\mathbf{r})),$$
> > > where the computation in the last step above is similar to our proof of Lemma E.5. Hence, provided that $p_i\in \mathcal{P}=[\underline{p},\overline{p}]$, we have that $$\left|\dfrac{\partial\Pi_i(\mathbf{p},\mathbf{r})}{\partial p_i}\right|\leq 1+ \overline{p}\cdot(b_i+c_i)/4,$$ where we use the fact that $x\cdot(1-x)\leq 1/4$ for any $x\in [0,1]$. Hence, it suffices to choose $\ell = 1+ \overline{p}\cdot(b_i+c_i)/4$.
> > >
> > > Combining together, we have shown that $R_i^T\leq C\sqrt{T}$ for some constant $C$. We thank the reviewer again for the insightful comment, which we believe will further enrich the content of our paper. We hope this response provides further clarification : )

---

### Official Review · Reviewer_8Smn · 2023-07-06

**Soundness:** 3 good
**Presentation:** 3 good
**Contribution:** 3 good
**Rating:** 6
**Confidence:** 3

**Summary:**

The author consider a multi-period pricing competition problem between two firms, dubbed H and L. Each of these firms sells 1 product. At each time step, the demands on each product is governed by an MNL model, which is paramaterized by not only the firms' offered prices in the current time period, but also the current reference prices of the firms. These reference prices are dependent on the past prices. In the game theoretic setting, each of the firm is to set prices under the uncertainty on the other firms strategy. The authors' main result is that, when each of the firms applies the standard online gradient descent algorithm, they both converge to a stationary Nash equilibrium, and the rate of convergence is also provided.

**Strengths:**

The pricing problem combines both the game theoretic element in a duopoly pricing setting, as well as the reference price setting, which are both well known in the literature. The provided algorithm is sensible and reasonable, and these results are supported with numerical experiments. The analysis in the gradient descent algorithm seems interesting, since it seems to deviate from the traditional Online Gradient Descent works that require convexity.

**Weaknesses:**

- The model assumption on the demand model seems quite strong, since it requires the random noise to follow a particular probability distribution (Gumbel), in order for the result to hold. While there is another linearity assumption on the utility, I find it a milder assumption than the Gumbel assumption. While I understand that the Gumbel distribution, which gives rise to the MNL model, is a common assumption in assortment optimization, I find that it could be a limiting assumption in the pricing setting. Crucially, if the noise distribution is changed, it is not quite clear of the OPGA still converges to a stationary Nash equilibrium.

- While Theorem 5.2 is a more refined form than Theorem 5.1 in that the former provides a finite time performance guarantee, I am not clear on what is the hidden parameter in the $\Theta(\cdot)$ notation for the learning rate $\eta_t$. While the OPGA can be implemented with any sequence of learning rates, it is desirable to have a set of default learning rates that come with an explicit performance guarantee.

**Questions:**

(1) Can the authors provide a discussion on how easy or how difficult it is to generalize from the Gumbel distribution noise to a more general class of noise (for example, 1-subGassuian)?

(2) As shown in Figure 1a, b, the choice of learning rate is rather important. Can the authors demonstrate a plot with the setting of learning rate $\eta^t = \Theta(1/t)$ in Theorem 5.2? I think it will be more informative than setting $\eta^t = 1/\sqrt{t}$ in Figure 1a. In addition, can the authors provide explicit expressions for the coefficients $d_r, d_p$ in Theorem 5.2, on how they depend on the MNL assumption and the model parameters?

**Limitations:**

This is a theoretical work, there is minimal negative societal impact.

---

> ### Author Rebuttal · Authors · 2023-08-04
>
> Thank you for your insightful comments. We hope the responses below address your concerns. We highly appreciate a re-evaluation of our work and a kind reconsideration of the review score.
> > W1: The model assumption ... equilibrium.
>
> A: We agree with you that the Gumbel noise is somewhat restrictive. Yet the MNL is by far the most popular choice model, due to its elegant closed-form expression for market share. Beyond the assortment problem, the MNL choice model and its variants (e.g. nested logit) have also been widely studied in the pricing literature. We kindly refer the reviewer to [R1-- R9] for notable works on the pure pricing problem under the MNL and its variants.
> > W2: While Theorem 5.2 ... performance guarantee.
>
> A: We appreciate the reviewer's advice about having a default learning rate that guarantees the $\Theta(1/t)$ convergence rate. The hidden parameter in the choice of $\eta^t=\Theta(1 / t)$, denoted by $d_\eta$, hinges on Eq.(72):$$(\gamma d_\eta-1)d_p\geq C_3.(\*)$$
>
> Recall that $\gamma$ is defined in Lemma E.3 and $C_3>0$ is defined in Eq.(68). Hence, if $\gamma d_\eta-1>0$, there exists suitable $d_p$ to make Eq.$(*)$ hold, which subsequently guarantees the $\Theta(1/t)$ convergence rate. Therefore, analytically, firms can adopt any step-size $\eta^t=d_\eta/(t+1)$ with $d_\eta>1/\gamma$. The only subtlety lies in $\gamma$, whose exact value depends on both firms' parameters and the SNE market share (Eq.(89)) and the derivation afterward). Below, we describe a mechanism for firms to obtain a suitable step-size without disclosing private information.
>
> It suffices to find a lower bound $\underline\gamma\in(0,\gamma)$ and let the firms agree on the choice $d_\eta=1/\underline\gamma>1/\gamma$. By Eq.(89), we have$$\gamma\geq\min_i\{b_i+c_i\}\cdot\min_i\{d_i^{\star\star}\}\cdot(1-d_H^{\star \star}-d_L^{\star \star}).(\**)$$
>
> In Prop 3.1, we establish lower and upper bounds for $p_i^{\star\star}$. For brevity, denote Eq.(6) as $m_i<p_i^{\star\star}<M_i$, and the utility at SNE as $u_i=a_i-b_ip_i^{\star\star}$. Then, each firm $i$ can bound $u_i$ by $u_i^m:=a_i-b_iM_i<u_i<a_i-b_im_i=:u_i^M$. Instead of sharing the true values, each firm $i$ can disclose an arbitrary lower bound $m_i^{bc}>0$ for $b_i+c_i$, and two arbitrary numbers $\underline u_i^m$ and $\overline u_i^M$ with $\underline u_i^m\leq u_i^m$ and $\overline u_i^M\geq u_i^M$. Using Eq $(**)$, the two firms can compute that $$\gamma\geq\min_i\\{m_i^{bc}\\}\cdot\min_i\left\\{\frac{\exp(\underline u_i^m)}{1+\exp(\underline u_i^m)+\exp(\overline u_{-i}^M)}\right\\}\cdot\frac{1}{1+\exp(\overline u_{H}^M)+\exp(\overline u_{L}^M)}=:\underline\gamma.$$
>
> Thus, firms can obtain the constant $d_\eta = 1/\underline{\gamma}$ without disclosing their information.
> > Q1: Can the authors ... (for example, 1-subGassuian)?
>
> A: We kindly refer the reviewer to **General Response Q3** for background on choice models and the popularity of Gumbel noise. Shifting from Gumbel noise introduces challenges. For example, multivariate Gaussian noise results in the probit model, while arbitrary noise distributions lead to the mixed logit model. Both lack closed-form expressions for market share [R10], thereby making the properties of their revenue functions undefined and elusive. Further, changes in noise distribution can significantly affect equilibrium behavior in price competition [R5]. Specifically, while MNL models guarantee the existence and uniqueness of a Nash equilibrium (NE) [R11], such NE may not even exist in models with general noise distributions [R5]. This absence creates significant hurdles, as the existence of NE is fundamental for achieving a stable equilibrium.
> Extending noise distributions beyond Gumbel is an intriguing research direction, with the key first step may be to show that NE exists under distributions of interest.
> > Q2: As shown in Figure 1a, b, ... parameters?
>
> A: We performed additional experiments with $\eta^t=\Theta(1/t)$ (see Figure 2 in the rebuttal PDF), and we will add this figure to the revised paper.
> Below, we elaborate on the expressions of $d_p$ and $d_r$. According to Eq.$(\*)$, $d_p$ depends on the choice of $d_\eta$. Without loss of generality, let $d_\eta=2/\gamma$ (it is easy to observe $d_\eta = O(1/\gamma)$ yields the best $d_p$). By Eq.$(*)$, it suffices to take $d_p = C_3$ (see definition in Eq.(68)), which further implies that $d_p\leq C_1(d_\eta)^2+d_\eta kd_{rp}$ (the definitions of $C_1,k,d_{rp}$ can be found in Eqs.(54), (59), and (65)). After unfolding the definitions and using simple algebras, we observe that $$C_1=O(\sum_{i} (b_i+c_i)^2/\underline{p}^2)$$$$k= O((c_H+c_L)^2\sum_{i} (b_i+c_i)^2/\gamma)$$
> $$d_{rp}=O\left(\frac{1+\alpha^2C_1}{(1-\alpha^2)^2\gamma^2} + \frac{3+\alpha^2}{1-\alpha^2}(\bar{p}-\underline{p})^2\right).$$
>
> Combining the above pieces, we obtain that $$d_p = O\left(\frac{B^2}{\gamma^2(1-\alpha^2)}\left[\frac{B}{\gamma^2\underline{p}^2(1-\alpha^2)}+(\bar{p}-\underline{p})^2 \right]\right),$$
>
> where $B=\sum_i(b_i^2+c_i^2)$. Finally, as $d_r=2d_p+2d_{rp}$ (see Eq.(76)), we conclude that $d_r$ has the same order as $d_p$.
> ### Reference
> [R1] Optimal bundle pricing.
>
> [R2] Product line selection and pricing with modularity in design.
>
> [R3] Pricing multiple products with the multinomial logit and nested logit models: Concavity and implications.
>
> [R4] Optimal pricing for a multinomial logit choice model with network effects.
>
> [R5] Price competition under mixed multinomial logit demand functions.
>
> [R6] Multiproduct price optimization and competition under the nested logit model with product-differentiated price sensitivities.
>
> [R7] Dynamic pricing of perishable assets under competition.
>
> [R8] Optimal pricing of correlated product options under the paired combinatorial logit model.
>
> [R9] Optimizing Risk-Balancing Return Under Discrete Choice Models.
>
> [R10] Discrete choice methods with simulation.
>
> [R11] Discrete choice theory of product differentiation.

---

> ### Author Response · Authors · 2023-08-14
>
> Thank you again for your insightful questions. We hope our further clarifications have addressed your concerns, and we'll include them in the revised paper. We are more than happy to discuss any additional concerns you may have. Looking forward to your reply : )

---

### Official Review · Reviewer_M19q · 2023-07-06

**Soundness:** 3 good
**Presentation:** 4 excellent
**Contribution:** 3 good
**Rating:** 7
**Confidence:** 4

**Summary:**

The authors' objective is to develop an algorithm to aid the firms in converging to a stationary Nash equilibrium (SNE). In pursuit of this, the authors have:

* Proposed an online projected gradient ascent (OPGA) algorithm.
* Proven the global convergence of the OPGA to SNE within the given problem setting.
* Established the convergence rate of the proposed algorithm to SNE.

**Strengths:**

I commend the authors for their clear and engaging writing. The background, related literature, concepts, algorithms, and more, are all explained lucidly. The proposed algorithm, and the results derived from it, are noteworthy contributions to the field. The technical addition to extend the algorithm's convergence beyond linear demand and convex loss function is particularly valuable.

**Weaknesses:**

I noticed certain assumptions and explanations that might require further substantiation. Please refer to the "questions section" for a detailed account. I believe addressing these points could add considerable depth and credibility to the paper's findings and arguments.

**Questions:**

Despite the significant merits of the work, I do have a few queries. If the authors could address these, I would be more than pleased to highly rate the paper:

* My first question might be rather straightforward. Could you please elaborate on why online mirror descent, proposed to solve a similar problem, does not apply in your problem setting? If online mirror descent does apply, what advantages does your OPGA algorithm offer?
* As for the assumptions listed above, I'm somewhat puzzled:
    - The last assumption posits that each firm can access its own demand from the last period. However, I would argue that this "demand" differs from the one defined by
$$d_i(\bm{p}^t, \bm{r}^t) = \frac{\exp(u_i(p_i^t, r_i^t))}{1+\exp(u_i(p_i^t, r_i^t)) + \exp(u_{-i}(p_{-i}^t, r_{-i}^t))}.$$
which, to the best of my understanding, represents the market share of firm $i$ at period $t$. While the "demand" that firm $i$ can access using the proposed method is the quantity of product sold by firm $i$, such as 1000 electronic devices. Aren't these two different concepts? I may be overlooking something, but it would be helpful if you could clarify the potential connection. It would also be instructive if you could provide an example explaining the use of $d_i(\bm{p}^t, \bm{r}^t)$ in the numerical example.
    - It is assumed that $b_i$ and $c_i$ are known to firm $i$. However, I believe these parameters can only be learned from data. Therefore, what is the role of the learning process in the convergence to SNE? Do we handle them as a two-stage problem, or can we combine the two learning processes?

---

> ### Author Rebuttal · Authors · 2023-08-06
>
> We thank the reviewer for the insightful questions. Please find our responses below. We highly appreciate your re-evaluation of our work and a kind reconsideration of the review score.
> > Q1: Discussion about online mirror descent.
>
> A: For consistency, we assume a maximization form and use the term Online Mirror Ascent (OMA).
> * Firstly, we clarify that we did not eliminate the application of OMA to our problem. We are aware that OMA is a widely used online learning method, and OPGA is indeed a special case of OMA by using the Euclidean distance as the Bregman divergence. We've conducted numerical tests on other OMA variants (see Figure 1 in the rebuttal PDF), such as the multiplicative weight update (MWU). The results suggest that OMA could very likely converge to the SNE for our problem.
> * We choose OPGA due to its simplicity, efficiency, and the geometric structure of our problem. The general OMA is known for its capability in exploiting specific geometric structures of the problem (e.g. when the feasible region is a probability simplex). However, the feasible price region in our problem is just a rectangle, making the simple gradient update a natural and intuitive choice. Indeed, our numerical results also demonstrate that OPGA and MWU show comparable performances with OPGA exhibiting less fluctuant price updates at the early stages. Besides, with OPGA, we do not suffer from the complexity of evaluating certain complicated mirror maps.
> * Lastly, we discuss a potential way to generalize our proof to OMA. Let $\Phi(x)$ be a $\kappa$-strongly convex function and define the associated Bregman divergence as $D_\Phi(x,y)=\Phi(x)-\Phi(y)-\langle\nabla\Phi(y),x-y\rangle$. The OMA update can be written as$$p_i^{t+1}=\text{Proj}_{\mathcal{P}}[(\nabla\Phi)^{-1}(\nabla\Phi(p_i^t)+\eta^t D_i^t)].$$
>
>     Recall that Eq. (29) and Eq. (52) are two key inequalities we use for the convergence of OPGA. Now, we derive a corresponding inequality for OMA and use Bregman divergence to measure the convergence. By the generalized Pythagoras identity, we have$$D_\Phi(p_i^{\star\star},p_i^{t+1})\leq D_\Phi(p_i^{\star\star},p_i^t)-D_i(p_i^t,p_i^{t+1})+\eta^tD_i^t(p_i^{\star\star}-p_i^t+p_i^t-p_i^{t+1}).$$
>
>     Applying the strong convexity to $D_i(p_i^t,p_i^{t+1})$ and maximizing the right-hand side (RHS) w.r.t. $(p_i^t-p_i^{t+1})$, we derive$$D_\Phi(p_i^{\star\star},p_i^{t+1})-D_\Phi(p_i^{\star\star},p_i^t)\leq \eta^tD_i^t(p_i^{\star\star}-p_i^t)+\frac{(\eta^tD_i^t)^2}{2\kappa}. (\*)$$
>
>     Similar to the proof in our paper, the RHS of $(\*)$ is controlled by the first term when $\eta^t$ is small. In addition, when $\mathbf{r}^t$ and $\mathbf{p}^t$ are close, we have $D_i^t\approx(b_i+c_i)\cdot G_i(\mathbf{p}^t,\mathbf{p}^t)$. Hence, by Lemma E.2, we can show from $(\*)$ that the price of at least one product $i$ will strictly approach the SNE during the update unless it is already close to $p_i^{\star\star}$. After $\mathbf{p}^t$ enters a neighborhood of SNE with a small enough step-size, we consider the sum of Eq. $(\*)$ for two products. By Lemma E.3, we can show that the RHS of the summation is non-positive (compare to Eq. (52)-(54)), which implies that the price will stay in the neighborhood.
>
>     While there may exist subtleties to work out, we believe this proof sketch is a promising starting point for proving the convergence of OMD. A comprehensive study on this topic will be part of our future work.
> > Q2-1: Connection between demand (quantity sold) and market share.
>
> Due to the length limit in the response for each reviewer, we kindly refer you to our **General Response Q1 and Q2**, where we clarify the assumption and the connection between demand and market share.
> > Q2-2: Estimation of $b_i$ and $c_i$.
>
> A: Thank you for your insightful feedback. We believe both approaches you mentioned are feasible. Below, we elaborate on the estimation process.
> * When treated as a two-stage problem, the firms can estimate $b_i,c_i$ from historical data. This falls in the domain of MNL parameter estimation, a well-explored topic in economics. The most common method is maximizing the log-likelihood, i.e., the product of market share for all firms. Notably, [R1] shows that the log-likelihood function for MNL is globally concave in parameters $(\mathbf{a}, \mathbf{b}, \mathbf{c})$. This nice concavity property facilitates the computational algorithms. Specifically, several prominent algorithms (i.e., (1) Newton-Raphson, (2) BHHH-2 [R3], and (3) the steepest ascent) studied in Chapter 8 of [R2] are all guaranteed to get an improved estimation at each iteration. We refer the reviewer to [R2] for a more detailed analysis of the strengths and drawbacks of each algorithm.
> * Second, the learning process of $b_i, c_i$ can also be integrated with the OPGA aglorithm. Indeed, existing papers like [R4, R5] have done so for the monopoly setting. The benefit of this method is that little historical data is required, and the firms can refine their estimations along the repeated competition. Meanwhile, the firms may incur additional regret during the exploration stage. However, as long as firms can estimate $b_i, c_i$ accurately, our existing analysis ensures that the firm can reach the SNE. Hence, the overall regrets for both firms are still sublinear.
>
>     Finally, we remark that, when $b_i$ and $c_i$ must be estimated from data, it is more practical to assume that firms can obtain a noisy estimation for the derivatives. Though our current analysis is based on the exact gradient, we believe the generalization to the noisy first-order feedback is a promising yet interesting future research.
>
> ### Reference
> [R1] Conditional logit analysis of qualitative choice behavior.
>
> [R2] Discrete choice methods with simulation.
>
> [R3] Estimation and inference in nonlinear structural models.
>
> [R4] Multi-product dynamic pricing in high-dimensions with heterogeneous price sensitivity.
>
> [R5] Demand learning and pricing for varying assortments.

---

> > ### Author Response · Authors · 2023-08-20
> > **Additional Response to Reviewer M19q**
> >
> > We thank the reviewer again for the comments. We've been continuously thinking about the second question you raised these days, and we would like to share our new thought with you. We believe this will enrich the content of our paper and provide further justification for our assumption. We highly appreciate your time in reading our response and your kind reconsideration of the review score.
> >
> > As we explained in our previous response, there are multiple ways to obtain the market share and estimate the gradient. Yet, a practical issue to consider is the approximation accuracy: the estimation is likely to bring noises to the gradient, but our current analysis assumes the exact gradient. Now, we have rigorously proved that: **if the noises are bounded by some constant $\delta>0$, the price and reference price would converge to a neighborhood with radius $\mathcal{O}(\delta)$ of the SNE**. This is a typical type of conclusion in optimization with noisy first-order oracles. The proof is built on our current analysis, and we detail the roadmap below.
> >
> > Recall that, the basis of our proof for Theorem 5.1 is Eq. (29), which shows that the price path would steadily approach the SNE if it stays in the same quadrant defined in Eq. (28). Subsequently, through a contradiction-based argument, we demonstrate that even when the price path does not stay in the same quadrant, it will still converge toward the SNE along the boundary regions. In both parts, our proof relies on the properties of $G_i(\mathbf{p}^t,\mathbf{p}^t)$ (e.g., Lemma E.2), which serves as a good approximation for the scaled derivative $G_i(\mathbf{p}^t,\mathbf{r}^t)$ provided that $\mathbf{p}^t$ and $\mathbf{r}^t$ are close. The only difference under the noisy first-order oracle is that, besides the difference term $||\mathbf{p}^t - \mathbf{r}^t||_2$, we will have an additional error term proportional to $\delta$ in Eq. (29) and subsequent steps. Therefore, if $\delta$ has the same order as $||\mathbf{p}^t - \mathbf{r}^t||_2$, our current analysis is directly applicable and the convergence follows. More practically, when $\delta$ is a fixed threshold, some steps in our analysis can fail if $\delta$ dominates the quantity $G_i(\mathbf{p}^t,\mathbf{p}^t)$. However, if this happens, we can show that the current price is already pretty close to the SNE.
> >
> > More precisely, what we need to prove is that $\mathcal{G}(\mathbf{p})=\mathcal{O}(\delta)$ also implies $||\mathbf{p} - \mathbf{p}^{\star\star}||_2=\mathcal{O}(\delta)$ (recall that $\mathcal{G}(\mathbf{p})$ is a sum of $G_i(\mathbf{p}, \mathbf{p})$ defined in Eq. (80)).
> >
> > This can be viewed as a refinement of current Lemma E.2: instead of showing that $\mathcal{G}(\mathbf{p}) \geq M_{\epsilon}$ if $\varepsilon(\mathbf{p}) \geq \epsilon$, we can show that
> >
> > $$\mathcal{G}(\mathbf{p})\geq C\cdot||\mathbf{p}-\mathbf{p}^{\star\star}||_2.$$
> >
> > Hence, this ensures that $||\mathbf{p}-\mathbf{p}^{\star\star}||_2 = \mathcal{O}(\delta)$ as long as $\mathcal{G}(\mathbf{p}) = \mathcal{O}(\delta)$. The above inequality can be derived from the current proof of Lemma E.2. Suppose $p_H>p_H^{\star \star} \text { and } p_L \geq p_L^{\star \star}$. Similar as Eq. (81), we have that
> >
> > $$\mathcal{G}(\mathbf{p})=\mathcal{G}(\mathbf{p})-\mathcal{G}(\mathbf{p}^{\star\star})=\sum_{i}\dfrac{1}{b_i+c_i}\left(\dfrac{1}{p_i^{\star\star}}-\dfrac{1}{p_i}\right) + d_0(\mathbf{p},\mathbf{p})-d_0(\mathbf{p}^{\star\star},\mathbf{p}^{\star\star}).$$
> >
> > By definition, it is easy to observe that $d_0(\mathbf{p},\mathbf{p})-d_0(\mathbf{p}^{\star\star},\mathbf{p}^{\star\star})>0$. Hence, we have that $$\mathcal{G}(\mathbf{p})>\sum_{i}\dfrac{1}{b_i+c_i}\cdot\dfrac{p_i-p_i^{\star\star}}{p_i^{\star\star}p_i}\geq\sum_{i}\dfrac{1}{b_i+c_i}\cdot\dfrac{p_i-p_i^{\star\star}}{p_i^{\star\star}\overline{p}}=\sum_i \dfrac{|p_i-p_i^{\star\star}|}{(b_i+c_i)p_i^{\star\star}\overline{p}}.$$
> > The right-hand side of the above equation is a weighted $\ell_1$ distance between $\mathbf{p}$ and $\mathbf{p}^{\star\star}$. By the equivalence of norms in Euclidean space, we know there exists some constant $C$ such that $\mathcal{G}(\mathbf{p})\geq C\cdot ||\mathbf{p}-\mathbf{p}^{\star\star}||_2$. When $\mathbf{p}$ belongs to other quadrants, we can derive the same lower bound for $\mathcal{G}(\mathbf{p})$ like above. For example, if $p_H\leq p_H^{\star \star} \text { and } p_L \geq p_L^{\star \star}$, the derivation is similar to Eq. (82). This concludes the proof.
> >
> > In summary, when the level of noise is small compared to the gradient, our current analysis is still applicable, showing that the price and reference price must converge toward the SNE. If the size of noise becomes comparable to the gradient, the above argument demonstrates that the price path is already within a $\mathcal{O}(\delta)$ neighborhood of the SNE, where $\delta$ characterizes the size of noises.

---

> > > ### Comment · Reviewer_M19q · 2023-08-20
> > >
> > > I am truly appreciative of the authors' comprehensive explanation concerning the observation of market share. In the context of pricing or assortment optimization, the Multinomial Logit (MNL) model is commonly employed to delineate the purchasing probabilities of customers. Typically, non-purchases are not directly observed, which raises some concerns in estimation of the model. However, online retailers like Amazon have the capacity to monitor non-purchases, thereby allowing for a more robust estimation of the MNL model. In light of this, I would kindly suggest that the authors consider framing their research in settings where the observation of market size is both relevant and accurate. Until then, I retain my reservations regarding the assumption of known gross market size.
> > >
> > > Nonetheless, I am in full agreement with the authors on their diligent efforts to address all posed questions and to furnish rigorous proofs, particularly those involving noise gradients. I would be pleased to see the authors include the underlying ideas of these proofs as an extension in the final version of the paper. If this is done, I would be inclined to elevate my review score to a 7 and recommend the paper for acceptance.

---

> > > > ### Author Response · Authors · 2023-08-20
> > > > **Further Response to Reviewer M19q**
> > > >
> > > > We are truly grateful to the reviewer for carefully considering our responses and mentioning the relevant application where the size of non-purchases can be observed. In terms of the revision, we are not allowed to upload an updated manuscript to the system due to the NeurIPS policy this year (not the same as the last year). However, we are more than happy to formalize the idea of noise gradient in the extension section of the paper and send it to you in our next response. If our paper is fortunate enough to be accepted, we will incorporate this extension section into the camera-ready version, as it greatly enhances the credibility and practicality of the OPGA algorithm. Since the discussion period is ending tomorrow, we will try our best to prepare a draft today and will further calibrate the writings later. We would also greatly appreciate it if the reviewer could kindly change the review score and give us the opportunity to incorporate this extension in the final version. We sincerely thank the reviewer for helping us improve the quality of our paper.

---

> > > > > ### Comment · Reviewer_M19q · 2023-08-20
> > > > >
> > > > > Thank you very much for your thoughtful consideration of incorporating our suggested changes into the final version of your paper. I would like to bring to your attention that the example concerning "the relevant application where the size of non-purchases can be observed" does not factor in competition, which is a critical element in your current work. Therefore, I eagerly look forward to seeing how the authors might devise their own application setting that not only captures this aspect of competition but also substantiates the assumptions and justifications for their model.
> > > > >
> > > > > In summary, the review process has illuminated three principal questions:
> > > > > 1, The feasibility of obtaining the market share and the exact gradient.
> > > > > 2, The strong assumptions embedded in the Multinomial Logit (MNL) model, particularly concerning the Gumbel distribution error term.
> > > > > 3, The relationship between last-iteration convergence and sub-linear regret.
> > > > > Upon a careful review of the authors' responses, it is my understanding that they have sufficiently addressed these questions. In light of their satisfactory explanations and their commitment to finalize the paper as promised, I would like to formally recommend the paper for acceptance.

---

> > > > > > ### Author Response · Authors · 2023-08-20
> > > > > >
> > > > > > We want to express our sincere gratitude to the reviewer for allowing us to further improve our manuscript!!! We are currently writing the extension section, and hopefully, we can share our draft with you by the end of the day. We thank the reviewer for mentioning the subtlety of the example in the competition setting. We will also think carefully about the relevant application.

---

> > > > > > > ### Author Response · Authors · 2023-08-21
> > > > > > > **Extension Section and Proof (1/3)**
> > > > > > >
> > > > > > > We thank the reviewer again for re-evaluating our contribution! As promised, we attach our draft for the extension section to be included in the final version of the paper. Please see the three responses below.
> > > > > > >
> > > > > > > ## **6 Extension**
> > > > > > > In previous sections, we assume that each firm $i$ possesses accurate information regarding its sensitivity parameters $(b_i, c_i)$ as well as its realized market share $d_i^t$ in each period of the competition. This is equivalent to having access to an exact first-order oracle. However, another more practical setting worth mentioning is that the firm can only obtain a rough approximation for its market share and need to estimate the sensitivities from historical data. This would bring extra noise to the computation of the first-order derivative $D_i^t$ in Eq. (9). In this section, we first elaborate on the feasibility of estimating the market share and sensitivities from realistic data. Then, we discuss the impact of a noisy first-order oracle on the convergence results.
> > > > > > >
> > > > > > > We consider the approximation of market share $d_i^t$ and the calibration of sensitivities $(b_i,c_i)$ under both ***uncensored*** and ***censored*** data cases. With uncensored data, both purchase and no-purchase data are available. This is the case in real-world competitions where two firms sell substitutes on a third-party online retailing platform such as Amazon. The firms grant the platform permission to track non-purchase statistics by monitoring consumers who visited its website but did not purchase any product. In this uncensored case, the overall market size can be directly estimated, allowing the firm to obtain the market share, given that its sale quantity is accessible through market feedback. Moreover, the sensitivity parameters can be easily calibrated using the classical maximum likelihood estimation (MLE) method, with prominent algorithms including Newton-Raphson, BHHH-2 [R1], and the steepest ascent. Due to the concavity of the log-likelihood function for the MNL model with respect to its parameters, these numerical algorithms are guaranteed to improve at each iteration and achieve fast convergence [R2, R3].
> > > > > > >
> > > > > > > Meanwhile, it is common that the traditional transaction data only captures the realized demand (i.e., the quantity sold), while non-purchases are not directly observed in practice. This scenario is referred to as the censored data case. In this case, the sensitives and the total market size of the MNL model can be estimated via the generalized expectation-maximization (GEM) gradient method proposed by [R2], which is an iterative algorithm that draws inspiration from the expectation-maximization (EM) approach. The capability to approximate the sensitivity parameters, the market share, and thereby the gradient $D_i^t$, enhances the credibility and practicability of the OPGA algorithm, suggesting its great potential for broader applications in retailing.
> > > > > > >
> > > > > > > In the presence of approximation errors, firms cannot precisely compute the derivative $D_i^t$. Hence, the convergence results in Section 5 are not directly applicable. Indeed, if the noises are disruptive enough, one can anticipate Algorithm 1 to possibly not show any convergent behavior. However, if the noises are uniformly bounded by some small number $\delta$, the following theorem demonstrates that both the price and reference price paths converge to a $\mathcal{O}(\delta)$-neighborhood of the unique SNE. This is a typical type of conclusion in optimization with noisy first-order oracles.
> > > > > > >
> > > > > > > **Theorem 6.1 (Noisy first-order oracle)** Suppose both firms adopt Algorithm 1 and they have access to a noisy first-order oracle such that $|D_i^t - {\partial \log (\Pi_i(\mathbf{p}^t, \mathbf{r}^t))}/{\partial p_i}|\leq \delta$, $\forall i\in \\{H,L\\}, t\geq 0$. Let the step-sizes $\\{\eta^t\\}\_{t\geq 0}$ be a non-increasing sequence such that $\lim_{t\rightarrow \infty} \eta^t = 0$ and $\sum_{t=0}^{\infty} \eta^t = \infty$ hold. Then, the price paths reference price paths generated by Algorithm 1 converge to a neighborhood with radius $\mathcal{O}(\delta)$ of the unique SNE.
> > > > > > >
> > > > > > > We remark that the noisy first-order oracle studied in our work is different from the stochastic gradient, which generally assumes a zero-mean noise with finite variance. In that case, it is possible to derive the convergence to a limiting point in expectation or with high probability. In contrast, the noise in $D_i^t$ is a kind of approximation error without any distributional properties. Therefore, with the step-sizes defined in Theorem 6.1, we expect the price and reference price paths to converge to the neighborhood of the SNE, but continue to fluctuate around the neighborhood without admitting a limiting point.
> > > > > > >
> > > > > > > ### Reference
> > > > > > > [R1] Estimation and inference in nonlinear structural models.
> > > > > > >
> > > > > > > [R2] Consumer choice models with endogenous network effects.
> > > > > > >
> > > > > > > [R3] Discrete choice methods with simulation.

---

> > > > > > > > ### Author Response · Authors · 2023-08-21
> > > > > > > > **Extension Section and Proof (2/3)**
> > > > > > > >
> > > > > > > > The proof of Theorem 6.1 is based upon the proof of Theorem 5.1. Specifically, we demonstrate that the noisy gradient still guides the price path toward the SNE if the magnitude of the true gradient dominates the noise. Conversely, if noise levels are comparable with the true gradient, we show that the price path is already close to the SNE. Since the step-sizes decrease to zero, Lemma E.1 implies that the reference price also converges to the $\mathcal{O}(\delta)$-neighborhood.
> > > > > > > >
> > > > > > > > ### **Proof of Theorem 6.1 (Appendix)**
> > > > > > > > Recall the definition of $G_i(\mathbf{p},\mathbf{r})$ from Eq. (25). We can write the noisy derivative as $D_i^t = (b_i+c_i)G_i(\mathbf{p}^t,\mathbf{r}^t) + n_i^t$, where $n_i^t$ represents the noise satisfying $|n_i^t| < \delta$, $\forall i\in \\{H,L\\}, t\geq 0$.
> > > > > > > >
> > > > > > > > We also recall the two-part proof for Theorem 5.1 (see the sketch below Theorem 5.1): in Part 1, we show that the price path and reference price paths converge towards the SNE and visit the neighborhood $N_{\epsilon}^1$ infinitely many times, for any $\epsilon >0$; in Part 2, we establish that, given $\epsilon$ is below a certain threshold $\epsilon_0$, the price vector would remain in the neighborhood $N_\epsilon^2$ after entering it with a sufficiently small step-size. Since the relative scale of $\delta$ and $\epsilon_0$ is unsure, our proof below is mainly based on Part 1.
> > > > > > > >
> > > > > > > > In Part 1, the original proof assuming the exact gradient oracle employs a contradiction-based argument. Suppose the price path does not converge to the SNE, the proof consists of the following major steps (different from the current Appendix C, below, we use $\epsilon$ in lieu of $\epsilon_0$ to avoid confusion):
> > > > > > > >
> > > > > > > > * In Eq. (29), it is demonstrated that the price path steadily approaches the SNE if it stays in the same quadrant defined in Eq. (28). The main technique we use is $\mathcal{G}(\mathbf{p}) >  M_{\epsilon}$ when $\varepsilon(\mathbf{p}) > \epsilon$. The definition of $\mathcal{G}(\mathbf{p})$ and the validation for the technique are stated in Lemma E.2.
> > > > > > > >
> > > > > > > > * In Eq. (34), we show that the price path only oscillates between adjacent quadrants provided the step-sizes are sufficiently small.
> > > > > > > >
> > > > > > > > * From Eq. (37) to Eq. (41), we prove that even when the price path does not stay in the same quadrant, it will still converge toward the SNE if it is at the boundary regions between two quadrants. The pivotal inequality employed here is again $\mathcal{G}((p_H^{\star \star}, p_L^{T_1^{\epsilon_1}})) \geq M_{\epsilon-\epsilon_1}$, provided $\varepsilon((p_H^{\star \star}, p_L^{T_1^{\epsilon_1}}))>\epsilon-\epsilon_1$ (see the end of Eq. (38) and Eq. (40)).
> > > > > > > >
> > > > > > > > * Finally, in Eq. (44) and subsequent equations, we provide supplementary justification for the above bullet that the price path remains adjacent to the boundaries given that the step-size is small. A crucial consideration here is opting for a much smaller $\epsilon_1$ relative to $\epsilon$ when defining the boundary region. In addition, the inequality $\mathcal{G}((p_H^{T_2^{\epsilon_1}}, p_L^{\star \star})) \geq M_{\widehat{\epsilon}_1}$ is also utilized in Eq. (50).
> > > > > > > >
> > > > > > > > To summarize, the key to the proof is Lemma E.2., i.e., $\mathcal{G}(\mathbf{p}) >  M_{\epsilon} >0$ as long as $\varepsilon(\mathbf{p}) > \epsilon$. By definition, $\mathcal{G}(\mathbf{p}^t)$ approximately characterizes the difference $\varepsilon(\mathbf{p}^t) - \varepsilon(\mathbf{p}^{t+1})$ as seen in Eq. (29). As a result, the effect of the lower bound $M_\epsilon$ is that, if other terms can be upper-bounded by $M_\epsilon$, we can show that the updated price is closer to the SNE. For example, in the right-hand side of Eq. (29), once the term $2 l\_r||\mathbf{r}^t-\mathbf{p}^t||\_2$ is below $M_\epsilon$, it suggests that the price vector is heading towards the SNE.
> > > > > > > >
> > > > > > > > With noisy first-order oracles, the immediate consequence is that there will always be an error term accompanying $\mathcal{G}(\mathbf{p})$. For instance, given $D_i^t = (b_i+c_i)G_i(\mathbf{p}^t,\mathbf{r}^t) + n_i^t$, we observe that Eq. (29) evolves to
> > > > > > > >
> > > > > > > > $$\varepsilon(\mathbf{p}^{t+1})\leq \varepsilon(\mathbf{p}^{t}) -\eta^t(M_{\epsilon}-2 l_r||\mathbf{r}^t-\mathbf{p}^t||_2 - \sum_i n_i^t/(b_i+c_i)).$$
> > > > > > > >
> > > > > > > > This observation is consistent across the proof. Thus, if the noise $n_i^t$ is substantially smaller than $M_\epsilon$, the proof of Theorem 5.1 remains valid, implying that the price vector converges towards the SNE. The subtlety arises when the size of $n_i^t$ is comparable with $M_\epsilon$. Under such circumstances, the analysis in both Eq. (29) and Eq. (37) to Eq. (41) becomes invalid, as the noises are substantial enough to negate any assurance that the price path strictly approaches the SNE.

---

> > > > > > > > > ### Author Response · Authors · 2023-08-21
> > > > > > > > > **Extension Section and Proof (3/3)**
> > > > > > > > >
> > > > > > > > > However, if the noise $n_i^t$ is similar in magnitude to $\mathcal{G}(\mathbf{p})$, we can show that the price vector $\mathbf{p}$ is already close to the SNE $\mathbf{p}^{\star\star}$. More precisely, since the noises are bounded by $\delta$, it is equivalent to show that $\mathcal{G}(\mathbf{p})=\mathcal{O}(\delta)$ also implies $||\mathbf{p} - \mathbf{p}^{\star\star}||_2=\mathcal{O}(\delta)$. This can be proved by a refined version of Lemma E.2.
> > > > > > > > >
> > > > > > > > > **Lemma E.2 (Refined)** Let $\mathcal{G}(\mathbf{p})$ be the function defined in Eq. (80). Then, it holds that
> > > > > > > > > $$\mathcal{G}(\mathbf{p})\geq \sum_i \dfrac{|p_i-p_i^{\star\star}|}{(b_i+c_i)p_i^{\star\star}\overline{p}},$$
> > > > > > > > > i.e., $\mathcal{G}(\mathbf{p})$ is lower-bounded by a weighted $\ell_1$ distance between $\mathbf{p}$ and $\mathbf{p}^{\star\star}$. By the equivalence of norms in the Euclidean space, there exists some constant $C$ such that $\mathcal{G}(\mathbf{p})\geq C\cdot ||\mathbf{p}-\mathbf{p}^{\star\star}||_2$.
> > > > > > > > >
> > > > > > > > > **Proof of Lemma E.2 (Refined)** Similar to the proof of the original Lemma E.2, we separately consider the four possible scenarios where $\mathbf{p}$ belongs to one of the quadrant defined in Eq. (28).
> > > > > > > > >
> > > > > > > > > * Suppose $p_H>p_H^{\star \star} \text { and } p_L \geq p_L^{\star \star}$, i.e., $\mathbf{p}\in N_1$. Since $G_i(\mathbf{p}^{\star\star}) = 0$, we have that
> > > > > > > > > $$\mathcal{G}(\mathbf{p})=\mathcal{G}(\mathbf{p})-\mathcal{G}(\mathbf{p}^{\star\star})=\sum_{i}\dfrac{1}{b_i+c_i}\left(\dfrac{1}{p_i^{\star\star}}-\dfrac{1}{p_i}\right) + d_0(\mathbf{p},\mathbf{p})-d_0(\mathbf{p}^{\star\star},\mathbf{p}^{\star\star}).$$
> > > > > > > > > By the definition of the non-purchase probability, we observe that  $d_0(\mathbf{p},\mathbf{p})-d_0(\mathbf{p}^{\star\star},\mathbf{p}^{\star\star})>0$. Hence, the above equation implies that
> > > > > > > > > $$\mathcal{G}(\mathbf{p})>\sum_{i}\dfrac{1}{b_i+c_i}\cdot\dfrac{p_i-p_i^{\star\star}}{p_i^{\star\star}p_i}\geq\sum_{i}\dfrac{1}{b_i+c_i}\cdot\dfrac{p_i-p_i^{\star\star}}{p_i^{\star\star}\overline{p}}=\sum_i \dfrac{|p_i-p_i^{\star\star}|}{(b_i+c_i)p_i^{\star\star}\overline{p}},$$
> > > > > > > > > where we use the fact that $p_i \in \mathcal{P}=[\underline{p},\overline{p}]$ and the presumption $\mathbf{p}\in N_1$.
> > > > > > > > >
> > > > > > > > > * Suppose $p_H\leq p_H^{\star \star} \text { and } p_L > p_L^{\star \star}$, i.e., $\mathbf{p}\in N_2$. Again, using the fact that $G_i(\mathbf{p}^{\star\star}) = 0$, we derive that
> > > > > > > > > $$
> > > > > > > > > \mathcal{G}(\mathbf{p}) = \dfrac{1}{b_H+c_H}\left(\dfrac{1}{p_H}-\dfrac{1}{p_H^{\star\star}}\right) + \dfrac{1}{b_L+c_L}\left(\dfrac{1}{p_L^{\star\star}}-\dfrac{1}{p_L}\right) + \left[d_H(\mathbf{p},\mathbf{p}) - d_H(\mathbf{p}^{\star\star},\mathbf{p}^{\star\star})\right]+\left[d_L(\mathbf{p}^{\star\star},\mathbf{p}^{\star\star})-d_L(\mathbf{p},\mathbf{p})\right].
> > > > > > > > > $$
> > > > > > > > > Since $\mathbf{p}\in N_2$, we have that $d_H(\mathbf{p},\mathbf{p}) - d_H(\mathbf{p}^{\star\star},\mathbf{p}^{\star\star})>0$ and $d_L(\mathbf{p}^{\star\star},\mathbf{p}^{\star\star})-d_L(\mathbf{p},\mathbf{p})>0$. Hence, similar to the first case, it follows that
> > > > > > > > > $$\mathcal{G}(\mathbf{p})>\dfrac{1}{b_H+c_H}\left(\dfrac{1}{p_H}-\dfrac{1}{p_H^{\star\star}}\right) + \dfrac{1}{b_L+c_L}\left(\dfrac{1}{p_L^{\star\star}}-\dfrac{1}{p_L}\right)\geq \sum_i \dfrac{|p_i-p_i^{\star\star}|}{(b_i+c_i)p_i^{\star\star}\overline{p}}.$$
> > > > > > > > >
> > > > > > > > > The same conclusion can be drawn when $\mathbf{p}\in N_3\cup N_4$, and the proof is intrinsically the same. Hence, we conclude the proof of Lemma E.2 (Refined).
> > > > > > > > >
> > > > > > > > > Finally, due to the refined Lemma E.2, we have that $||\mathbf{p}-\mathbf{p}^{\star\star}||_2 \leq 1/C\cdot \mathcal{G}(\mathbf{p})$. Therefore, when $\mathcal{G}(\mathbf{p}) = \mathcal{O}(\delta)$, we also have $||\mathbf{p}-\mathbf{p}^{\star\star}||_2=\mathcal{O}(\delta)$, i.e., the price vector is already in a $\mathcal{O}(\delta)$ neighborhood of the SNE. This completes the proof of Theorem 6.1.

---

> ### Author Response · Authors · 2023-08-14
>
> Thank you again for your valuable comments. We hope our response addressed your questions, and we'll incorporate them into the revised paper. If you have additional concerns, we are very delighted to discuss them further. Looking forward to hearing from you : )

---

### Official Review · Reviewer_CZUh · 2023-07-07

**Soundness:** 3 good
**Presentation:** 4 excellent
**Contribution:** 3 good
**Rating:** 6
**Confidence:** 3

**Summary:**

This paper investigates the problem of dynamic pricing in a competitive environment, where there are two revenue-maximizing competing firms selling substitutable products to customers and each of them has no access to the information of its competitor. In addition, customer’s utility follows a linear function with current price and reference prices, while the demand function is captured by a MNL model. The authors propose a simple online projected gradient ascent algorithm to update the price based on the historical prices and observed demands. The main contribution of this paper is that the authors provide the convergence guarantee of this algorithm to a unique stable Nash equilibrium and shows the convergence rate when choosing step size optimally.

**Strengths:**

1. This paper analyzes a very interesting problem --- dynamic pricing in the presences of competition across firms and reference prices of customers and provides a very succinct formulation to abstract this problem.

2. The paper is well-written and I really enjoy reading this paper even though I don't have much background on MNL model.

3. The results are promising that the authors show that a simple online projected gradient ascent algorithm works well in practice and it can converge to the unique stable NE.

4. The paper is technically solid and the proof is nontrivial to me, however, I have no expertise to determine the novelty of the proof shown in the paper compared with the related work [21].

**Weaknesses:**

In general, I find the paper interesting, technically solid and shows a very promising result. Though I still have some questions needed to be clarified.

1. I find the uniqueness of the stable NE in this paper very important and crucial to the convergence results. Can authors discuss whether this property can be generalized? In addition, with this property, can we replace online projected gradient ascent algorithm by other online optimization algorithm to achieve similar convergence rate?

2. If I understand correctly, the authors need $d_i(p^t, r^t)$ can be directly observed, right? If so, please clarify it more explicitly in the algorithm.


**Questions:**

See above.

---

> ### Author Rebuttal · Authors · 2023-08-06
>
> We thank the reviewer for appreciating our work. We hope our responses below provide further clarity.
> > S4: Comparison with the related work [21].
>
> A: We thank the reviewer for the feedback. We'd like to clarify the distinctions between our work and [21] from two perspectives.
>
> **Model Formulation:**  Our paper employs the MNL demand, which is distinct from the linear demand used in [21]. The linear demand in [21] results in a quadratic revenue function, which possesses favorable properties such as supermodularity and concavity. These properties  facilitate the convergence analysis of online games (see e.g., [R1]). On the other side, the revenue function under the MNL model lacks these properties, and it does not satisfy other useful characteristics such as cocoercive or variational stability (see the discussion in Section 2). Moreover, in Section 4 and Appendix A, we demonstrate that standard techniques from multi-agent games and dynamical systems are also not applicable to our problem, making it necessary to develop a novel analysis for the convergence of the OPGA.
>
> Besides the demand, another difference lies in the reference price formulation. In particular, the duopoly competition in [21] assumes that the two firms share a common reference price, whereas we allow firms to have different reference prices. Our formulation offers greater flexibility in modeling. We note that our analysis also applies to the scenario where two firms share a common reference price.
>
> **Proof Technique:** While both our work and [21] utilize a two-part proof for showing the asymptotic convergence, the essence of our proofs is significantly different from that of [21]. A key result in [21] is their Lemma 9.1, which ensures the following inequality holds globally (see Eq. (19) and Eq. (20) in [21] and note that their problem adopts the minimization formulation): $\sum_{i} (g_i^{\star\star}-g_i^t)(p_i^{\star\star}-p_i^t) > 0$, where $g_i$ is used to denote the derivative of their revenue function. This property is known as variational stability (see e.g., [R2]), under which the convergence results have been established for various algorithms. Our work, however, doesn't benefit from such properties. To address this, we introduce two distinct metrics respectively for the two parts of our convergence analysis. In Part 1, we divide the feasible price range into four quadrants with the SNE $\mathbf{p}^{\star\star}$ being the origin. Our proof is based on drawing contradiction: suppose that the price vector does not converge to the SNE, we can prove the following in sequence (1) the price path cannot always stay in the same quadrant; (2) when $t$ is large, the price path can only oscillate between adjacent quadrants; (3) the price path would converge to the boundaries between some adjacent quadrants; (4) afterward, the price path would stay close to the boundaries and converge to the SNE, resulting in the contradiction. In Part 2, we exploit a local property of our model (see Lemma E.3) by showing that the auxiliary function $\mathcal{H}(\mathbf{p})$ is lower-bounded by some quadratic function around the SNE. This enables us to establish our final conclusion.
>
> In summary, we believe that our paper is technically novel and significantly different from [21].
>
> > W1: I find the uniqueness of the stable NE ... by other online optimization algorithm to achieve similar convergence rate?
>
> A: Thank you for your insightful comments. The uniqueness of SNE plays an important role in the convergence analysis of this paper. This property naturally results from the MNL demand model used in this paper. Yet, we believe similar convergence results can also be derived for certain models where the SNE is not unique.
>
> In fact, we are currently considering an extension to the asymmetric reference effects, which exhibits the non-uniqueness property as the reviewer mentioned. Under asymmetric reference effects, the utility for product $i$ changes to
> $$u_i(p_i^t, r_i^t)=a_i-b_i \cdot p_i^t+c_i^{+} \cdot(r_i^t-p_i^t)_{+}+c_i^{-} \cdot(r_i^t-p_i^t)\_-,$$
>
> where the notations $(\cdot)\_{+}:=\max \\{\cdot, 0\\}$ and $(\cdot)\_{-}:=\min \\{\cdot, 0\\}$ are adopted to account for consumers' potentially asymmetric reactions to discounts and surcharges, respectively. When $c_i^- <c_i^+, \forall i\in \\{H,L\\}$ (referred to as the loss-averse scenario), it can be shown that there exists a set of SNEs. With some slight modifications of the OPGA, we have established the convergence result. The difference is that, instead of converging to a unique SNE, the algorithm converges to the set of SNEs, and the limit may depend on the initialization. Due to the length restriction on the rebuttal response, we regret that we couldn't elaborate more on this extension. If the reviewers are interested, we are delighted to share more details about this modified algorithm and analysis during the discussion period.
>
> Finally, we believe it is possible to generalize to other online learning methods, such as online mirror ascent (OMA). In Figure 1 of the rebuttal PDF, we verify that another variant of OMA, the Multiplicative Weight Update (MWU) method, also converges to the SNE. Thus, we expect the general OMA can also converge for our problem. We kindly refer the reviewer to our response to reviewer **M19q (Q1)** for a more detailed discussion on OMA and a potential method to generalize our proof.
>
>
> > W2: The authors need $d_i(p^t, r^t)$ can be directly observed.
>
> A: Yes, we have discussed this assumption in Section 3.2. In the revised paper, we'll declare this requirement more clearly in Section 4 when presenting the OPGA. Due to the length limit, we kindly refer the reviewer to **General Response Q1 and Q2** for a detailed discussion about the assumption needed for observing $d_i(\mathbf{p}^t,\mathbf{r}^t)$.
>
> ### Reference
> [R1] Bandit learning in concave N-person games.
>
> [R2] Learning in games with continuous action sets and unknown payoff functions.

---

> > ### Comment · Reviewer_CZUh · 2023-08-18
> >
> > Thanks authors for their detailed response. I am happy to see this response to appear in the next version of this paper. My score remains the same.

---

> > > ### Author Response · Authors · 2023-08-18
> > >
> > > Thank you for your reply! We will incorporate these new discussions into the revised paper.

---

### Author Rebuttal · Authors · 2023-08-06

# General Response
We would like to express our sincere gratitude to the reviewers for reading our paper and providing valuable feedback. Below, we answer two common questions and provide a background for the discrete choice model. Please find our responses to other questions in the personalized rebuttals.
> Q1: Clarification on observing market share $d_i$, and the connection between demand (quantity sold) and market share (CZUh W2, M19q Q2-1).

A: Sorry for the confusion. To observe $d_i$, each firm needs to know its own last-period demand (quantity sold), and both firms should agree on an estimated market mass. While the first point is rather intuitive, we'll elaborate on the market mass.

The realized demand and market share $d_i$ are connected via the market mass (the largest potential demand, including no-purchase quantity). For instance, if firm H sells 1000 units and firm L sells 1500, with a market mass of 5000, then their market shares are $d_H=0.2$ and $d_L=0.3$, respectively, and the no purchase market share is $d_0=1-d_H -d_L=0.5$. Hence, given the market mass, a firm can easily convert the sold quantity into its market share. We'll now justify the assumption of knowing the market mass from two points.

* First, adding market mass information is feasible and practical. Indeed, it is sufficient for the two firms to agree on an estimated market mass. Under this agreement, the market will still reach an SNE under the OPGA. We only need to account for the error in market mass estimation. This is equivalent to using the noisy first-order oracle, which is a direct extension of our current analysis. Additionally, we note that in the fields of economics and operations research, the assumption of knowing the market mass is prevalent and standard under the MNL model (see, e.g., [R6--R13]), where this stream of papers typically uses the market share as the demand function while doing the pricing and assortment optimization on the MNL model.

* Second, having the information on market mass will NOT disclose a firm's market share or demand to its competitor, thereby safeguarding the firm's privacy. Note that even with an agreement on the market mass, firms remain unaware of their rivals' market share, as the market mass encompasses the quantity of no purchases. Thus, having this piece of information will NOT compromise the objective of this paper, i.e., achieving the SNE while preserving firms' privacy.

> Q2: Accessing the gradient $D_i^t$ under the opaque market (CZUh W2, M19q Q2-1, aAW7).

A: Obtaining $D_i^t$ in Eq.(9) is identical to accessing the first-order oracle, which is a common assumption in the optimization and gradient-based online learning literature [R1,R2,R3,R4,R5]. In this paper, rather than abstractly assuming direct access to a first-order oracle, we'd like to explain it in a more concrete and interpretable way, i.e., what elements do we need to get this oracle? Indeed, to compute $D_i^t$, a firm requires only (i) its own sensitivity parameters $(b_i, c_i)$ and (ii) its last-period demand. This aligns with the opaque market setup, i.e., there's no dependency on a rival's information.

A more practical setting may be accessing a noisy first-order oracle or even zeroth-order feedback. Yet, the main message we want to convey in this paper is that under the MNL, firms can reach an SNE while protecting their privacy, and our conclusions in the context of an exact first-order oracle are already highly non-trivial. We believe further relaxing the exact first-order oracles would be a very interesting future work (in fact, the generalization to a noisy first-order oracle is likely to be straightforward, with the zero-order feedback being the exciting one).

> Q3: Background on discrete choice models and prevalence of Gumbel noise.

A: The discrete choice models, built on random utility theory [R14], help predict consumer choices among multiple options. The utility $U_i$ for product $i$ comprises a deterministic part $u_i$, and a random part $\epsilon_i$. Consumers are assumed to pick the product with the highest utility (with no-purchase utility being 0). The choice probability for product $i$ thus becomes $$P(U_i > U_j, \forall j \neq i)=P(\epsilon_j - \epsilon_i < u_i - u_j, \forall j \neq i). (\*)$$

In fact, only when $\epsilon_i$ follows the Gumbel does the market share in Eq.$(*)$ have the closed-form expression [R15]. The MNL model, while simple, offers accurate predictions and speedy parameter estimation through techniques like MLE. In contrast, models using other noise distributions lack closed-form expressions, requiring less efficient numerical simulation for parameter estimation. More detailed discussion on choice models can be found in this book [R15].
### Reference
[R1] Introductory lectures on convex optimization: A basic course.

[R2] Online first-order framework for robust convex optimization.

[R3] Learning in games with continuous action sets and unknown payoff functions.

[R4] Faster first-order methods for stochastic non-convex optimization on Riemannian manifolds.

[R5] Adaptive learning in continuous games: Optimal regret bounds and convergence to Nash equilibrium.

[R6] Discrete choice analysis: theory and application to travel demand.

[R7] Discrete choice theory of product differentiation.

[R8] The theory and practice of revenue management.

[R9] When prospect theory meets consumer choice models: Assortment and pricing management with reference prices.

[R10] Multiproduct price optimization and competition under the nested logit model with product-differentiated price sensitivities.

[R11] Pricing multiple products with the multinomial logit and nested logit models: Concavity and implications.

[R12] Product line selection and pricing with modularity in design.

[R13] Dynamic pricing and inventory control of substitute products.

[R14] Conditional logit analysis of qualitative choice behavior.

[R15] Discrete choice methods with simulation.

---

### Decision · Program_Chairs · 2023-09-21

**Decision:**

Accept (poster)

**Comment:**

I am recommending this paper for acceptance. The final scores are (7,6,6,6) --- so all reviewers are supportive. Importantly, the rebuttal and discussion phase could clarify some technical concerns of the reviewers. We would be happy to see those being incorporated in the camera-ready version.